# A Large Recurrent Action Model: xLSTM Enables Fast Inference for Robotics Tasks

## Abstract

In recent years, there has been a trend in the field of Reinforcement Learning (RL) towards large action models trained offline on large-scale datasets via sequence modeling. Existing models are primarily based on the Transformer architecture, which result in powerful agents. However, due to slow inference times, Transformer-based approaches are impractical for real-time applications, such as robotics. Recently, modern recurrent architectures, such as xLSTM and Mamba, have been proposed that exhibit parallelization benefits during training similar to the Transformer architecture while offering fast inference. In this work, we study the aptitude of these modern recurrent architectures for large action models. Consequently, we propose a Large Recurrent Action Model (LRAM) with an xLSTM at its core that comes with linear-time inference complexity and natural sequence length extrapolation abilities. Experiments on 432 tasks from 6 domains show that LRAM compares favorably to Transformers in terms of performance and speed.

## 1 Introduction

Reinforcement Learning (RL) has been responsible for impressive success stories such as game-playing [Silver et al., 2016; Vinyals et al., 2019; Berner et al., 2019; Patil et al., 2022], plasma control for fusion [Degrave et al., 2022], or navigation of stratospheric balloons [Bellemare et al., 2020]. While these successes were based on classical RL approaches, in which agents have been trained online with RL objectives, recently there has been a trend towards offline RL settings [Levine et al., 2020; Schweighofer et al., 2022] and sequence models trained via behavior cloning [Chen et al., 2021; Janner et al., 2021]. Such approaches, in which agents are trained on large-scale offline datasets with causal sequence modeling objectives, have been driven by the proliferation of Transformer-based architectures and gave rise to what we refer to as *Large Action Models* (LAMs) to highlight their similarity to large language models (LLMs) [Radford et al., 2018]. LAM approaches can also be used in multi-task settings to develop generalist agents such as Gato [Reed et al., 2022].

Existing LAMs are primarily based on the Transformer [Vaswani et al., 2017] architecture. Because of their powerful predictive performance, robotics has become an emergent application area for large models [Brohan et al., 2023b;a; Octo Model Team et al., 2024; Gu et al., 2023; Wang et al., 2023] and a number of large multi-task datasets were collected [Jia et al., 2024; Embodiment Collaboration et al., 2024; Jiang et al., 2023; Mandlekar et al., 2023]. This development bears the potential to produce robotics agents that learn to master complex tasks in a wide range of environments and even different embodiments. For example, recently it has been demonstrated, albeit in restricted settings, that sequence models trained on multi-episodic contexts can perform in-context learning (ICL) [Laskin et al., 2020; Lee et al., 2023]. One potential application of ICL can be to learn new related tasks in robotics without the need for re-training or fine-tuning.

One of the key reasons for the success of Transformer-based models is their ability to scale to large datasets through their efficient parallelization during training. However, despite numerous success stories in RL, language modeling [Brown et al., 2020] or computer vision [Dosovitskiy et al., 2021; He et al., 2022], a persistent drawback of Transformer-based architectures is their high inference cost in terms of both speed and memory [Kim et al., 2023]. Consequently, deploying Transformer-based models in resource-constrained scenarios, such as on devices with limited hardware capacity and/or real-time constraints, e.g., robots or smartphones, is prohibitive because of the required fast inference times [Firoozi et al., 2023; Hu et al., 2023]. A basic principle of control theory is that the controller

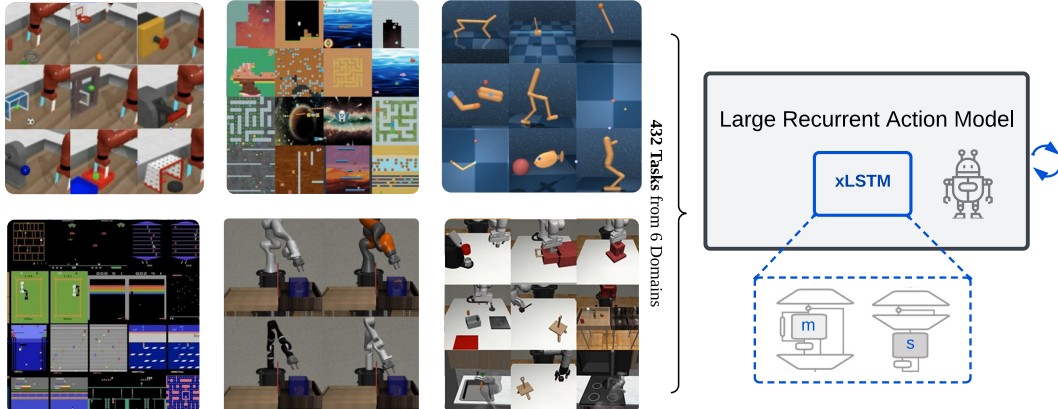

**Figure 1:** Illustration of our Large Recurrent Action Model (LRAM) with an xLSTM [Beck et al., 2024] at its core.

sample rate should be in the order of magnitude of the sample rate of the sensors [Franklin et al., 1998, Ch. 11]. To illustrate this, for typical robots such as drones or industrial robot arms rates of 100Hz-1000Hz are required to keep the system stable [Salzmann et al., 2023; El-Hussieny, 2024; Hu et al., 2023; Chignoli et al., 2021]. This implies inference times of less than 10ms. At 1000Hz, a 15-second movement of the agent corresponds to a sequence of 15K steps [El-Hussieny, 2024] resulting in long context lengths even without ICL. While there exists a range of techniques to make large models faster, such as quantization [Frantar et al., 2023], distillation [Hinton et al., 2015], or pruning [LeCun et al., 1989], the quadratic-time complexity of self attention still remains.

Recently, *modern recurrent architectures* have been proposed, which exhibit similar parallelization properties during training as the Transformer architecture while offering linear-time inference complexity. These modern recurrent architectures include xLSTM [Beck et al., 2024] and state-space models (SSMs), such as Mamba [Gu & Dao, 2023; Dao & Gu, 2024] and Griffin/Hawk [De et al., 2024], and have challenged the dominance of the Transformer in language modeling but also in other domains such as computer vision [Alkin et al., 2024; Zhu et al., 2024], and biomedicine [Schmidinger et al., 2024]. More importantly, their linear-time inference makes them suitable for deployment in scenarios with limited compute, large context sizes, and real-time requirements, such as robotics.

In this work, we assess the aptitude of modern recurrent architectures, such as xLSTM and Mamba, as large action models. To this end, we introduce a Large Recurrent Action Model (LRAM) with an xLSTM at its core (see Figure 1). We train our agents on 432 tasks from 6 domains using a supervised learning setting similar to that of the Decision Transformer [Chen et al., 2021, DT]. We use data collected during online-RL training of single-task specialist agents and compile these trajectories alongside other expert demonstrations into a large-scale multi-domain dataset comprising 894M transitions. Due to their parallelization properties, the modern recurrent architectures considered in this work can process this large-scale training set as efficiently as the Transformer while being faster at inference. Experiments across 4 models sizes with our multi-task models indicate that xLSTM compares favorably to Transformers in terms of both performance and speed. In addition, we study the effect of modern recurrent architectures on fine-tuning performance and in-context learning abilities, and find that they exhibit strong performance in both dimensions.

The main purpose of this paper is to test the hypothesis that *modern recurrent model architectures are better suited for building LAMs than Transformers*. Hereby, we make the following **contributions**.

- We propose a Large Recurrent Action Model (LRAM) with an xLSTM at its core that enables efficient inference.
- We assess the aptitude of modern recurrent architectures as backbones for large-action models with respect to their efficiency at inference time and overall performance in multi-task, fine-tuning, and in-context learning settings.
- To foster further research on large action models, we release our data preparation pipeline and generated datasets.

## 2 RELATED WORK

**Sequence Models in RL.** LSTM [Hochreiter & Schmidhuber, 1997] is the dominant backbone architecture for partially observable online RL problems and has been behind achievements such as mastering Starcraft II [Vinyals et al., 2019], Dota 2 [Berner et al., 2019], and Atari [Espeholt et al., 2018; Kapturowski et al., 2019]. After the success of the Transformer in NLP [Devlin et al., 2019; Radford et al., 2019; Brown et al., 2020], computer vision [Dosovitskiy et al., 2021; He et al., 2022; Radford et al., 2021; Fürst et al., 2022] and speech recognition [Radford et al., 2022; Baevski et al., 2020], the architecture has found its way into RL. Chen et al. [2021] proposed the Decision Transformer (DT) a GPT-style model [Radford et al., 2018], that learns to predict actions from offline trajectories via behavior cloning. Trajectory Transformer [Janner et al., 2021] predicts actions along with states and rewards, which allows for dynamics modeling. Other follow-up works build on the DTs [Zheng et al., 2022; Wang et al., 2022; Shang et al., 2022; Meng et al., 2021; Siebenborn et al., 2022; Schmied et al., 2024a] or replace the Transformer with Mamba [Ota, 2024; Dai et al., 2024]. Furthermore, sequence models trained were found to exhibit ICL if conditioned on previous trajectories [Laskin et al., 2022; Lee et al., 2022; Kirsch et al., 2023], albeit in limited scenarios.

**Large Action Models (LAMs).** LAMs, such as the Decision Transformer, are well suited for multi-task settings. Lee et al. [2022] found that a multi-game DT can learn to play 46 Atari games. Reed et al. [2022] introduced a generalist agent trained on over 600 tasks from different domains, ranging from Atari to manipulation of a robot arm. Jiang et al. [2022] a Transformer for robot manipulation based on multi-modal prompts, that allow to steer the model to perform new tasks. Recently, Raad et al. [2024] introduced an agent instructable via language to play a variety of commercial video games. Since then, robotics has become an emergent area for developing LAMs [Brohan et al., 2023b;a; Octo Model Team et al., 2024; Gu et al., 2023; Wang et al., 2023; Kim et al., 2024], also due to the availability of large-scale robotics datasets [Jia et al., 2024; Embodiment Collaboration et al., 2024; Jiang et al., 2023; Mandlekar et al., 2023].

**Next-generation Sequence Modeling Architectures.** Linear recurrent models, such as state-space models (SSM, Gu et al., 2021; 2022b; Smith et al., 2023; Orvieto et al., 2023) have challenged the dominance of the Transformer [Vaswani et al., 2017] architecture on long-range tasks [Tay et al., 2020]. The key insight of those linear RNNs was to diagonalize the recurrent state matrix and enforce stable training via an exponential parameterization [Gu et al., 2022a; Orvieto et al., 2023]. Since then, there have been efforts to include features such as gating from RNNs [Elman, 1990; Jordan, 1990; Hochreiter & Schmidhuber, 1997; Cho et al., 2014]. Non-linear gates are believed to have higher expressivity, but are harder to train. Griffin [De et al., 2024] mixes gated linear recurrences with local attention to achieve more training data efficiency than Llama-2 [Touvron et al., 2023] and better sequence extrapolation. Mamba [Gu & Dao, 2023] introduces a selection mechanism similar to gating into SSMs, which makes its state and input matrix time dependent. This is similar to the gating mechanism of RNNs but also bears resemblance to approaches like fast weights [Schmidhuber, 1992] and Linear Attention [Katharopoulos et al., 2020]. Mamba-2 [Dao & Gu, 2024] highlight the connection between SSMs with input dependent state and input matrices and (Gated) Linear attention variants. Most recently, the xLSTM [Beck et al., 2024] was proposed as an improvement over the classic LSTM [Hochreiter & Schmidhuber, 1997] that combines gating, linear recurrences and recurrent weights into a single architecture for language modeling. First, xLSTM leverages exponential gating with stabilization to RNNs for stronger emphasis on important inputs. Second, xLSTM is composed of two variants, the mLSTM variant with an emphasis on memory that proves important in language modeling and the sLSTM variant that keeps the non-diagonalized recurrent matrix to enable state-tracking [Merrill et al., 2024]. State tracking is important in logic tasks and cannot be modeled fundamentally by linearized recurrent or state-space models like Mamba, Griffin or Transformers.

## 3 LARGE RECURRENT ACTION MODELS

### 3.1 BACKGROUND

**Reinforcement Learning.** We assume the standard RL formulation via a Markov Decision Process (MDP) represented by a tuple of $(\mathcal{S}, \mathcal{A}, \mathcal{P}, \mathcal{R})$, where $\mathcal{S}$ and $\mathcal{A}$ denote state and action spaces, respectively. At every timestep $t$ the agent observes state $s_t \in \mathcal{S}$, predicts action $a_t \in \mathcal{A}$, and receives

a scalar reward $r_t$. The reward is determined by the reward function $\mathcal{R}(r_t \mid s_t, a_t)$. $\mathcal{P}(s_{t+1} \mid s_t, a_t)$ defines the transition dynamics and constitutes a probability distribution over next states $s_{t+1}$ when executing action $a_t$ in state $s_t$. The goal of RL is to learn a policy $\pi(a_t \mid s_t)$ that predicts an action $a_t$ in state $s_t$ that maximizes $r_t$.

**Decision Transformer** [Chen et al., 2021] casts the RL problem setting as next action prediction task via causal sequence modeling. At training time, DT aims to learn a policy $\pi_\theta$ that maps future rewards to actions, which is often referred to as upside-down RL [Schmidhuber, 2019]. At inference time, the DT is conditioned via a target return to emit high-reward actions. Consequently, we assume access to a dataset $\mathcal{D} = \{\tau_i\}_{i=1}^N$ containing $N$ trajectories $\tau_i$ consisting of quadruplets $\tau_i = (s_1, \hat{R}_1, a_1, r_1, \ldots, s_T, \hat{R}_T, a_T, r_T)$ of state $s_t$, return-to-go (RTG) $\hat{R}_t = \sum_{t'=t}^T r_{t'}$, action $a_t$, and reward $r_t$. Here, $T$ refers to the length of the trajectory. The DT $\pi_\theta$ is trained to predict the ground-truth action $a_t$ conditioned on sub-trajectories from the dataset:

$$\hat{a}_t \sim \pi_\theta(\hat{a}_t \mid s_{t-C}, \hat{R}_{t-C}, a_{t-C}, r_{t-C}, \ldots, s_{t-1}, \hat{R}_{t-1}, a_{t-1}, r_{t-1}, s_t, \hat{R}_t), \tag{1}$$

where $C \leq T$ is the size of the context window. In fact, Equation 1 describes the setting of the multi-game DT [Lee et al., 2022], which also includes rewards in the sequence representation.

## 3.2 LARGE RECURRENT ACTION MODELS (LRAMs)

Our LRAM has a modern recurrent architecture at its core (see Figure 1), which comes with a parallel training and a recurrent inference mode. We instantiate LRAM with three different variants, two different xLSTM configurations and Mamba. Furthermore, we use a training protocol similar to that of Lee et al. [2022] and Reed et al. [2022] with some differences.

**Multi-modal sequence representation.** To encode input from different environments with varying state and action spaces, we use separate encoders per modality that are shared across tasks and domains. For encoding images we use a CNN similar to Espeholt et al. [2018], whereas for low-dimensional inputs we use a fully connected network. We refrain from patchifying images and tokenizing continuous states to avoid unnecessarily long sequences. Similarly, we use linear layers to encode rewards and RTGs. We omit actions in our sequence formulation, as we found that this can be detrimental to performance, in particular for continuous control tasks (see Section 4.3). Consequently, our trajectories have the form $\tau_i = (s_1, \hat{R}_1, r_1, \ldots, s_T, \hat{R}_T, r_T)$ and we train our policy $\pi_\rho$ to predict the ground-truth action $a_t$ as:

$$\hat{a}_t \sim \pi_\rho(\hat{a}_t \mid s_{t-C}, \hat{R}_{t-C}, r_{t-C}, \ldots, s_{t-1}, \hat{R}_{t-1}, r_{t-1}, s_t, \hat{R}_t). \tag{2}$$

**Shared action head.** Action spaces in RL typically vary across environments. For example, in the environments we consider, there are 18 discrete actions and a maximum of 8 continuous dimensions for continuous control environments. Therefore, we employ discretization of continuous action dimensions into 256 uniformly-spaced bins, similar to Reed et al. [2022] and Brohan et al. [2023b]. Unlike prior work, we leverage a shared action head to predict all discrete actions or continuous action dimensions at jointly. We found this setup significantly reduces inference time compared to using autoregressive action prediction of continuous actions.

**Recurrent inference mode.** At inference time, we leverage the recurrent backbone and maintain the hidden states of the last timestep. This enables fast inference with linear-time complexity along the sequence length. In addition, the recurrent-style inference is well suited for online fine-tuning via RL objectives, similar to LSTM-based policies in online RL. To further speed-up inference, we leverage custom kernels for the xLSTM backbone (see Appendix 22).

Our unified discrete action representation enables consistent training of our agents via the cross-entropy loss as training objective across all tasks and domains, similar to Reed et al. [2022]. We use separate reward scales per domain and target returns per task. Furthermore, we do not make use of timestep encodings as used by Chen et al. [2021], which are detrimental when episode lengths vary. We provide additional implementation details in Appendix B.

## 4 EXPERIMENTS

We study the aptitude of modern recurrent architectures as LAMs on 432 tasks from 6 domains: Atari [Bellemare et al., 2013], Composuite [Mendez et al., 2022], DMControl [Tassa et al., 2018],

**Table 1:** Dataset statistics for all 432 training tasks.

| Dataset | Tasks | Trajectories | Mean Trj. Length | Total Transitions | Repetitions |
|---|---|---|---|---|---|
| Atari | 41 | 136K | 2733 | 205M | 1.03× |
| Composuite | 240 | 480K | 500 | 240M | 0.87× |
| DMControl | 11 | 110K | 1000 | 110M | 1.92× |
| Meta-World | 45 | 450K | 200 | 90M | 2.34× |
| Mimicgen | 83 | 83K | 300 | 25M | 8.5× |
| Procgen | 12 | 2185K | 144 | 224M | 0.94× |
| **Total** | 432 | 3.4M | - | 894M | - |

Meta-World [Yu et al., 2020b], Mimicgen [Mandlekar et al., 2023], and Procgen [Cobbe et al., 2020b]. To this end, we compile a large-scale dataset containing 894 million transitions (see Section 4.1).

Across all experiments, we compare four backbone variants: xLSTM [7:1], xLSTM [1:0] [Beck et al., 2024], Mamba [Gu & Dao, 2023], and the GPT-2 style Transformer employed in the DT [Chen et al., 2021]. Following [Beck et al., 2024], we use the bracket notation for xLSTM, which indicates the ratio of mLSTM to sLSTM blocks. For example, xLSTM [1:0] contains only mLSTM blocks.

In Section 4.2, we conduct a scaling comparison for four model sizes ranging from 16M to 208M parameters that shows that modern recurrent architectures achieve performance comparable or favorable to the Transformer baseline across different model sizes. In Section 4.3, we study the impact of the recurrent backbones on fine-tuning performance and ICL abilities, and further analyze our trained recurrent backbones. Finally, in Section 4.4, we empirically examine the differences at inference time in terms of latency and throughput between xLSTM-based and Transformer-based agents, which indicate a clear advantage for the recurrent backbone.

## 4.1 DATASETS & ENVIRONMENTS

**Datasets.** We compile a large-scale dataset comprising 432 tasks from six domains. We leverage datasets from prior works. For Atari, we extract 5M transitions per task from the DQN-Replay dataset released by Agarwal et al. [2020]. For Composuite, we leverage the datasets released by [Hussing et al., 2023]. For Meta-World, we use 2M transitions per task released by [Schmied et al., 2024a]. For DMControl, we generate 10M transitions per task using task-specific RL agents. For Mimicgen, we use the datasets for the 21 tasks released by [Mandlekar et al., 2023] and generate trajectories for the remaining 62 tasks. Finally, for Procgen, we extract 20M transitions from the datasets released by [Schmied et al., 2024b]. Our final dataset contains 3.4M trajectories and in total 894M transitions (see Table 4.1). We reserve an additional 37 tasks from the same domains for zero-shot evaluation. To foster future research, we release our data-preparation pipeline and generated data at `Anonymized`.

**Environments.** Atari and Procgen come with image observations and discrete action. In contrast, the remaining four domains exhibit state-based observations and continuous actions. Consequently, our experiments involve a mixture of state and action spaces as well as varying episode lengths (see Table 4.1). Periodically evaluating the trained agents on all 432 tasks sequentially is time-consuming and we, therefore, distributed the evaluation across GPUs and parallel processes (see Appendix B).

Additional details on our datasets, environments are available in Appendix A.

## 4.2 SCALING COMPARISON

To conduct our main comparisons, we train our four backbone variants on the full training task mixture of 432 tasks. For each architecture backbone, we report performance scores for four model sizes: 16M, 48M, 108M, and 206M parameters. We train all models for 200K updates with a batch size of 128 and context length of 50 timesteps. All domains are represented with approximately equal proportion, resulting in 33K updates per domain. Additional implementation details and hyperparameters for every backbone variant and model size are available in Appendix B.

**Sequence prediction performance.** In Figure 2a, we report the validation set perplexity for all backbones and model sizes averaged over the individual scores from all domains. To achieve this,

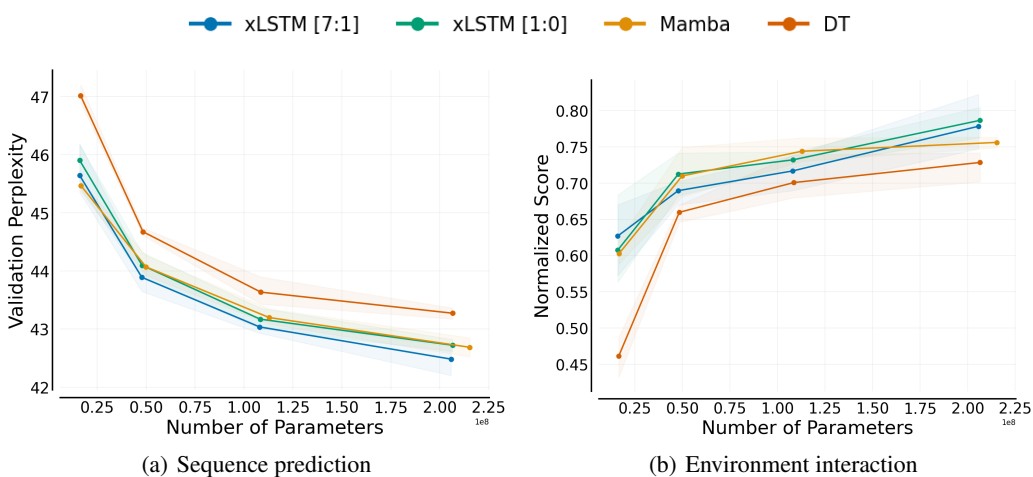

(a) Sequence prediction            (b) Environment interaction

**Figure 2:** Scaling comparison. We compare xLSTM, Mamba, DT in four model sizes: 16M, 48M, 110M, and 206M parameters. We show the **(a)** validation perplexity on the hold-out datasets, and **(b)** normalized scores obtained from evaluating in the training task environments, averaged over all 6 domains.

we maintain a hold-out set of trajectories for each training task (2.5%) and compute the perplexities after every 50K steps. Both recurrent backbones outperform the Transformer baseline considerably, especially as the model sizes increase. We provide the perplexities on the training set in Figure 13.

**Evaluation performance.** During training, we evaluate our agents after every 50K step in all 432 training environments. In Figure 2b, we report the resulting normalized performances averaged across all six domains. The recurrent backbones outperform the Transformer one across model sizes. While xLSTM and Mamba performs similarly at smaller scales, xLSTM tends to outperform Mamba at larger scales (206M). This is an important advantage of xLSTM, as LRAM agents can strongly benefit from more data and consequently larger models. Note, that Mamba has a significantly higher number of parameters than competitors.For the zero-shot evaluation performances on the 37 hold-out tasks, we refer to Figure 15 in Appendix C.2.

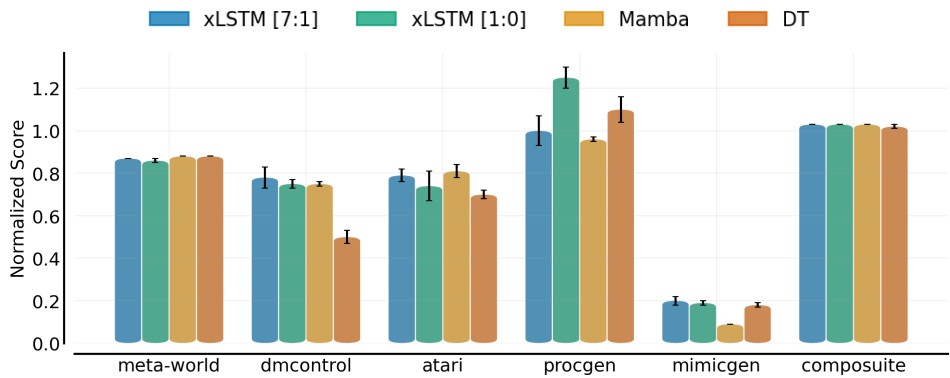

**Figure 3: Normalized scores per domain** for model size 206M. For Meta-World, DMControl, Mimicgen, Composuite and Procgen we report data-normalized scores, for Atari we report human-normalized scores.

**Performance per domain.** In Figure 3, we report the normalized scores for the 206M parameter models attained on all six domains. For Meta-World, DMControl, Mimicgen, Composuite, and Procgen we use data-normalized scores, as suggested by [Levine et al., 2020]. For Atari, we report human-normalized scores. Overall, we observe that the xLSTM backbone outperforms competitors on three of the six domains, while all methods perform similarly on the remaining 3 domains.

These experiments suggest that modern recurrent backbones can be attractive alternatives to the Transformer architecture for building LAMs.

### 4.3 ANALYSES & ABLATIONS

**Fine-tuning.** To assess the effect of the recurrent backbones on fine-tuning performance, we fine-tune our models on 37 held-out environments from all 6 domains. We evaluate the fine-tuning performance of the xLSTM architecture for both the 16M parameter pretrained models and compared it against an xLSTM trained from scratch. The pretrained LRAM outperforms the randomly initialized xLSTM model in most domains. For detailed results, see Appendix C.3. This suggests that fine-tuning performance is not affected negatively by switching the backbone.

**In-context Learning.** Next, we study the ICL abilities of our recurrent backbones on the Dark-Room environment considered in prior work on in-context RL [Laskin et al., 2022; Lee et al., 2023; Schmied et al., 2024b]. To study ICL in isolation, we train models from scratch with a multi-episodic context, which results in a large context length (we refer to Appendix C.4 for details on the experiment setup). In particular, we adopt the Algorithm Distillation (AD, Laskin et al., 2022) framework and exchange the Transformer backbone architecture with modern recurrent architectures. In Figure 17, we report the ICL performance on (a) 80 train and (b) 20 hold-out tasks. We find that xLSTM [7:1] attains the highest overall scores both on training and hold-out tasks, which we attribute to the state-tracking abilities [Merrill et al., 2024] of sLSTM blocks.

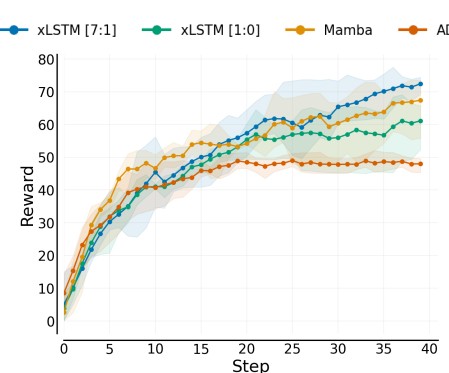

**Figure 4:** ICL with modern recurrent architectures on Dark-Room $10 \times 10$.

**Embedding space analysis.** In Figure 5, we analyze the representations learned by our model. To this end, we sample 32 sub-trajectories from every task, extract the sequence representation at the last layer, cluster them using UMAP [McInnes et al., 2018], and color every point by its domain. Appendix E describes the setup in greater detail. We find that tasks from the same domain cluster together. Furthermore, xLSTM exhibits a more refined domain separation compared to DT, which may contribute to the better down-stream performance.

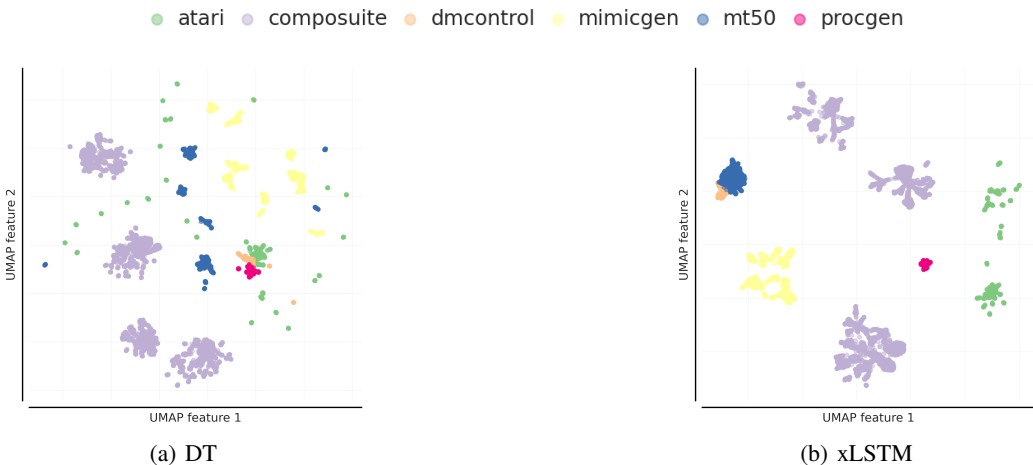

(a) DT

(b) xLSTM

**Figure 5:** Embedding space comparison. UMAP clustering of hidden states for all tasks for 16M models, colored by domain. xLSTM exhibits a better domain separation than DT.

**Removing Actions & Effect of Context Length.** We found that removing actions from the context results in better performance across backbones. While context lengths beyond 1 hurt performance

on Meta-World and DMControl and when training with actions, the reverse is true when training without actions (see Figures 23, 24, 26). This is in contrast to recent works, which did not benefit from longer contexts [Octo Model Team et al., 2024]. While removing actions improves performance on Meta-World and DMControl, it does not affect performance on discrete control environments. For Meta-World and DMControl, we observed that the models become overly confident (high action logits), which is problematic if poor initial actions are produced. We assume this is because many robotics environments exhibit smoothly changing actions and by observing previous actions the agent learns shortcuts. A similar issue has been observed by Wen et al. [2020] and termed the *copycat problem*. Removing actions from the input prevents the agent from using shortcuts and alleviates the copycat problem. Importantly, the evaluation performance improves across domains as the sequence length increases, which indicates that the history helps to predict the next action (e.g., by observing mistakes made in the recent past, see Figures 25, 27).

**Return-conditioning vs. Behavior Cloning.** Across our experiments, we utilized a sequence representation that includes return-to-go tokens as commonly used in DTs [Chen et al., 2021; Lee et al., 2022]. However, many recent works focus on behavior cloning without return conditioning [Reed et al., 2022; Brohan et al., 2023a; Octo Model Team et al., 2024]. Therefore, we study the effect of excluding the RTG tokens from the sequence representation at the 206M parameter scale, to validate that our findings transfer to the behavior cloning setting. Indeed, we find that the same trends hold (see Figure 28 in Appendix D.2).

**mLSTM-to-sLSTM Ratio.** Throughout our experiments, we compare two xLSTM variants: xLSTM [7:1] and xLSTM [1:0]. These ratios were proposed by Beck et al. [2024] and we maintain the same ratios for consistency (see Appendix B.3). While mLSTM is fully parallelizable, sLSTM enables state-tracking [Merrill et al., 2024]. To better understand the effect of this ratio, we conduct ablation studies both on the full 432 tasks and on Dark-Room (see Appendix D.3), similar to Beck et al. [2024]. We find that other ratios, such as [3:1], can be effective (see Figure 30). In addition, we find it important to place sLSTM blocks a lower-level layers. However, the effectiveness of sLSTM layers is dependent on the task at hand. We believe that complex tasks with long horizons or partial observability, as are common in real-world applications, may benefit from the state-tracking abilities provided by sLSTM blocks.

We present additional ablations on the effect of reducing the number of layers in xLSTM and disabling Dropout on DT in Appendix D.5 and D.4, respectively.

## 4.4 INFERENCE TIME COMPARISON

Finally, we empirically examine the difference between xLSTM-based and Transformer-based agents at inference time. Similar to De et al. [2024], we report both latency and throughput. We focus our analysis on latency, as it is the more important dimension for real-time applications.

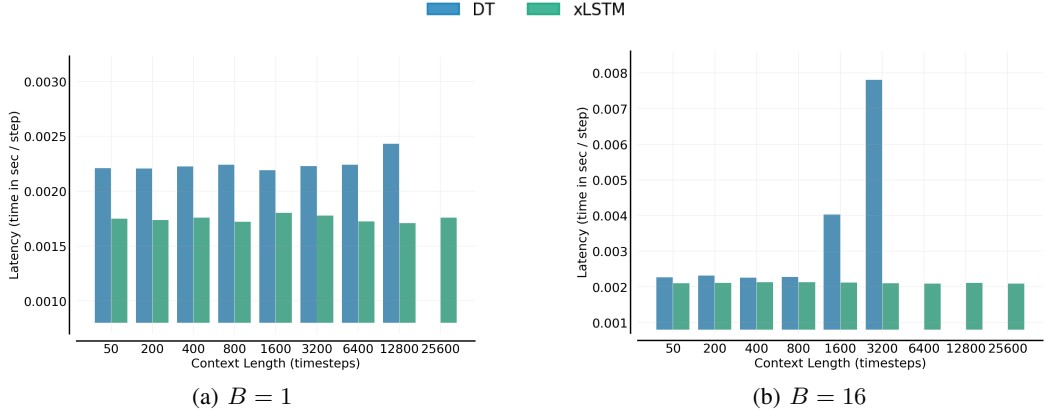

(a) $B = 1$

(b) $B = 16$

**Figure 6: Latency** comparison on A100. We report latency for varying context lengths (in timesteps) with fixed batch sizes $B$ of 1 and 16. We compare DT to xLSTM with the same number of layer blocks and parameters on Atari `Freeway`. Missing bars for DT indicate out-of-memory (OOM).

**Setup.** We conduct all inference time tests on A100 GPUs with 40GB of RAM using 206M parameter models. For the Transformer, we use KV-caching and FlashAttention [Dao, 2023] as supported by PyTorch [Paszke et al., 2019]. For xLSTM, we use recurrent-style inference using custom kernels to accelerate the computations (see Figure 22 for the impact of kernel acceleration). For both backbones, we use `torch.compile`. The Transformer with KV-caching has a linear time complexity per step and quadratic in the sequence length. In contrast, the xLSTM has a constant time complexity per step and linear in the sequence length. Therefore, we expect speed-ups especially for longer sequences and larger batch sizes, as observed by De et al. [2024]. To ensure a fair comparison, we compare DT and xLSTM with the same number of layer blocks and increase the hidden size of xLSTM to match the number of parameters of DT (see Appendix D.5 for evaluation performance of these models). We provide further details on our inference time tests in Appendix C.5.

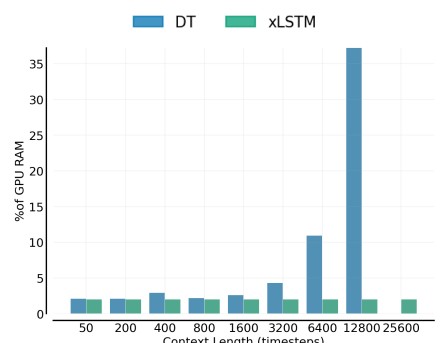

**Figure 7:** Memory consumption during **Latency** comparison on A100 (% of GPU memory) for varying context lengths and $B = 1$.

**Environment.** We conduct all inference time tests on the environment that exhibited the longest average episode lengths in our experiments, the Atari game `Freeway`. Every episode in `Freeway` lasts for 8192 steps, which is equivalent to 24576 tokens (s/rtg/r). We evaluate all models for 5 episodes and preserve the KV-cache/hidden state across episode boundaries. The reported latencies and throughputs are averaged across all evaluation episodes, except for the first episode, which we discard to exclude compilation times and prefilling. We opted for measuring the inference times during environment interaction, i.e., including simulator latency, rather than mere token generation.

**Latency.** Similar to De et al. [2024], we measure latency by the average time (in seconds) taken to perform a single inference step with a fixed batch size $B$ (lower is better). In Figure, 6, we report the latencies for varying context lengths, $C \in [50, 25600]$ and two batch sizes $B \in \{1, 16\}$. Note that $C$ is in time steps and every time step contains 3 tokens (state, reward-to-go, reward). Hence, the effective sequence length for the largest $C$ is 76800. As expected, we find that the recurrent backbone attains lower inference latencies than the Transformer one. As the sequence length increases, DT runs out of memory due to the increasing size of the KV cache (see Figure 7). In contrast, the inference speeds for xLSTM are independent of the context length, and therefore enable significantly longer context lengths. This property is particularly interesting for in-context RL, which requires keeping multiple episodes in the context [Laskin et al., 2022]. Nevertheless, our experiments highlight that the materialization of the complexity advantage (quadratic vs. linear) depends on the device, model size, batch size and the context length, which is similar to findings by De et al. [2024].

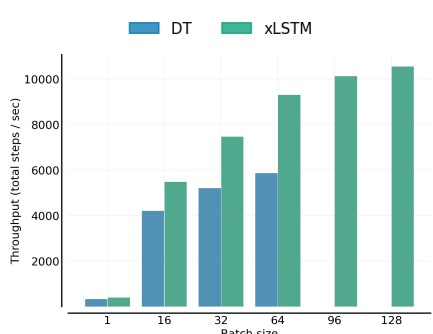

**Figure 8: Throughput** comparison on A100 for varying batch sizes with $C = 1600$ timesteps on the Atari `Freeway` environment. Missing bars for DT indicate OOM.

**Throughput.** Throughput is measured by the total amount of inference steps performed per second for a model with a fixed context length. In Figure, 8, we report the throughputs for varying batch sizes, $B \in [1, 128]$ for a fixed context length of $C = 1600$. Here, the batch size can be interpreted as the number of parallel environments the agent interacts with. As expected, we find that xLSTM attains considerably higher throughputs than the DT. The benefit of xLSTM increases with larger batch sizes. While the DT with quadratic complexity in the sequence length goes OOM for batch sizes above 64, the xLSTM with linear complexity can easily handle larger batch sizes. In both experiments, the recurrent xLSTM performs favorably over the Transformer backbone.

## 5 CONCLUSION

In this work, we study the aptitude of modern recurrent architectures as alternatives to Transformers for building LAMs. We found that our LRAM with an xLSTM or Mamba at its core compare favorably to the Transformer in terms of evaluation performance across different model scales (see Section 4.2). Moreover, we demonstrated that xLSTM-based LRAMs exhibit higher inference speeds, especially at large context sizes (see Section 4.4). Thus, the empirical evidence suggests, that recurrent backbones such as the xLSTM can be attractive alternatives for LAMs. Notably, the linear-time inference complexity of xLSTM may enable applications that require long context lengths, such as in-context RL, and facilitate the application of large-scale agents for real-time applications, such as robotics.

Nevertheless, modern recurrent architectures and Transformers come with different pros and cons. Both xLSTM and Mamba, on the one hand, exhibit a fundamental computational complexity advantage over Transformers. Their linear complexity ensures that the computational requirements increase slower with the sequence length. This property enables more efficient inference, which can be particularly relevant for edge-applications. While we conduct our inference time comparisons on a high-end data-center GPU, applications on edge-devices may have to deal with less powerful accelerators. Importantly, we found that LAMs strongly benefit from longer sequences (see Section 4.3). Transformers, on the other hand, are particularly effective for applications that require exact recall of tokens in a sequence, which can be important for decision-making [Ni et al., 2024]. Finally, xLSTM in particular enables state-tracking via sLSTM blocks, which Transformers and Mamba cannot perform [Merrill et al., 2024]. State tracking can be important for logic tasks and for dealing with partial observability in RL environments (see Section 4.3) and may be a useful tool for practicioners. Given these differences, different backbones should be considered depending on the task at hand.

**Limitations.** The primary target application of LAMs is robotics. While the majority of our experiments involve robotic simulations, we do not yet provide empirical evidence for real robots. We do, however, believe that our findings translate to real-world scenarios and aim to provide further evidence in future work. Moreover, the fine-tuning experiments in this work are limited to offline RL. We envision that an agent pre-trained by behavioral cloning on large-scale offline RL datasets may be successfully fine-tuned in an online RL setting to explore new strategies that do not appear in the training data. Modern recurrent architectures offer both parallel and recurrent training mode, which might be the key to success for such applications. While we provide initial evidence of improved ICL abilities of modern recurrent architectures, we only consider a limited grid-world setting. Consequently, we aim to further investigate the in-context RL abilities of recurrent backbones on more complex environments in future work.

## 6 ETHICS STATEMENT

While we conduct all our experiments in simulated environments, the primary target application of our method is robotics. We believe that our work can positively impact applications in the near future, which require efficient inference, on-device processing, or have real-time constraints. However, robotics applications in the real world are not without risks. In particular, in areas where humans are involved, such as factory settings, special care is required. LAMs are trained via next-action prediction similar to LLMs. Consequently, LAMs may also suffer from hallucinations in unknown scenarios. We therefore strongly discourage users from blindly following the predictions made by real-world LAMs without appropriate safeguards regarding safety and robustness. It is essential to ensure responsible deployment of such future technologies, and we believe that more research on the robustness of LAMs is necessary.

## 7 REPRODUCIBILITY

Upon publication, we will make the code-base used for our experiments publicly available, and release the datasets we generated. Both will be available at: `Anonymized`. As part of this submission, we also include the source code in the supplementary material. We describe the environments we use for our experiments and provide dataset statistics in Appendix A. Furthermore, in Appendix B, we provide implementation details for all methods and a list of hyperparameters used for our experiments.

In Appendix C, we present additional figures that accompany our results in the main text (e.g., all model sizes). Finally, in Appendices D and E, we provide further details on the conducted ablation studies and the embedding space analysis, respectively.

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

APPENDIX

# Contents

## A ENVIRONMENTS & DATASETS

### A.1 GENERAL

We compile a large-scale dataset comprising 432 tasks from six domains, 3.4M trajectories, and 894M transitions in total (see Table 4.1). To enable fast and targeted data-loading, every trajectory is stored in a separate `hdf5` file. We trade off some data-loading speed for disk space efficiency, by compressing trajectories that contain image-based observations.

### A.2 ATARI

The Arcade Learning Environment (ALE) [Bellemare et al., 2013] is the standard benchmark for evaluating RL agents and consists of 57 Atari games. Input observations in Atari are RGB images, but as is standard practice we gray-scale and crop frames ($|\mathcal{S}| = 1 \times 64 \times 64$). There are 18 discrete action across all 57 Atari games ($|\mathcal{A}| = 18$), but individual games may use only use a subset of these

actions. Furthermore, we adopt the standard Atari recipe as used in prior works, including a frame skip of 4, maximum number of no-ops of 30, resetting on life loss, and reward clipping to $[-1, 1]$ [Mnih et al., 2015; Hessel et al., 2017].

**Tasks.** Similar to Lee et al. [2022], we assign 41 games to the training set, and 5 additional tasks to the hold-out set. The 41 training tasks include:

```
amidar, assault, asterix, atlantis, bank-heist, battle-zone, beam-rider,
boxing, breakout, carnival, centipede, chopper-command, crazy-climber,
demon-attack, double-dunk, enduro, fishing-derby, freeway, frostbite,
gopher, gravitar, hero, ice-hockey, jamesbond, kangaroo, krull,
kung-fu-master, name-this-game, phoenix, pooyan, qbert, riverraid,
road-runner, robotank, seaquest, time-pilot, up-n-down, video-pinball,
wizard-of-wor, yars-revenge, zaxxon
```

The 5 hold-out tasks include: `alien`, `pong`, `ms-pacman`, `space-invaders`, `star-gunner`

**Dataset.** For Atari, we leverage the DQN-Replay dataset released by Agarwal et al. [2020]. The dataset contains the trajectories seen over the entire training of the DQN agent (50M frames), We extract a subset of the last 5M transitions for every task, amounting to 205M transitions in total for the 41 training tasks. The number of episodes, the episodes lengths and total achieved rewards vary across tasks, as shown in Table 2.

### A.3 META-WORLD

The Meta-World benchmark [Yu et al., 2020a] consists of 50 manipulations tasks using a Sawyer robotic arm, ranging from opening or closing windows, to pressing buttons. Meta-World is based on the MuJoCo physics engine [Todorov et al., 2012b]. Observations in Meta-World are 39-dimensional continuous vectors ($|\mathcal{S}| = 1 \times 64 \times 39$), and actions are represented by 6 continuous dimensions ($|\mathcal{A}| = 18$) in range $[-1, 1]$. All tasks share a common action and state space. Following Wolczyk et al. [2021] and Schmied et al. [2024a], we limit the episode lengths to 200 interactions.

**Tasks.** We follow Yu et al. [2020a] and split the 50 Meta-World tasks into 45 training tasks (MT45) and 5 evaluation tasks (MT5).

The 45 training tasks are:

```
reach, push, pick-place, door-open, drawer-open, drawer-close,
button-press-topdown, peg-insert-side, window-open, window-close,
door-close, reach-wall, pick-place-wall, push-wall, button-press,
button-press-topdown-wall, button-press-wall, peg-unplug-side,
disassemble, hammer, plate-slide, plate-slide-side, plate-slide-back,
plate-slide-back-side, handle-press, handle-pull, handle-press-side,
handle-pull-side, stick-push, stick-pull, basketball,soccer,
faucet-open, faucet-close, coffee-push, coffee-pull, coffee-button,
sweep, sweep-into, pick-out-of-hole, assembly, shelf-place, push-back,
lever-pull, dial-turn
```

The 5 evaluation tasks are: `bin-picking`, `box-close`, `door-lock`, `door-unlock`, `hand-insert`

**Dataset.** For Meta-World, we use the datasets released by [Schmied et al., 2024a], which contain 2M transitions per tasks and consequently 90M transitions in total for the training set. All episodes last for 200 environment interaction steps, and consequently there are 10K episodes for every task. For detailed dataset statistics per task, we refer to their publication.

### A.4 DMCONTROL

The DMControl benchmark [Tassa et al., 2018] consists of 30 different robotic tasks. Unlike Meta-World, the benchmark contains robots with different morphologies instead of a single common Sawyer arm. Due to the different robot morphologies, the state, and action spaces vary across tasks ($3 \le |\mathcal{S}| \le 24$, $1 \le |\mathcal{A}| \le 6$), with all actions in range $[-1, 1]$.

**Table 2:** Atari Dataset Statistics.

| Task | # of Trajectories | Mean Length | Mean Return |
|------|-------------------|-------------|-------------|
| amidar | 1813 | 2753 | 145 |
| pooyan | 2773 | 1800 | 176 |
| frostbite | 5218 | 766 | 18 |
| video-pinball | 1023 | 3902 | 266 |
| wizard-of-wor | 3059 | 1314 | 15 |
| chopper-command | 5452 | 738 | 18 |
| breakout | 3780 | 1300 | 39 |
| phoenix | 3307 | 1509 | 49 |
| asterix | 5250 | 951 | 55 |
| enduro | 571 | 8720 | 636 |
| kung-fu-master | 1775 | 2812 | 131 |
| hero | 3022 | 1345 | 168 |
| assault | 3782 | 1170 | 77 |
| demon-attack | 1649 | 2431 | 116 |
| qbert | 3939 | 1138 | 155 |
| jamesbond | 2841 | 1758 | 11 |
| bank-heist | 4146 | 1204 | 62 |
| up-n-down | 3246 | 1538 | 99 |
| centipede | 6879 | 582 | 81 |
| boxing | 4796 | 1041 | 63 |
| battle-zone | 1933 | 2134 | 15 |
| name-this-game | 988 | 5049 | 389 |
| zaxxon | 2561 | 1950 | 12 |
| beam-rider | 1232 | 3248 | 77 |
| time-pilot | 3886 | 1029 | 11 |
| ice-hockey | 1465 | 3407 | -6 |
| riverraid | 2645 | 1512 | 143 |
| krull | 3032 | 1319 | 528 |
| gopher | 1817 | 2338 | 185 |
| freeway | 2438 | 2048 | 33 |
| seaquest | 2807 | 1779 | 150 |
| double-dunk | 1774 | 2815 | 0 |
| road-runner | 3308 | 1217 | 135 |
| atlantis | 186 | 26349 | 1394 |
| gravitar | 6187 | 646 | 1 |
| yars-revenge | 4094 | 1036 | 96 |
| crazy-climber | 1105 | 3954 | 572 |
| kangaroo | 1787 | 2792 | 50 |
| fishing-derby | 2737 | 1825 | 0 |
| carnival | 21131 | 194 | 37 |
| robotank | 747 | 6652 | 56 |
| **Average** | 3321 | 2734 | 153 |

**Tasks.** We do not use all 30 tasks contained in the DMControl benchmark, but select 16 of the 30 tasks that have been used in prior works [Hafner et al., 2019; Schmied et al., 2024a;b], which we refer to as DMC11 and DMC5 respectively.

The 11 training tasks are:

```
finger-turn_easy,   fish-upright,   hopper-stand,   point_mass-easy,
walker-stand,   walker-run,   ball_in_cup-catch,   cartpole-swingup,
cheetah-run, finger-spin, reacher-easy
```

The 5 evaluation tasks are:

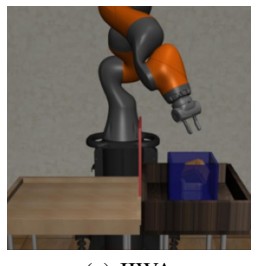 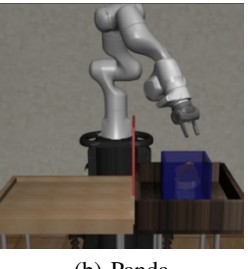 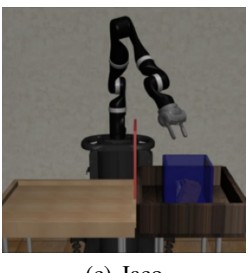 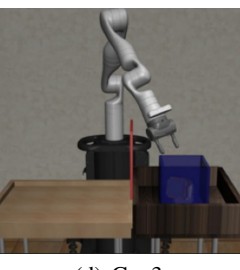

| (a) IIWA | (b) Panda | (c) Jaco | (d) Gen3 |

**Figure 9:** Illustration of the four supported robot arms in **Composuite** [Mendez et al., 2022].

```
cartpole-balance, finger-turn_hard, pendulum-swingup, reacher-hard,
walker-walk
```

**Dataset.** For DMControl, we generate 10M transitions per task by training task-specific SAC [Haarnoja et al., 2018] agents, using the same setup as Schmied et al. [2024a]. Episodes in all DMControl tasks last for 1000 environment steps and per time-step a maximum reward of +1 can be achieved, which results in a maximum reward of 1000 per episode. Consequently, our training set contains 10K episodes per tasks, amounting to 110K episodes and 110M transitions in total across all tasks. We list the dataset statistics for all 11 tasks in Table 3.

**Table 3:** DMControl Data statistics.

| Task | # of Trajectories | Mean Length | Mean Return |
|------|------------------|-------------|-------------|
| point_mass_easy | 10K | 1K | 851 |
| cheetah_run | 10K | 1K | 385 |
| walker_run | 10K | 1K | 230 |
| ball_in_cup_catch | 10K | 1K | 969 |
| hopper_stand | 10K | 1K | 460 |
| walker_stand | 10K | 1K | 939 |
| finger_turn_easy | 10K | 1K | 954 |
| reacher_easy | 10K | 1K | 938 |
| cartpole_swingup | 10K | 1K | 817 |
| fish_upright | 10K | 1K | 815 |
| finger_spin | 10K | 1K | 966 |
| **Average** | 19628 | 152 | 8.2 |

### A.5 COMPOSUITE

The Composuite benchmark [Mendez et al., 2022], is a robotics benchmark for grasping and object manipulation. The benchmark is implemented on top of `robotsuite` [Zhu et al., 2020], which in turn leverages the MuJoCo simulator under the hood [Todorov et al., 2012a]. Composuite contains a mix of 4 simulated robot arms: `IIWA`, `Jaco`, `Gen3`, and `Panda` (see Figure 9). All arms share a common state and action space containing 93 continuous state dimensions and 8 continuous action dimensions, respectively ($|\mathcal{S}| = 93, |\mathcal{A}| = 8$).

**Tasks.** CompoSuite is designed as a compositional multi-task benchmark for RL, in which a particular *robot* manipulates a particular *object* given an *objective*, while avoiding *obstacles*. Overall, there are 4 robots arms, 4 objects, 4 obstacles, and 4 task objectives. This results in 256 possible robot/object/objective/obstacles combinations. For our experiments, we assign 240 tasks to the training set and use the remaining 16 tasks as hold-out set (`Panda` and `Object_Wall`) combinations. For a list of all 256 tasks, we refer to Mendez et al. [2022].

**Dataset.** For Composuite, we leverage the datasets released by Hussing et al. [2023]. For every task, we select 2000 episodes, which last on average for 500 steps. This amounts to 1M transitions per

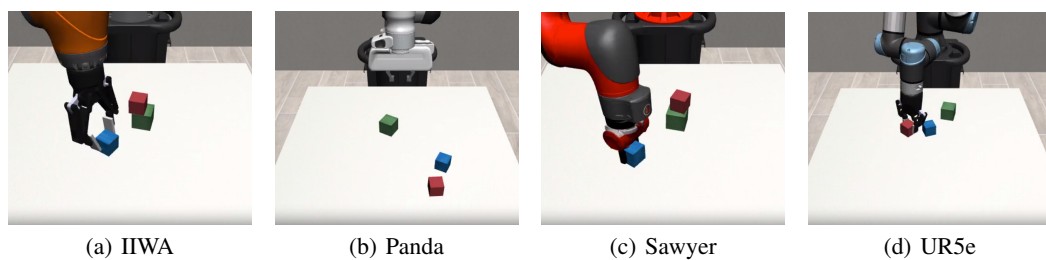

|(a) IIWA|(b) Panda|(c) Sawyer|(d) UR5e|

**Figure 10:** Illustration of the four supported robot arms in **Mimicgen** [Mandlekar et al., 2023] solving the `stack-three` task.

task, and 240M transitions across all 240 training tasks. For dataset statistics, we refer to Hussing et al. [2023].

### A.6 MIMICGEN

Similar to Composuite, Mimicgen [Mandlekar et al., 2023] is based on `robosuite` and the MuJoCo simulator. Mimicgen is designed for automatically synthesizing large-scale datasets from only a handful of human demonstrations. Observations in Mimicgen can be represented as images (from multiple cameras) or low dimensional continuous states. For our experiments, we opt for the low-dimensional state representation to simplify learning. Therefore, observations and actions are represented by 37-dimensional and 7-dimensional continuous vectors, respectively ($|\mathcal{S}| = 37$, $|\mathcal{A}| = 7$). Similar to Composuite, Mimicgen supports 4 different robot arms: `Panda`, `IIWA`, `Sawyer`, and `UR5e` (see Figure 10).

**Tasks.** Mimicgen consists of 24 diverse tasks, including stacking blocks, re-assembling objects, and even long-horizon tasks like coffee preparation. These 24 tasks can be performed with the four supported robot arms, amounting to 96 tasks in total.

**Dataset.** Mandlekar et al. [2023] released dataset for the 24 tasks using the default robot arm `Panda`. To increase the dataset diversity, we additionally generated data for the remaining 3 robot arms. However, not all data generation runs produce successful trajectories, and we discard with too few successful trajectories. Our final dataset for Mimicgen contains 83 training and 2 evaluation tasks. For each task, we collect 1000 successful demonstrations (we do not include unsuccessful trajectories). Episode lengths vary across tasks, ranging from 260 to 850 environment steps.

### A.7 PROCGEN

Procgen benchmark consists of 16 procedurally-generated video games [Cobbe et al., 2020a]. Observations in Procgen are RGB images of dimension $3 \times 64 \times 64$. However, for training efficiency, we apply gray-scaling to image observations ($|\mathcal{S}| = 1 \times 64 \times 64$). All 16 environments share a common action space of 15 discrete actions ($|\mathcal{A}| = 16$). Procgen is designed to test the generalization abilities of RL agents. Consequently, procedural generation is employed to randomize background and colors, while retaining the game dynamics.

**Tasks.** Following prior works [Raparthy et al., 2023; Schmied et al., 2024b], we assign 12 and 4 tasks to training and hold-out set, respectively. The 12 training tasks are:

```
bigfish, bossfight, caveflyer, chaser, coinrun, dodgeball,
fruitbot, heist, leaper, maze, miner, starpilot
```

The 4 hold-out tasks are: `climber`, `ninja`, `plunder`, `jumper`

**Dataset.** We leverage the datasets released by Schmied et al. [2024b], which contain 20M transitions per task. The datasets were generated by recording all transitions observed by training RL agents for 25M steps, followed by uniform subsampling to 20M transitions. Consequently, the dataset contains mixed quality trajectories ranging from random (beginning of training) to expert (end of training). We list the dataset statistics for all 16 tasks in Table 4.

**Table 4:** Procgen Data statistics.

| Task | # of Trajectories | Mean Length | Mean Return |
|------|------|------|------|
| bigfish | 82835 | 230 | 6.251 |
| bossfight | 112459 | 141 | 1.946 |
| caveflyer | 151694 | 105 | 7.745 |
| chaser | 93612 | 212 | 3.248 |
| coinrun | 261117 | 51 | 9.473 |
| dodgeball | 144364 | 137 | 2.884 |
| fruitbot | 73653 | 270 | 16.094 |
| heist | 101361 | 196 | 8.405 |
| leaper | 296084 | 67 | 4.446 |
| maze | 482245 | 41 | 9.432 |
| miner | 288818 | 68 | 11.8 |
| starpilot | 96468 | 206 | 17.3 |
| **Average** | 182059 | 144 | 8.3 |

**Table 5:** Hyperparameters for RA-DT.

| Parameter | Value |
|------|------|
| Gradient steps | 200K |
| Evaluation frequency | 50K |
| Evaluation episodes | 5 |
| Optimizer | AdamW |
| Batch size | 128 |
| Gradient accumulation | 6 |
| Lr schedule | Linear warm-up + Cosine |
| Warm-up steps | 4000 |
| Learning rate | 1e-4 $\rightarrow$ 1e-6 |
| Weight decay | 0.01 |
| Gradient clipping | 0.25 |
| Dropout | 0.2 |
| Context len (timesteps) | 50 |
| Reward scale | per-domain |
| Target return | per-task |

## B    EXPERIMENTAL & IMPLEMENTATION DETAILS

### B.1    TRAINING & EVALUATION.

In our experiments, we compare two variants of xLSTM, Mamba and DT. For our main experiments in Section 4.2, we train all models for 200K updates, and evaluate after every 50K update steps. We report the mean and 95% confidence intervals over three seeds in our experiments, as suggested by Agarwal et al. [2021]. For every evaluation tasks, we take the average of 3 evaluation seeds.

We train our agents with a batch size of 128 and gradient accumulation across the 6 domains, such that every domain is represented with the same proportion. Consequently, the effective batch size is 768. We use a learning rate of $1e^{-4}$, 4000 linear warm-up steps followed by a cosine decay to $1e^{-6}$, and train using the AdamW optimizer [Loshchilov & Hutter, 2018]. In addition, we employ gradient clipping of 0.25, weight decay of 0.01 for all models. We do not employ Dropout, as is standard practice in DTs, as we found that it negatively affects performance (see Section 4.3). We use separate reward scales of 200, 100 and 20 for Meta-World, DMControl and Atari, respectively. Furthermore, for all domains, we set the target return to the maximum return achieved for a particular task in the training datasets. This is particularly useful for domains, where the maximum returns differ heavily across tasks (e.g., Atari). We list all hyperparameters in Table 5.

### B.2 CONTEXT LENGTHS.

By default, we train all models with a context length $C = 50$ timesteps. For every timestep there are three tokens (s/rt/r) and consequently, the effective context length is 150. We found that performance improves for longer context lengths (see Section D.1), but limit our experiments to $C = 50$ to reduce the computational cost.

### B.3 MODEL ARCHITECTURES.

We train models across 4 models sizes: 16M, 48M, 110M, and 206M. We follow Lee et al. [2022] in selecting the number of layers and hidden dimensions. For xLSTM and Mamba, we use twice the number of layers blocks to match the number of parameters of the Transformer [Beck et al., 2024; Gu et al., 2024] (see Table 6) For our xLSTM [7:1] variant, which contains sLSTM blocks, we strive to maintain the same ratio as proposed by Beck et al. [2024]. Not all our model sizes are divisible by 8 and only the 16M and 110M models exhibit the exact 7:1 ratio of mLSTM to sLSTM blocks. For consistency, however, we maintain the same notation as Beck et al. [2024]. We place sLSTM blocks at positions [1], [1, 3], [1, 3], and [1, 3, 5] for the 16M, 48M, 110M, 206M, respectively.

Across backbones, we use linear layers to encode continuous states, reward returns-to-go, similar to Chen et al. [2021]. The maximal state-dimension across continuous control environments is 204 in our experiments. To use a shared linear embedding layer for continuous states, we pad states that have lower number of dimensions to 204 dimensions using zeros. To encode image inputs on visual domains, we use the IMPALA-CNN proposed by Espeholt et al. [2018] and adopted by previous works on Procgen [Cobbe et al., 2020a] and Atari [Schmidt & Schmied, 2021; Schwarzer et al., 2023]. Consequently, we do not make use of discretization of continuous states or patchification of images. This design choice significantly reduces the sequence length to only three tokens per time-step (see Appendix B.2) and consequently results in faster inference.

For continuous actions, we make use of discretization and discretize of every action dimension into 256 uniformly-spaced bins, similar to Reed et al. [2022] and Brohan et al. [2023b]. We experimented with lower/higher number of bins, but did not observe a benefit beyond 256 bins. Consequently, this resolution is sufficient for the environments we consider. We use a shared action head to predict the action bins of all continuous dimensions jointly. The maximum number of continuous action dimensions is 8 in our experiments and consequently the number of discrete action classes is 2048. In addition, there are 18 discrete actions originating from Atari and Procgen. Therefore, our action head learns to predict the correct action among the 2066 discrete classes. While different environments may have different action dimensions, the model predicts all action dimensions jointly. At inference time, the number of action dimensions of the current environment is known, and we extract the respective dimensions from the joint predictions. We opt for the shared action head representation, as this further speeds up inference and does not require autoregressive action prediction.

For the Transformer baseline, we use global positional embeddings similar to Chen et al. [2021]. For the recurrent backbones, we do not make use of positional encodings.

### B.4 HARDWARE & TRAINING TIMES.

We train all our models on a server equipped with 4 A100 GPUs. We use distributed data parallel to distribute the workload, as supported in PyTorch [Paszke et al., 2019]. Training times range from 5 hours for the smallest DT model to 30 hours for the largest Mamba model. Throughout all our experiments, we use mixed precision training [Micikevicius et al., 2017] as supported in PyTorch to speed up training time.

We evaluate our models after every 50K steps. However, periodically evaluating the trained agents on all 432 tasks sequentially is time-consuming. Therefore, we perform parallel evaluation with 4 processes at a time. For multi-GPU setups, we distribute the evaluation workload among the available GPUs. For example, with 4 available GPUs and 4 evaluation processes per GPU, 16 environments are evaluated simultaneously. Consequently, the total evaluation time for all 432 tasks, ranges from 18 minutes for the smallest DT model to roughly 2 hours for the largest Mamba model.

**Table 6:** Model Sizes.

| Model | Layers | Hidden Dim | Heads | Parameters |
|---|---|---|---|---|
| Transformer | 4 | 512 | 8 | 16M |
| Transformer | 6 | 768 | 12 | 48M |
| Transformer | 8 | 1024 | 16 | 110M |
| Transformer | 10 | 1280 | 20 | 206M |
| Mamba | 8 | 512 | - | 16M |
| Mamba | 12 | 768 | - | 48M |
| Mamba | 16 | 1024 | - | 110M |
| Mamba | 20 | 1280 | - | 206M |
| xLSTM | 8 | 512 | 4 | 16M |
| xLSTM | 12 | 768 | 4 | 48M |
| xLSTM | 16 | 1024 | 4 | 110M |
| xLSTM | 20 | 1280 | 4 | 206M |

## C ADDITIONAL RESULTS

### C.1 TRAINING TASKS

In Figures 11 and 12, we report the normalized scores obtained per domain and the average learning curves across tasks for all four model sizes.

In Figure 13, we report the training perplexity on the 432 training tasks over 200K updates. Here, we observe that the training perplexity behaves similar to the validation perplexity. This is expected, as our models see most transitions only a single time (see Table 4.1 for the number of repetitions per domain).

Furthermore, we report the scaling curves with an additional model size of 408M parameters in Figure 14. Due to the high computational cost of the 408M models, we were currently only able to conduct a single run for this size. However, we aim to provide further empirical evidence for this model sizes in future work.

### C.2 HOLD-OUT TASKS

In Figure 15, we show the zero-shot evaluation performance on the hold-out tasks 15. We want to highlight, that the performance declines for all methods and model sizes compared to performance on training tasks. This is because, hold-out tasks exhibit severe shifts in state-spaces, action-spaces and reward functions.

### C.3 FINE-TUNING

In Figure 16, we present the fine-tuning evaluation performance on the held-out tasks. We compare xLSTMs trained from scratched against xLSTMs initialized with the pre-trained weights. We do observe consistent improvement of the pre-trained models over the models trained from scratch. However, while we train on a substantial number of environments, the total amount of data used is still only a fraction of that employed in training other large-scale models, such as LLMs. Consequently, we do not observe comparable few-shot generalization. WHowever, we anticipate that few-shot generalization capabilities will emerge as we increase both data volume and model size.

### C.4 IN-CONTEXT LEARNING

We assess the ICL abilities of modern recurrent architectures on the Dark-Room environment considered in prior works on in-context RL [Laskin et al., 2022; Lee et al., 2023; Schmied et al., 2024b]. In Dark-Room, the agent is located in a dark room. The task is to navigate to an invisible goal location in that dark room. The state is partially observable, as the agent only observes its own x-y position on the grid ($|\mathcal{S}| = 2$). The action space consists of 5 discrete actions: move up, move

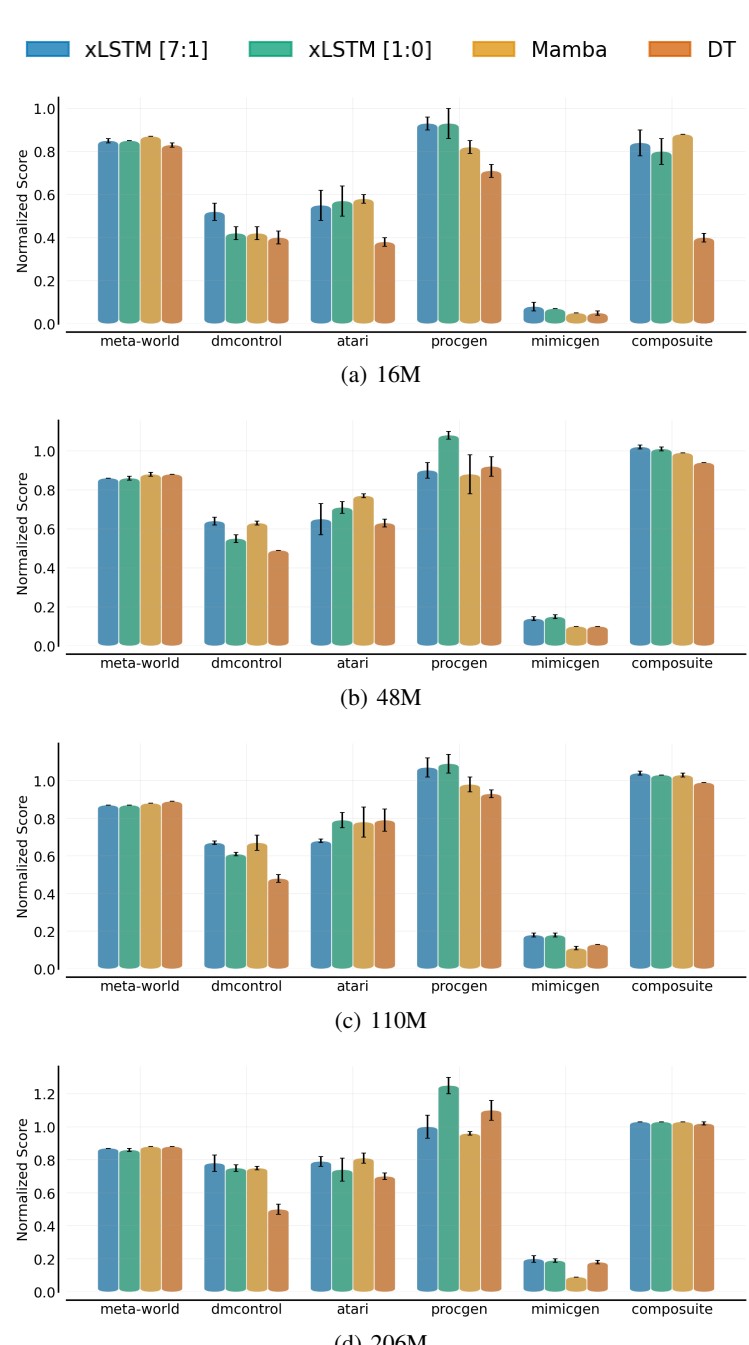

**Figure 11:** Normalized scores per-domain all four model sizes: 16M, 48M, 110M, and 206M. For Meta-World, DMControl, Mimicgen, Composuite, and Procgen we report data-normalized scores, for Atari we report human-normalized scores.

down, move left, move right, stay ($|\mathcal{A}| = 5$). Upon reaching the goal location, the agent receives a reward of +1 for every step in the episode it resides on the goal location. Consequently, the agent first has to explore the room to find the goal. Once the goal location is found (as indicated by the positive reward), the agent can exploit this knowledge. Given a multi-episodic context, the agent should be able to exploit information contains in the previous trials (e.g., exploiting one path vs. avoiding another).

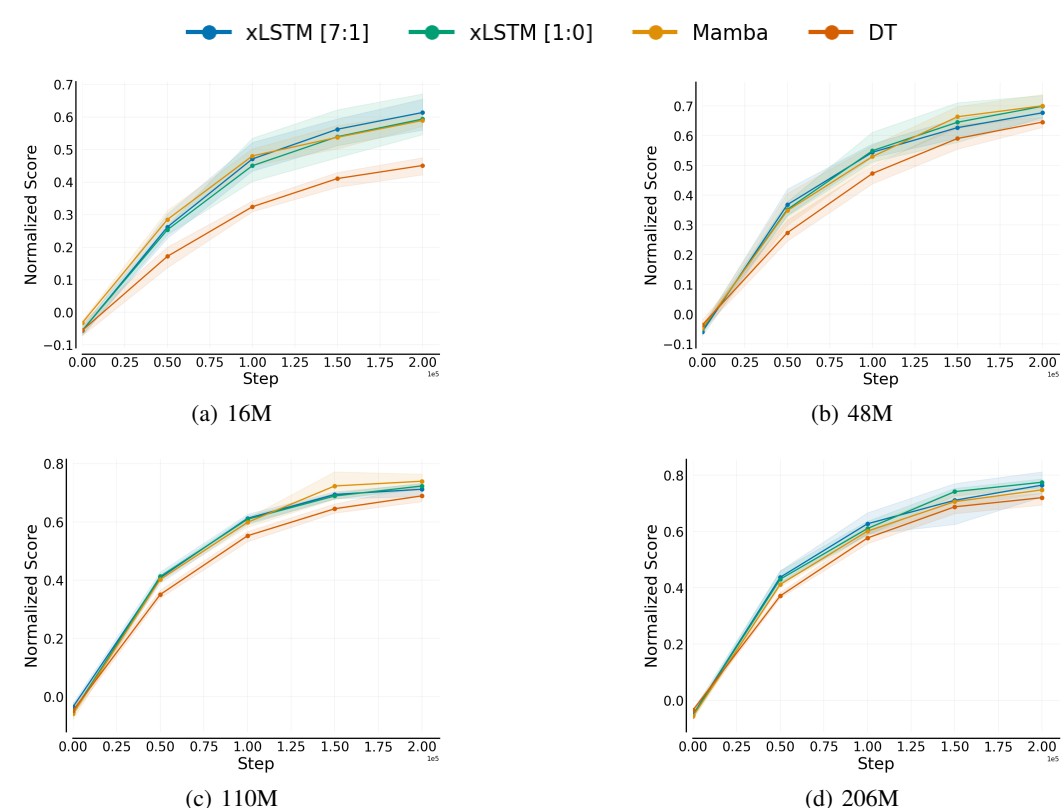

**Figure 12:** Learning curves for all four model sizes, 16M, 48M, 110M, and 206M, on the training tasks.

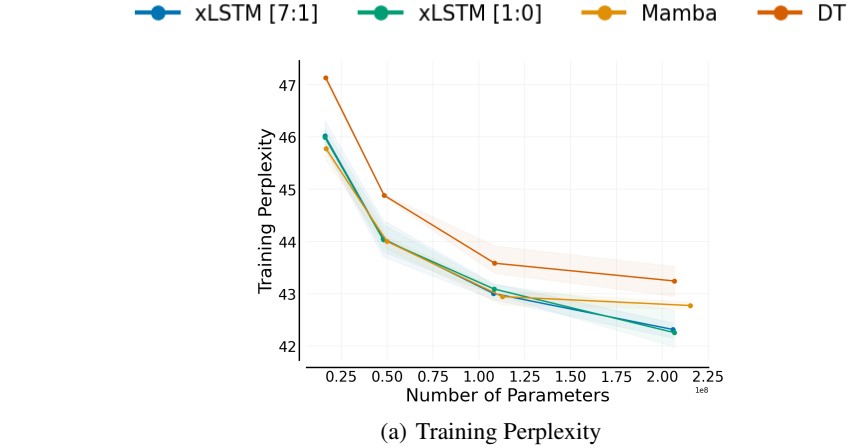

(a) Training Perplexity

**Figure 13:** Scaling comparison. We compare xLSTM, Mamba, DT in four model sizes: 16M, 48M, 110M, and 206M parameters. We show the **training perplexity** on the training dataset to evaluate the sequence prediction performance.

In our experiments, the Dark-Room is a $10 \times 10$ grid and episodes last for 100 steps, starting in the top left corner of the grid. We adopt the same experiment setup as Schmied et al. [2024b] and leverage their datasets. We train 16M parameter agents on datasets from 80 randomly selected goal locations in the grid. The datasets contain 100K transitions per task and are obtained by training task-specific PPO [Schulman et al., 2018] agents. Then, we evaluate the in-context abilities of our

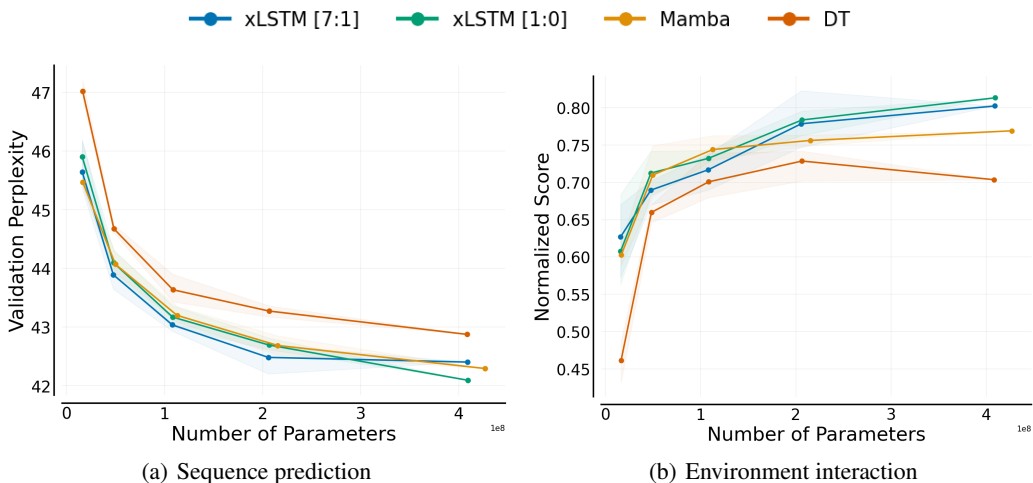

(a) Sequence prediction        (b) Environment interaction

**Figure 14:** Scaling comparison with additional 408M parameter models. We show the **(a)** validation perplexity on the hold-out datasets, and **(b)** normalized scores obtained from evaluating in the training task environments, averaged over all 6 domains.

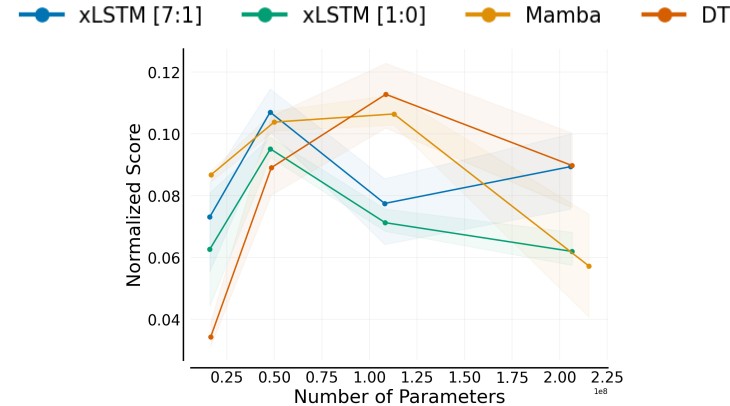

**Figure 15:** Scaling comparison. **Zero-shot performance on hold-out tasks** at four models sizes, 16M, 48M, 110M, and 206M. Note that performance declines for all methods and model sizes compared to performance on training tasks. This is because, hold-out tasks exhibit severe shifts in state-spaces, action-spaces and reward functions.

agents on 20 hold-out goal locations. During evaluation, the agent is given 40 episodes to interact with the environment, which we refer to as ICL-trials. Furthermore, we adopt the AD [Laskin et al., 2022] framework for training our agents with a multi-episodic context. We use the same sequence representation as used in our main experiments, consisting of states, returns-to-go (target return set to 80 during evaluation), and rewards. Note that this differs from the sequence representation used by Laskin et al. [2022]. We set the context length for all agents to the equivalent of two episodes, which amounts to 200 timesteps in total.

In Figure 17, we report the ICL performance over the 40 ICL trials on (a) 80 training locations and (b) 20 hold-out locations for the 4 different backbones considered in this work. We observe that the recurrent backbones attain considerably higher scores than the Transformer backbone. Furthermore, we find that xLSTM [7:1] attains the highest overall scores, which we attribute to the state-tracking abilities [Merrill et al., 2024] of sLSTM blocks. We aim to explore the ICL abilities of modern recurrent backbones more in future work.

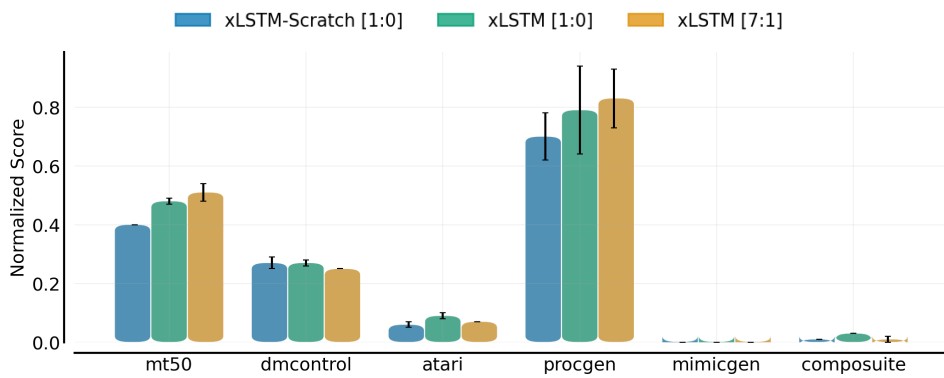

**Figure 16: Fine-tune performance on hold-out tasks**. We compare the performance of a pretrained xLSTM against an xLSTM trained from scratch, both with 16 million parameters. We select the top 5% percent of trajectories from our held-out tasks based on performance and used this subset to fine-tune the models. We perform 25K update steps during fine-tuning and show the normalized scores, averaged across held-out tasks from each domain.

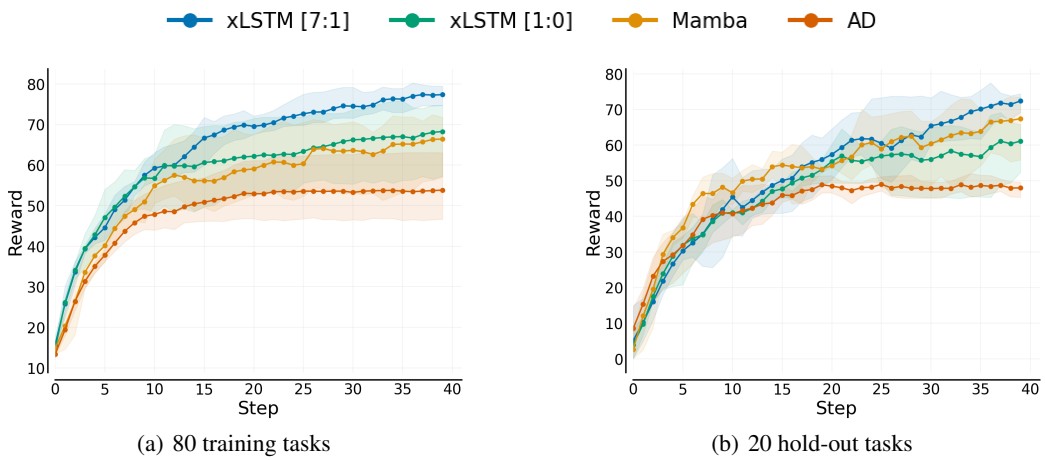

(a) 80 training tasks          (b) 20 hold-out tasks

**Figure 17:** In-context Learning on Dark-Room $10 \times 10$.

## C.5   INFERENCE TIME COMPARISONS

We empirically examine the difference in inference speed between of our models. Similar to De et al. [2024], we report both latency and throughput. For real-time applications, latency is the more important dimension, and therefore we focus our analysis on latency.

### C.5.1   LATENCY

In Figures 18 and 19, we report the latencies for DT and xLSTM with the same number of layer blocks as DT, and twice the number of layers blocks as DT, respectively. We conduct our comparison for two different batch sizes and across varying sequence lengths.

### C.5.2   THROUGHPUT

In Figures 20 and 21, we similarly report the attained throughput for DT and xLSTM with the same number of layer blocks as DT, and twice the number of layers blocks as DT, respectively. We conduct our comparison for two fixed context lengths and varying batch sizes.

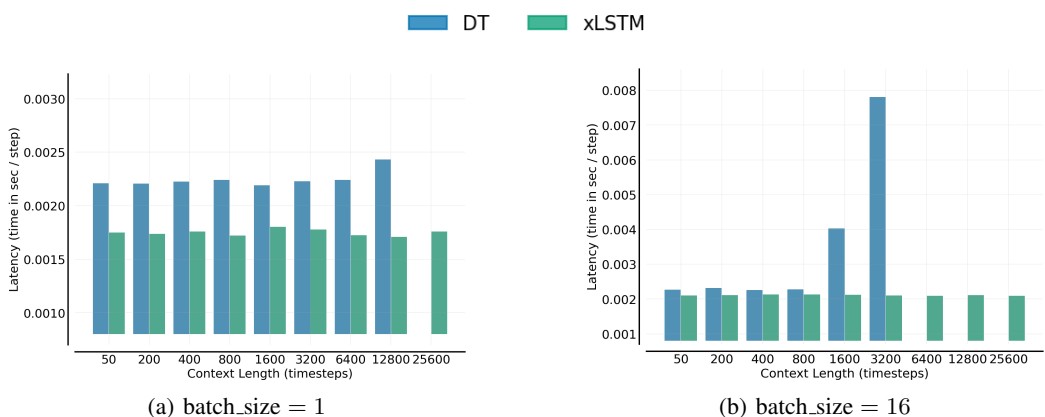

(a) batch_size = 1            (b) batch_size = 16

**Figure 18:** Latency. We report latency with (a) batch size of 1 and (b) batch size of 16 for DT and xLSTM with 206M parameters. For xLSTM we use the same number of layer blocks as DT and a higher hidden dimension to match parameters.

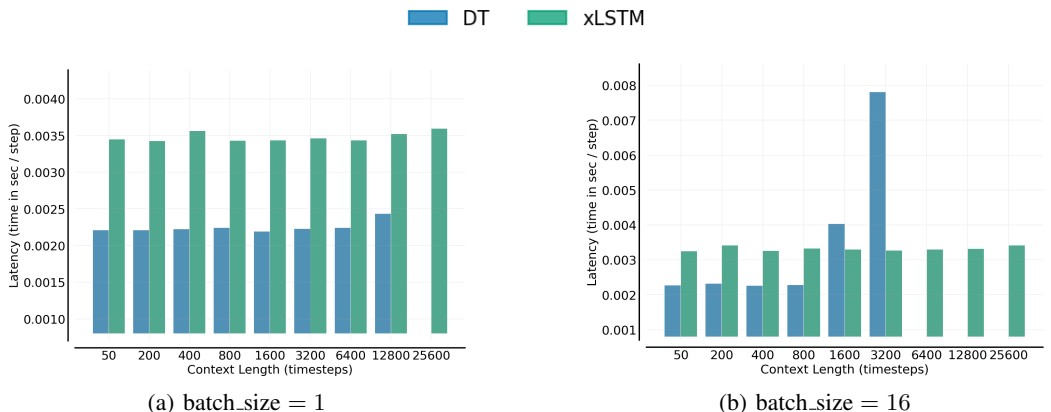

(a) batch_size = 1            (b) batch_size = 16

**Figure 19:** Latency. We report latency with (a) batch size of 1 and (b) batch size of 16 for DT and xLSTM with 206M parameters. For xLSTM, we use twice the number of layer blocks and the same hidden dimension as the Transformer.

### C.5.3 xLSTM KERNEL COMPARISONS

We leverage custom kernels for xLSTM to conduct our inference-speed comparisons. In particular, we compare 4 variants: recurrent-style inference with and without kernel acceleration, and chunkwise inference with and without kernel acceleration. In our experiments, every timestep contains 3 individual tokens. Consequently, regular recurrent-style inference requires iterating over the token sequence of length 3 in a loop given the hidden state of the previous timestep. This requires 3 forward passes. In contrast, the chunkwise implementation operates on chunks of timesteps given a hidden state. Consequently, this only requires a single forward pass. In Figure 22, we illustrate the impact of kernel acceleration. We find that our chunkwise kernels result in considerably lower latencies. Interestingly, we find that for $B = 1$, our chunkwise implementation without kernel acceleration is faster than the recurrent-style inference with kernel acceleration. However, as the batch size increases, this trend reverses. This highlights the importance of kernel acceleration for efficient inference.

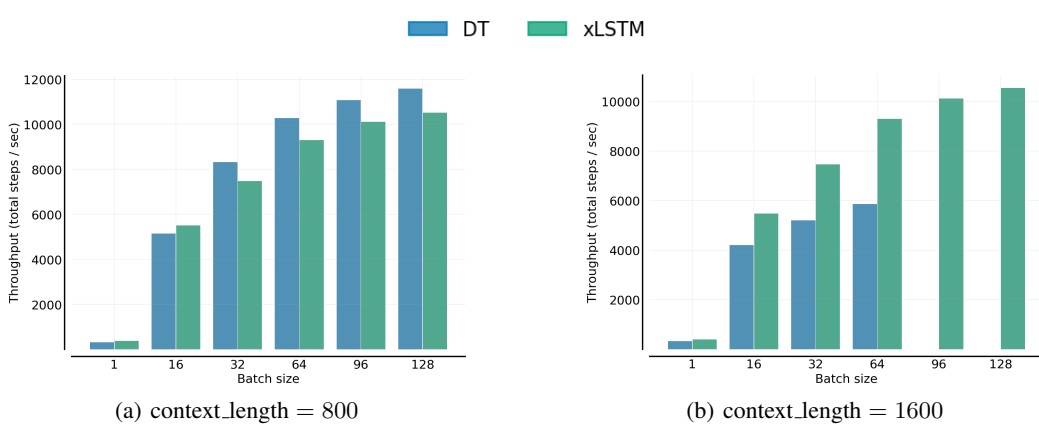

(a) context_length = 800

(b) context_length = 1600

**Figure 20:** Throughput. We report throughput with (a) context size of 800, and (b) context size of 1600 timesteps for DT and xLSTM with 206M parameters. For xLSTM we use the same number of layer blocks as DT and a higher hidden dimension to match parameters.

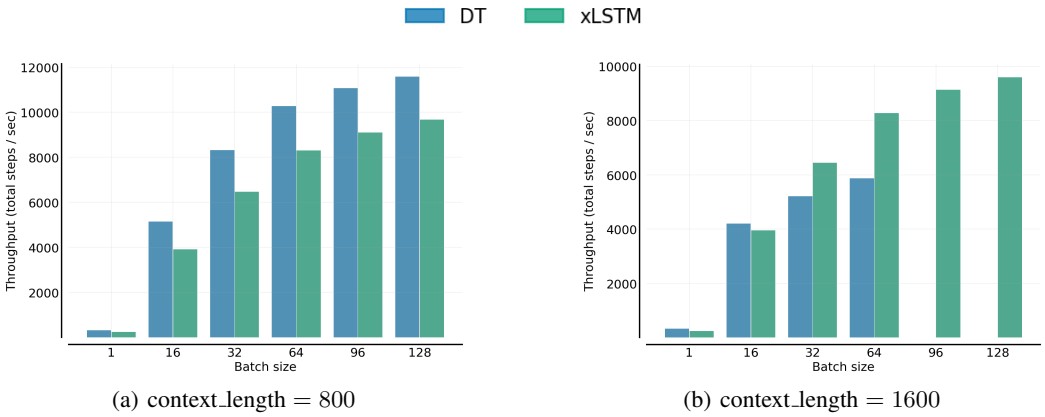

(a) context_length = 800

(b) context_length = 1600

**Figure 21:** Throughput. We report throughput with (a) context size of 800, and (b) context size of 1600 timesteps for DT and xLSTM with 206M parameters. For xLSTM, we use twice the number of layer blocks and the same hidden dimension as the Transformer.

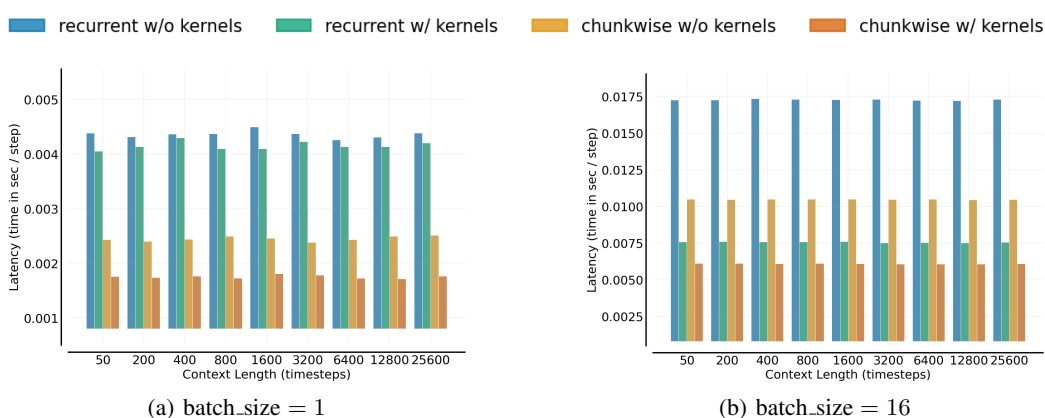

(a) batch_size = 1

(b) batch_size = 16

**Figure 22:** Impact of kernel acceleration. We report latency with (a) batch size of 1 and (b) batch size of 32 for DT and xLSTM with 206M parameters. For xLSTM we use the same number of layer blocks as DT and a higher hidden dimension to match parameters.

# D ABLATIONS

## D.1 REMOVING ACTION CONDITION

### D.1.1 DT ON META-WORLD

We found that removing actions from the context results in better performance across backbones. In Figure 23, we report the learning curves over 200K updates for DT with varying context lengths on Meta-World, both with and without actions in the context. While context lengths beyond 1 hurt performance when training with actions, the reverse is true when training without actions. This is in contrast to recent works, which did not benefit from longer contexts [Octo Model Team et al., 2024]. However, while removing actions improves performance on Meta-World, it does not affect performance on discrete control. On Meta-World, we observed that the models become overly confident (high action logits), which is problematic if poor initial actions are produced. We assume this is because in robotics actions change smoothly and by observing previous actions the agent learns shortcuts. A similar issue has been identified by Wen et al. [2020], and termed the *copycat problem*, because the agent is incentivized to copy previous actions. Our solution is to remove actions from the input sequence. This prevents the agent from learning shortcuts and alleviates the copycat problem.

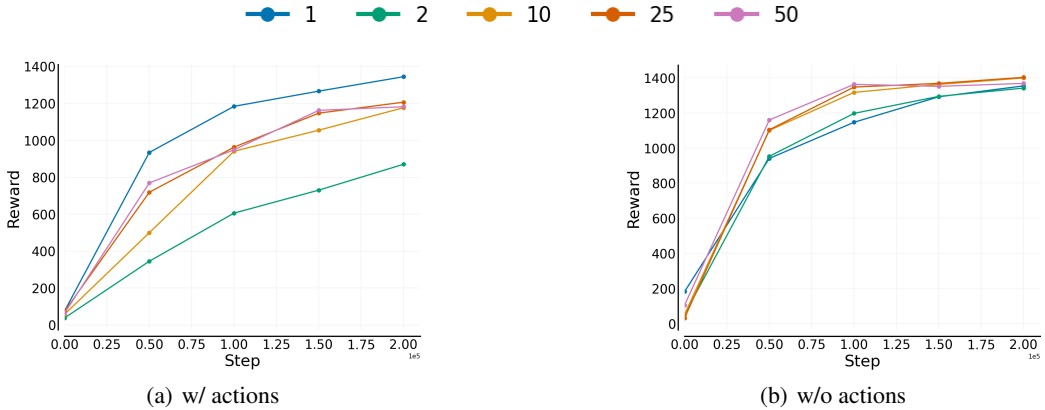

(a) w/ actions  (b) w/o actions

**Figure 23:** Ablation on removing the **action condition** for varying context lengths $C$. Performance of DT **(a)** with, and **(b)** without action condition on **Meta-World**. With action in the context, $C > 1$ harms performance due to overconfidence in action predictions. Without actions in the context, the performance of DT improves with increasing $C$.

### D.1.2 DT ON ALL 432 TASKS.

To further investigate the effect of removing actions from the context, we repeat this ablation on the full 432 tasks and 6 domains at the 206M model scale. In Figure 24, we report the learning curves for a DT with varying sequence lengths trained (a) with and (b) without actions in the agent's context. Similar to the single-domain study on Meta-World with smaller models, we find that providing a longer context does not improve performance, resulting in a normalized score of around 0.3 across domains. In contrast, without action in the context, we observe a consistent improvement in the evaluation performance as the sequence length increases. In fact, the normalized score increase from around 0.3 with $C = 1$ to 0.7 with $C = 50$. For computational reasons we only report one seed per sequence length in this experiment, but we believe that the overall trends are clear.

To better understand on which domains the longer context benefits or hurts our agents, we also present the normalized score per domain in Figure 25. Without actions in the context, we find that longer context consistently benefits the performance across domains. With actions in the context we observe that on Meta-World and DMControl, the performance deteriorates for $C > 1$. In contrast, on the discrete control domains Atari and Procgen, but also on the continuous continous control domain Composuite, performance tends to improve with $C > 1$. This suggests that the copycat problem is particularly present on Meta-World and DMControl. However, note that the final performances

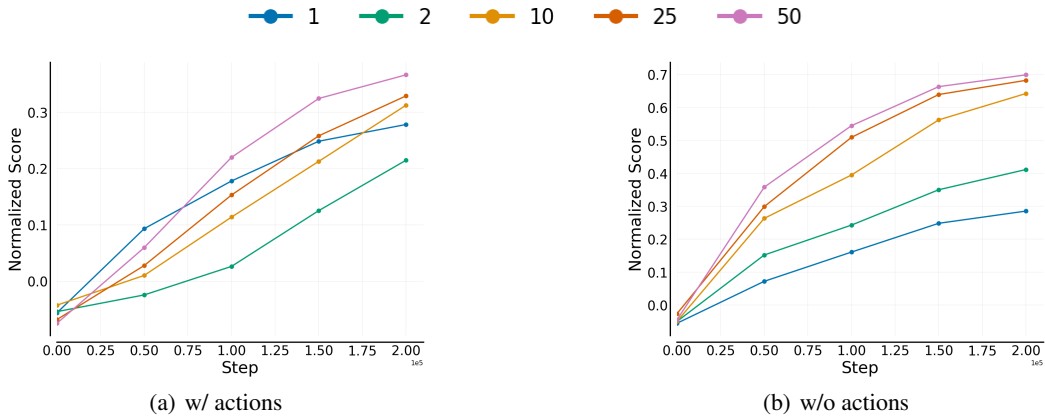

(a) w/ actions                                    (b) w/o actions

**Figure 24:** Ablation on removing the **action condition** for varying context lengths $C$. Performance of DT **(a)** with, and **(b)** without action condition on all **432 tasks**. Without actions in the context, the performance of DT improves with increasing $C$.

on Atari, Procgen and Mimicgen are considerably worse when actions are present in the context compared to when they are not.

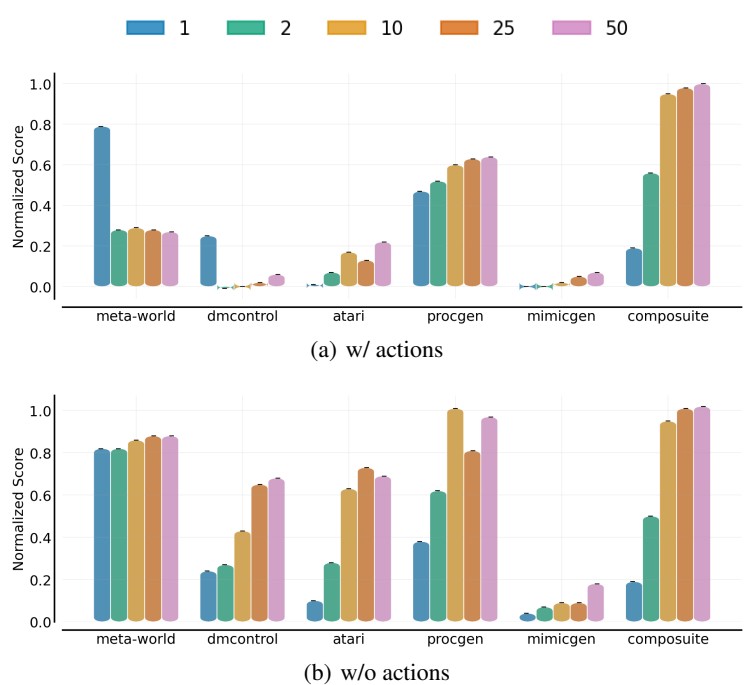

(a) w/ actions

(b) w/o actions

**Figure 25:** Ablation on removing the **action condition** for varying context lengths $C$. We show the normalized score **per domain** for all context lengths (a) with and (b) without actions.

To further investigate this, we compute the MSE between subsequent actions in the training dataset (similar to Wen et al. [2020]) for the continuous control domains and report them in Table 7. Indeed we find that Meta-World and DMControl exhibit significantly lower MSEs between subsequent actions than Composuite. While Mimicgen also exhibits a low MSE between consecutive actions, all backbones perform poorly on this challenging benchmark. Consequently, we conclude that removing actions from the agent's context is particularly effective for domains where actions change smoothly.

**Table 7:** Average MSE ($\pm$ standard deviation) between subsequent actions in robotics datasets.

|  | **Meta-World** | **DMControl** | **Composuite** | **Mimicgen** |
|---|---|---|---|---|
| **Avg. MSE** | $0.08_{\pm 0.09}$ | $0.2_{\pm 0.22}$ | $2.1_{\pm 0.3}$ | $0.015_{\pm 0.007}$ |

This result highlights the fact that large action models can strongly benefit from increased context length even on the simulated environments we consider in this work. Furthermore, we believe that this effect can be even bigger in complex real-world environments that require longer-term interactions.

### D.1.3 xLSTM ON ALL 432 TASKS.

To validate that modern recurrent backbones also benefit from training with longer sequence lengths, we repeat the same ablation as presented in Appendix D.1.2 using xLSTM [1:0]. We report the learning curves validation perplexities and evaluation performance across all 432 tasks for varying context lengths in Figure 26. Note that the validation perplexity curves in Figure 26a, start at step 50K for readability. Again, we observe considerable improvements in the validation perplexities and in the normalized scores (0.4 for $C = 1$ to 0.8 for $C = 50$) as the context length increases.

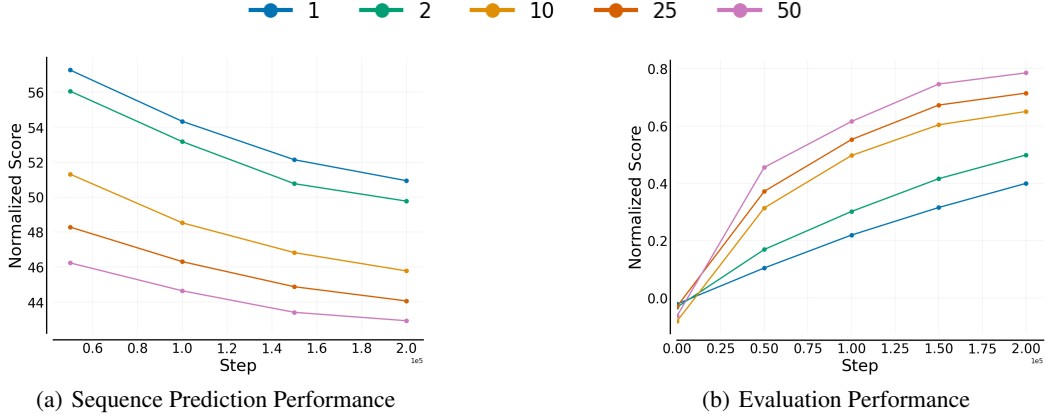

(a) Sequence Prediction Performance  (b) Evaluation Performance

**Figure 26:** Ablation on the effect of for varying the context length $C$ for xLSTM. We report (a) validation perplexity and (b) evaluation performance across the 432 training tasks for xLSTM [1:0]. Without actions in the context, the performance of DT improves with increasing $C$.

In addition, we provide the normalized scores per domain for xLSTM with varying sequence lengths in Figure 27. Across domains, we observe increasing performance with increasing $C$.

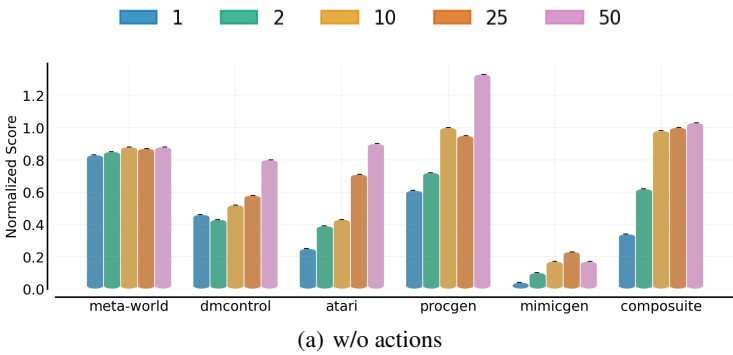

(a) w/o actions

**Figure 27:** Ablation on the effect of for varying the context length $C$ for xLSTM. We show the normalized scores **per domain** for all context lengths.

## D.2 RETURN-CONDITIONING VS. BEHAVIOR CLONING

Across experiments presented in the main text, except for the ICL experiments, we utilized a sequence representation that includes return-to-go tokens (RTG) as commonly used in the DT literature [Chen et al., 2021; Lee et al., 2022]. At inference time, the RTG allows to condition the model on a high target return to produce high-quality actions. This is particularly useful when the datasets contain a mixture of optimal and suboptimal trajectories. However, many recent works focus on behavior cloning without return conditioning [Brohan et al., 2023b;a; Octo Model Team et al., 2024].

To better understand whether our findings transfer to the behavior cloning setting, we conduct an ablation study in which we exclude the RTG tokens from the sequence representation. This means the sequence only consists of state and reward tokens. In Figure 28, we report the (a) validation perplexities and (b) evaluation performance on the 432 task for the four considered backbones. We retain the same training settings and datasets as reported in Appendix B (200K updates, evaluation after every 50K steps). We observe similar learning dynamics as for the 206M models that include RTG tokens in the sequence representation (see Figure 2 and Figure 12). Consequently, we conclude that the same performance trends holds for training the considered backbones with and without return condition. Note, that the final performances are lower compared to the models that include the RTG condition and that can be conditioned on a high return at inference time.

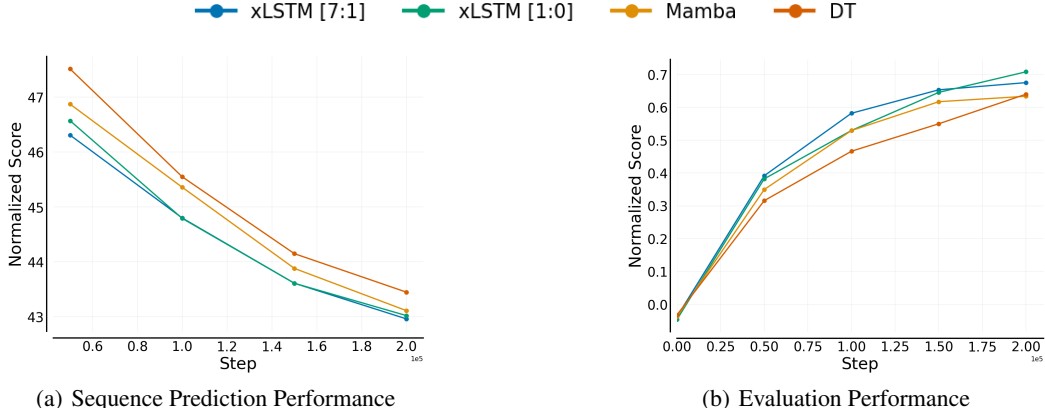

(a) Sequence Prediction Performance                    (b) Evaluation Performance

**Figure 28:** Ablation on the effect of omitting the RTG condition. We report the learning curves for (a) validation perplexity and (b) evaluation performance across the 432 training tasks for 206M parameter models. We observe similar performance trends as when including the RTG in the sequence.

## D.3 EFFECT OF MLSTM-TO-SLSTM RATIO.

Throughout our experiments, we compare two xLSTM variants: xLSTM [7:1] and xLSTM [1:0]. The bracket notation was introduced by [Beck et al., 2024], and denotes the ratio of mLSTM to sLSTM blocks. For example, xLSTM [7:1] contains 1 sLSTM block for every 7 mLSTM blocks. As described in Appendix B, we aim to maintain the same ration as proposed by Beck et al. [2024]. While mLSTM blocks are fully parallelizable, sLSTM blocks are not. However, sLSTM preserves the non-diagonalized recurrent matrix to enable state-tracking [Merrill et al., 2024]. As such, sLSTM can be attractive for tasks that require state-tracking (see Figure 4 in Beck et al. [2024]).

We first conduct an ablation study on the effect of the mLSTM-to-sLSTM ratio on the evaluation performance across all 432 tasks. For this experiment, we use the 16M parameter model that contains 8 xLSTM blocks in total. Consequently, we compare the following ratios [1:0] (only mLSTM), [0:1] (only sLSTM), [1:1], [1:3], [7:1]. In addition, we investigate the placement of sLSTMs across all 8 blocks. To indicate the placement, we use @ followed by the layer index (starting at 0). For example, [3:1] @ 1,3 indicates that the second and fourth layer are sLSTMs. In Figure 29 we report the validation perplexities and evaluation performance for different ratios and layer placements across the 432 tasks. For computational reasons, we conduct this experiment with only 1 seed per ratio. We find that at the 16M parameter scale, xLSTM [1:0] on average outperforms the variants that leverage

sLSTM blocks. This indicates that these domains do not strongly benefit from the state tracking abilities of sLSTM.

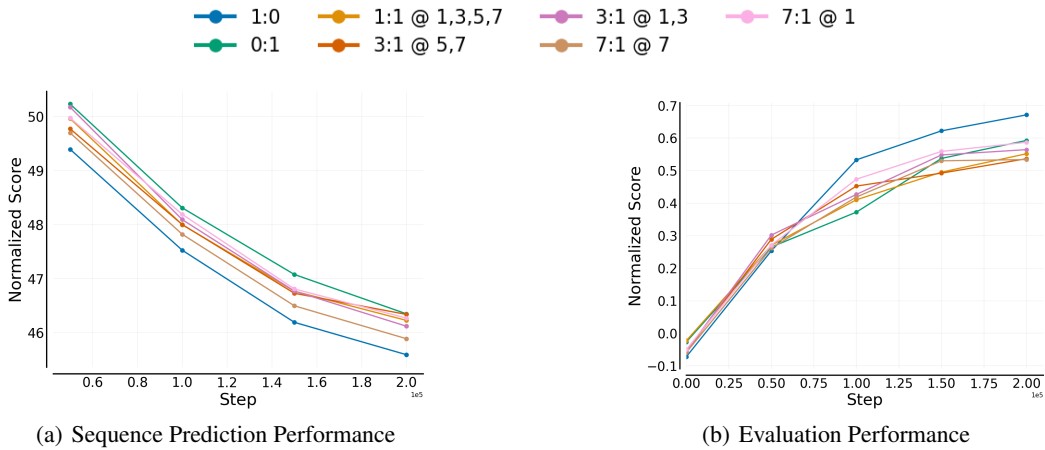

(a) Sequence Prediction Performance

(b) Evaluation Performance

**Figure 29:** Ablation on the effect of the **mLSTM-to-sLSTM ratio**. We report the learning curves for (a) validation perplexity and (b) evaluation performance across the 432 training tasks for 206M parameter models with varying ratios.

Next, conduct the same analysis on Dark-Room $10 \times 10$ ICL environment as used in Appendix C.4. Unlike most of the 432 tasks used in our main experiments, Dark-Room exhibits a partially-observable observation space and sparse rewards. Consequently, Dark-Room is more likely to require state tracking abilities. In fact, we already observed better performance for xLSTM [7:1] than for xLSTM [1:0] in Appendix 17. In Figure 30, we report the ICL curves for the 80 train tasks and 20 hold-out tasks. We observe that xLSTM variants that contain sLSTM blocks at lower-level positions, such as [7:1] @ 1 and [3:1] @ 1,3 outperform xLSTM [1:0]. In contrast, xLSTM variants that contain sLSTM blocks at deeper-level positions, such as [0:1] and 3:1 @ 5,7, perform poorly. This is similar to findings by Beck et al. [2024] who also place sLSTM layers at lower-level positions.

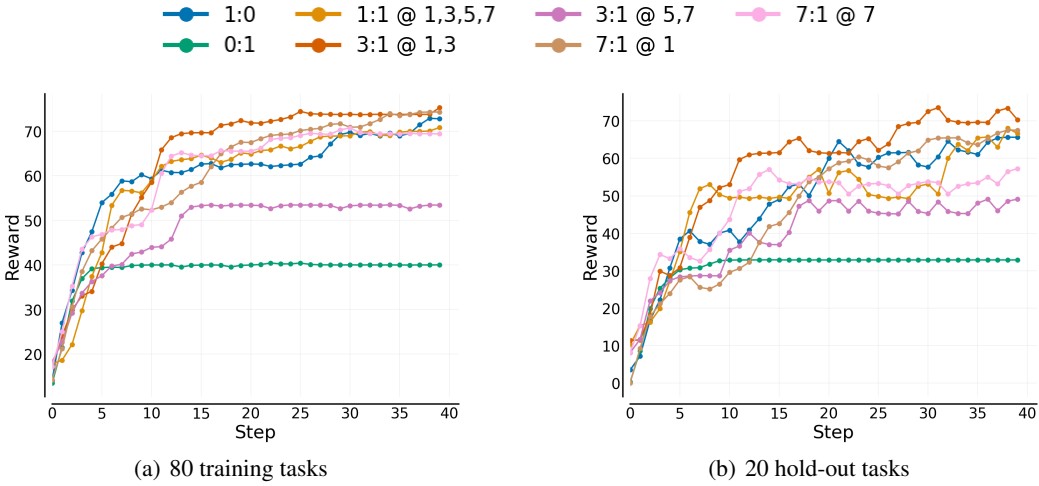

(a) 80 training tasks

(b) 20 hold-out tasks

**Figure 30:** In-context Learning on Dark-Room $10 \times 10$ for varying mLSTM-to-sLSTM ratios.

We conclude that sLSTM layers can be important building blocks for tasks that require state-tracking, such as Dark-Room. Most of the 432 tasks we consider in the main experiments of this work contain fully observable observation spaces and may not require state-tracking. However, we believe that more complex tasks with longer horizons or partial observability, as is common in real-world applications, could greatly benefit from the state-tracking abilities provided by sLSTM blocks. As

such equipping an agent with the ability to perform state-tracking by including sLSTM blocks may be valuable option for practicioners. This is a distinguishing factor of xLSTM from Mamba, which does not exhibit state-tracking.

## D.4 Effect of Dropout in DT

DTs use by default a Dropout [Srivastava et al., 2014] rate of 0.1. However, during our experiments, we found that Dropout has detrimental effects on the evaluation performance, particularly on continuous control domains like Composuite. In Figure 31, we show the validation perplexities and evaluation performance for a DT trained with and without Dropout. Consequently, we remove Dropout from our DT variant.

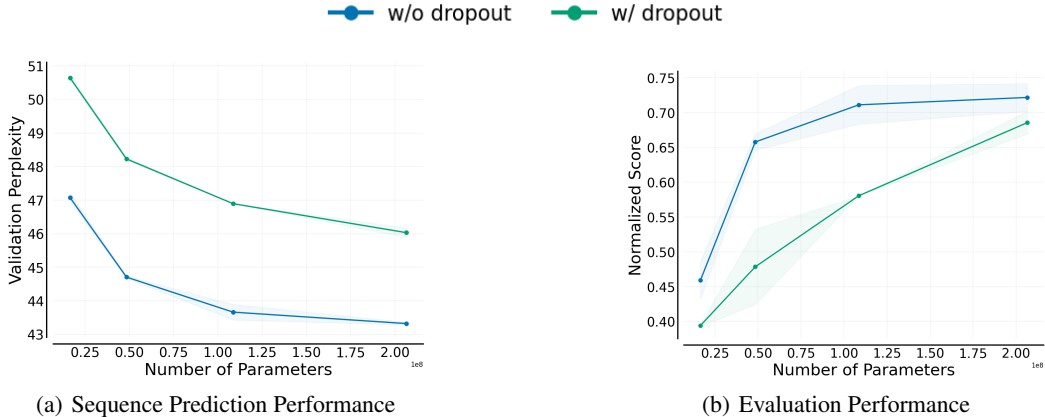

(a) Sequence Prediction Performance      (b) Evaluation Performance

**Figure 31:** Ablation on the **effect of dropout on DT** performance. We show the **(a)** validation perplexity and **(b)** evaluation performance on the training tasks. DT performance drops considerably if training with dropout.

## D.5 Effect of reducing number of layers in xLSTM

In prior works, xLSTM and Mamba use twice the number of layers blocks as the Transformer baseline, while maintaining the same hidden dimension [Gu & Dao, 2023; Beck et al., 2024]. For our inference-time comparisons, we therefore reduce the number of layer blocks in xLSTM by half. To ensure a fair comparison, we consequently adjust the hidden size of xLSTM to match the number of parameters of the Transformer baseline. In this section, we investigate the effect of these modifications of the xLSTM architecture on the model performance.

In Figure 32, report the validation perplexities and evaluation performance for the *regular* xLSTM with twice the number of layer blocks as DT, and an xLSTM with *half* the number of blocks. Reducing the number of layer blocks results in slight decrease in performance on both metrics. However, xLSTM still outperforms the Transformer baseline (see Figure 2).

## E Embedding Space Analysis

In Figure 5, we analyze the representations learned by our models using UMAP [McInnes et al., 2018]. Here, we explain the clustering procedure in more detail. For every task, we sample 32 sub-trajectories containing 50 timesteps (150 tokens) and encode them using our sequence models. Then, we extract the hidden states at the last layer of our model and aggregate them via mean pooling. We cluster all vectors using default hyperparameters of UMAP into a two-dimensional space. Finally, we color the resulting points by their domain. Generally, we find that tasks from the same domain cluster together.

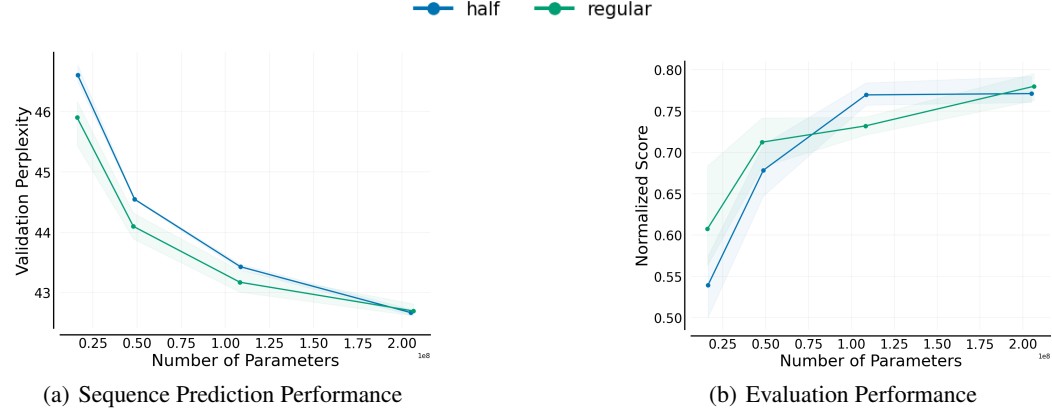

(a) Sequence Prediction Performance

(b) Evaluation Performance

**Figure 32:** Ablation on the effect of reducing the number of layer blocks in xLSTM. We show the **(a)** validation perplexity and **(b)** evaluation performance on the training tasks for the layer regular and layer-matched matched xLSTM models. Reducing the number of layer blocks in xLSTM results in a slight performance decrease.

## F RAW SCORES

In this section, we report the raw scores for all 432 training tasks for the 206M parameter scale. See Tables 8, 9, 10, 11, 12 for Procgen, Atari, Meta-World, DMControl, and Mimicgen, respectively. The raw scores for Composuite are available in Tables 13, 14, 15, and 16.

**Table 8:** Raw Scores for Procgen.

| Task | DT | Mamba | xLSTM [1:0] | xLSTM [7:1] |
|------|------|-------|-------------|-------------|
| bigfish | 2.53 | 2.0 | 4.6 | 5.13 |
| bossfight | 6.73 | 4.1 | 9.27 | 2.0 |
| caveflyer | 6.67 | 6.3 | 6.67 | 4.87 |
| chaser | 3.41 | 3.91 | 4.92 | 4.2 |
| coinrun | 10.0 | 9.0 | 10.0 | 10.0 |
| dodgeball | 2.8 | 3.4 | 4.27 | 3.87 |
| fruitbot | 13.33 | 19.8 | 19.73 | 19.27 |
| heist | 7.33 | 7.0 | 6.67 | 6.67 |
| leaper | 5.33 | 4.0 | 8.67 | 5.33 |
| maze | 8.67 | 10.0 | 7.33 | 7.33 |
| miner | 8.07 | 11.0 | 9.0 | 8.27 |
| starpilot | 24.93 | 10.1 | 21.8 | 28.2 |
| **Avg. Reward** | 8.32 | 7.55 | 8.73 | 8.76 |

**Table 9:** Raw Scores for Atari.

| Task | DT | Mamba | xLSTM [1:0] | xLSTM [7:1] |
|------|------|------|------|------|
| Amidar | 82.27 | 30.8 | 71.07 | 26.73 |
| Assault | 438.2 | 224.7 | 410.2 | 494.13 |
| Asterix | 573.33 | 540.0 | 763.33 | 583.33 |
| Atlantis | 42573.33 | 97240.0 | 83760.0 | 76973.33 |
| BankHeist | 2.67 | 9.0 | 0.0 | 8.67 |
| BattleZone | 2000.0 | 2400.0 | 2600.0 | 1733.33 |
| BeamRider | 126.13 | 61.6 | 176.0 | 243.47 |
| Boxing | 80.8 | 77.7 | 83.8 | 84.93 |
| Breakout | 68.13 | 136.6 | 92.93 | 93.73 |
| Carnival | 618.67 | 424.0 | 697.33 | 484.0 |
| Centipede | 1802.13 | 1238.2 | 2416.73 | 1806.6 |
| ChopperCommand | 813.33 | 800.0 | 813.33 | 766.67 |
| CrazyClimber | 96853.33 | 65960.0 | 106606.67 | 79873.33 |
| DemonAttack | 100.0 | 65.0 | 181.33 | 130.67 |
| DoubleDunk | -2.53 | -3.0 | -2.93 | -3.87 |
| Enduro | 34.53 | 65.5 | 98.73 | 48.53 |
| FishingDerby | -72.47 | -68.2 | -72.07 | -71.0 |
| Freeway | 29.0 | 29.8 | 30.0 | 28.6 |
| Frostbite | 774.67 | 1248.0 | 1162.67 | 1049.33 |
| Gopher | 314.67 | 34.0 | 132.0 | 12.0 |
| Gravitar | 116.67 | 175.0 | 176.67 | 136.67 |
| Hero | 14004.67 | 11381.0 | 14688.67 | 16522.0 |
| IceHockey | -4.8 | -6.3 | -7.6 | -5.93 |
| Jamesbond | 490.0 | 540.0 | 603.33 | 510.0 |
| Kangaroo | 1426.67 | 2880.0 | 2620.0 | 2653.33 |
| Krull | 8880.67 | 10090.0 | 8918.0 | 9569.33 |
| KungFuMaster | 8866.67 | 12700.0 | 8120.0 | 11233.33 |
| NameThisGame | 7976.67 | 7967.0 | 7789.33 | 7232.0 |
| Phoenix | 592.0 | 1600.0 | 1807.33 | 1052.67 |
| Pooyan | 283.33 | 87.5 | 371.67 | 406.67 |
| Qbert | 4306.67 | 1700.0 | 805.0 | 2613.33 |
| Riverraid | 2888.67 | 6923.0 | 6688.0 | 7446.67 |
| RoadRunner | 1320.0 | 350.0 | 1340.0 | 213.33 |
| Robotank | 18.67 | 13.2 | 23.07 | 25.13 |
| Seaquest | 182.67 | 396.0 | 448.0 | 209.33 |
| TimePilot | 2533.33 | 3520.0 | 3200.0 | 2966.67 |
| UpNDown | 10598.0 | 12043.0 | 15340.67 | 12815.33 |
| VideoPinball | 1669.07 | 0.0 | 220.4 | 140.6 |
| WizardOfWor | 113.33 | 160.0 | 160.0 | 206.67 |
| YarsRevenge | 14356.27 | 14499.0 | 16815.0 | 21403.67 |
| Zaxxon | 0.0 | 0.0 | 20.0 | 0.0 |
| **Avg. Reward** | 5556.81 | 6281.27 | 6705.61 | 6383.35 |

**Table 10:** Raw Scores for Meta-World.

| Task | DT | Mamba | xLSTM [1:0] | xLSTM [7:1] |
|---|---|---|---|---|
| reach | 1860.69 ± 12.51 | 1859.3 ± 5.79 | 1859.17 ± 12.62 | 1864.37 ± 6.57 |
| push | 1588.19 ± 207.0 | 1605.03 ± 107.81 | 1493.31 ± 238.01 | 1759.33 ± 3.89 |
| pick-place | 137.85 ± 99.18 | 161.74 ± 153.95 | 389.81 ± 37.36 | 296.21 ± 43.77 |
| door-open | 1552.95 ± 6.51 | 1562.39 ± 6.79 | 1569.35 ± 6.71 | 1570.16 ± 14.83 |
| drawer-open | 1735.13 ± 21.76 | 1714.4 ± 19.3 | 1740.48 ± 9.2 | 1747.33 ± 3.88 |
| drawer-close | 1856.67 ± 3.06 | 1858.05 ± 2.75 | 1858.7 ± 2.34 | 1859.33 ± 1.15 |
| button-press-topdown | 1322.3 ± 3.12 | 1326.55 ± 19.93 | 1341.5 ± 3.15 | 1322.83 ± 7.25 |
| peg-insert-side | 1557.59 ± 98.52 | 1607.59 ± 9.1 | 1640.43 ± 13.1 | 1574.75 ± 90.34 |
| window-open | 1594.16 ± 34.13 | 1568.55 ± 14.38 | 1576.82 ± 10.21 | 1578.18 ± 70.3 |
| window-close | 1474.26 ± 16.88 | 1443.94 ± 18.99 | 1459.83 ± 18.79 | 1452.21 ± 26.56 |
| door-close | 1538.02 ± 14.64 | 1544.31 ± 3.63 | 1546.0 ± 9.69 | 1541.64 ± 10.5 |
| reach-wall | 1837.64 ± 1.6 | 1845.12 ± 3.06 | 1837.76 ± 3.39 | 1777.17 ± 94.47 |
| pick-place-wall | 1041.54 ± 219.67 | 843.51 ± 224.6 | 206.88 ± 184.28 | 385.57 ± 151.52 |
| push-wall | 1689.67 ± 12.74 | 1701.7 ± 1.54 | 1599.63 ± 189.06 | 1487.69 ± 195.8 |
| button-press | 1512.08 ± 9.54 | 1488.1 ± 38.83 | 1541.77 ± 5.48 | 1527.3 ± 10.16 |
| button-press-topdown-wall | 1314.49 ± 62.73 | 1295.2 ± 6.62 | 1321.26 ± 17.59 | 1328.74 ± 24.16 |
| button-press-wall | 1359.83 ± 173.51 | 1547.14 ± 13.84 | 1326.57 ± 109.09 | 1267.11 ± 8.78 |
| peg-unplug-side | 1415.68 ± 162.54 | 1517.49 ± 25.27 | 1393.98 ± 173.0 | 1422.64 ± 192.05 |
| disassemble | 1452.0 ± 44.54 | 1441.18 ± 29.15 | 1220.27 ± 441.51 | 1072.31 ± 374.95 |
| hammer | 1446.68 ± 16.03 | 1683.04 ± 4.82 | 1669.54 ± 32.0 | 1642.34 ± 72.23 |
| plate-slide | 1673.66 ± 1.72 | 1676.83 ± 3.0 | 1682.41 ± 5.02 | 1677.52 ± 5.46 |
| plate-slide-side | 1719.4 ± 7.85 | 1694.35 ± 46.29 | 1686.38 ± 61.27 | 1690.72 ± 12.97 |
| plate-slide-back | 1790.96 ± 6.39 | 1787.65 ± 5.99 | 1797.78 ± 1.17 | 1797.17 ± 0.43 |
| plate-slide-back-side | 1773.26 ± 9.72 | 1763.24 ± 5.59 | 1785.11 ± 7.42 | 1788.61 ± 6.67 |
| handle-press | 1734.75 ± 220.82 | 1829.07 ± 29.91 | 1881.23 ± 15.62 | 1881.92 ± 10.56 |
| handle-pull | 1590.74 ± 35.98 | 1627.4 ± 34.18 | 1616.62 ± 52.0 | 1627.6 ± 21.86 |
| handle-press-side | 1852.25 ± 7.0 | 1857.4 ± 10.13 | 1847.95 ± 5.61 | 1857.36 ± 5.57 |
| handle-pull-side | 1651.05 ± 3.48 | 1607.3 ± 22.56 | 1655.75 ± 4.6 | 1651.77 ± 7.53 |
| stick-push | 1595.45 ± 6.88 | 1585.22 ± 5.17 | 1595.35 ± 3.29 | 1595.21 ± 0.88 |
| stick-pull | 1377.41 ± 108.31 | 1401.91 ± 32.79 | 1460.27 ± 57.13 | 1442.68 ± 43.23 |
| basketball | 1529.79 ± 11.41 | 1528.22 ± 18.23 | 1543.02 ± 2.49 | 1542.8 ± 17.81 |
| soccer | 649.69 ± 160.32 | 929.06 ± 64.35 | 792.21 ± 139.63 | 732.44 ± 290.49 |
| faucet-open | 1676.95 ± 121.6 | 1703.83 ± 41.97 | 1727.05 ± 45.15 | 1744.83 ± 15.93 |
| faucet-close | 1772.91 ± 9.23 | 1772.13 ± 2.35 | 1778.25 ± 3.96 | 1775.25 ± 0.79 |
| coffee-push | 340.21 ± 276.9 | 232.01 ± 225.2 | 61.35 ± 51.79 | 41.79 ± 40.9 |
| coffee-pull | 1346.29 ± 101.93 | 1261.39 ± 195.18 | 1409.68 ± 34.66 | 1293.92 ± 129.94 |
| coffee-button | 1595.94 ± 16.57 | 1592.77 ± 2.23 | 1593.15 ± 49.98 | 1562.92 ± 36.79 |
| sweep | 1485.79 ± 12.17 | 1452.38 ± 13.74 | 1508.58 ± 14.96 | 1471.73 ± 29.08 |
| sweep-into | 1796.25 ± 7.64 | 1472.64 ± 455.9 | 1804.27 ± 2.38 | 1786.27 ± 14.64 |
| pick-out-of-hole | 1437.38 ± 181.15 | 1499.35 ± 35.73 | 1529.83 ± 8.09 | 1415.91 ± 176.44 |
| assembly | 1229.39 ± 16.96 | 1216.34 ± 22.21 | 1236.68 ± 21.77 | 1227.81 ± 7.67 |
| shelf-place | 1446.07 ± 30.41 | 1448.75 ± 39.73 | 1485.4 ± 12.31 | 1463.53 ± 9.04 |
| push-back | 1226.32 ± 172.59 | 1022.98 ± 158.35 | 1011.25 ± 396.65 | 1027.48 ± 303.73 |
| lever-pull | 1604.74 ± 3.32 | 1634.06 ± 6.08 | 1639.31 ± 10.11 | 1626.09 ± 23.72 |
| dial-turn | 1688.33 ± 22.94 | 1667.37 ± 41.45 | 1713.38 ± 35.16 | 1686.59 ± 55.09 |
| **Avg. Reward** | 1486.05 | 1486.18 | 1455.15 | 1464.16 |

**Table 11:** Raw Scores for DMControl.

| Task | DT | Mamba | xLSTM [1:0] | xLSTM [7:1] |
|---|---|---|---|---|
| finger-turn-easy | 121.27 ± 104.6 | 396.4 ± 122.47 | 449.8 ± 186.65 | 640.13 ± 82.48 |
| fish-upright | 181.14 ± 70.82 | 154.59 ± 34.64 | 277.23 ± 105.37 | 241.73 ± 257.01 |
| hopper-stand | 296.15 ± 141.83 | 304.78 ± 32.65 | 413.95 ± 35.83 | 392.34 ± 152.75 |
| point_mass-easy | 342.26 ± 37.42 | 720.11 ± 42.95 | 734.95 ± 114.17 | 823.74 ± 57.3 |
| walker-stand | 911.72 ± 38.16 | 785.21 ± 23.53 | 947.31 ± 22.13 | 864.14 ± 181.56 |
| walker-run | 155.91 ± 73.84 | 274.83 ± 0.44 | 201.34 ± 34.77 | 145.01 ± 31.71 |
| ball_in_cup-catch | 976.93 ± 0.83 | 970.9 ± 4.67 | 977.33 ± 0.5 | 975.93 ± 0.42 |
| cartpole-swingup | 688.5 ± 42.6 | 762.4 ± 63.93 | 800.14 ± 13.64 | 591.08 ± 86.49 |
| cheetah-run | 81.21 ± 96.85 | 482.39 ± 17.23 | 358.52 ± 127.92 | 389.04 ± 4.11 |
| finger-spin | 209.27 ± 20.57 | 430.8 ± 61.66 | 673.47 ± 94.37 | 626.93 ± 29.21 |
| reacher-easy | 45.4 ± 5.21 | 180.7 ± 133.64 | 78.73 ± 20.59 | 58.0 ± 13.91 |
| **Avg. Reward** | 364.52 | 496.65 | 505.06 | 522.55 |

**Table 12:** Raw Scores for Mimicgen.

| Task | DT | Mamba | xLSTM [1:0] | xLSTM [7:1] |
|---|---|---|---|---|
| Panda_CoffeePreparation_D0 | 0.0 ± 0.0 | 0.0 ± 0.0 | 0.0 ± 0.0 | 0.13 ± 0.12 |
| Panda_CoffeePreparation_D1 | 0.0 ± 0.0 | 0.0 ± 0.0 | 0.0 ± 0.0 | 0.0 ± 0.0 |
| Panda_Coffee_D0 | 0.4 ± 0.2 | 0.0 ± 0.0 | 0.2 ± 0.2 | 0.07 ± 0.12 |
| Panda_Coffee_D1 | 0.2 ± 0.2 | 0.0 ± 0.0 | 0.2 ± 0.2 | 0.07 ± 0.12 |
| Panda_Coffee_D2 | 0.07 ± 0.12 | 0.0 ± 0.0 | 0.07 ± 0.12 | 0.0 ± 0.0 |
| Panda_HammerCleanup_D0 | 1.0 ± 0.0 | 0.9 ± 0.14 | 1.0 ± 0.0 | 1.0 ± 0.0 |
| Panda_HammerCleanup_D1 | 0.47 ± 0.5 | 0.1 ± 0.14 | 0.47 ± 0.23 | 0.47 ± 0.31 |
| Panda_Kitchen_D0 | 0.87 ± 0.23 | 0.6 ± 0.0 | 1.0 ± 0.0 | 1.0 ± 0.0 |
| Panda_Kitchen_D1 | 0.0 ± 0.0 | 0.0 ± 0.0 | 0.0 ± 0.0 | 0.0 ± 0.0 |
| Panda_MugCleanup_D0 | 0.13 ± 0.12 | 0.1 ± 0.14 | 0.6 ± 0.2 | 0.27 ± 0.12 |
| Panda_MugCleanup_D1 | 0.07 ± 0.12 | 0.0 ± 0.0 | 0.2 ± 0.2 | 0.07 ± 0.12 |
| Sawyer_NutAssembly_D0 | 0.07 ± 0.12 | 0.0 ± 0.0 | 0.0 ± 0.0 | 0.07 ± 0.12 |
| Sawyer_PickPlace_D0 | 0.0 ± 0.0 | 0.0 ± 0.0 | 0.0 ± 0.0 | 0.0 ± 0.0 |
| Panda_Square_D0 | 0.2 ± 0.2 | 0.0 ± 0.0 | 0.53 ± 0.12 | 0.53 ± 0.12 |
| Panda_Square_D1 | 0.0 ± 0.0 | 0.0 ± 0.0 | 0.2 ± 0.2 | 0.07 ± 0.12 |
| Panda_Square_D2 | 0.13 ± 0.12 | 0.0 ± 0.0 | 0.07 ± 0.12 | 0.07 ± 0.12 |
| Panda_StackThree_D0 | 0.0 ± 0.0 | 0.0 ± 0.0 | 0.07 ± 0.12 | 0.0 ± 0.0 |
| Panda_StackThree_D1 | 0.0 ± 0.0 | 0.0 ± 0.0 | 0.07 ± 0.12 | 0.0 ± 0.0 |
| Panda_Stack_D0 | 0.47 ± 0.12 | 0.2 ± 0.0 | 0.67 ± 0.31 | 0.73 ± 0.12 |
| Panda_Stack_D1 | 0.4 ± 0.2 | 0.0 ± 0.0 | 0.27 ± 0.12 | 0.4 ± 0.2 |
| Panda_Threading_D0 | 0.27 ± 0.12 | 0.2 ± 0.0 | 0.27 ± 0.12 | 0.2 ± 0.2 |
| Panda_Threading_D1 | 0.2 ± 0.35 | 0.0 ± 0.0 | 0.07 ± 0.12 | 0.07 ± 0.12 |
| Panda_ThreePieceAssembly_D0 | 0.0 ± 0.0 | 0.0 ± 0.0 | 0.0 ± 0.0 | 0.0 ± 0.0 |
| Panda_ThreePieceAssembly_D1 | 0.0 ± 0.0 | 0.0 ± 0.0 | 0.0 ± 0.0 | 0.0 ± 0.0 |
| IIWA_Coffee_D0 | 0.0 ± 0.0 | 0.0 ± 0.0 | 0.0 ± 0.0 | 0.0 ± 0.0 |
| Sawyer_Coffee_D0 | 0.27 ± 0.31 | 0.0 ± 0.0 | 0.13 ± 0.12 | 0.2 ± 0.2 |
| UR5e_Coffee_D0 | 0.33 ± 0.12 | 0.2 ± 0.0 | 0.47 ± 0.31 | 0.4 ± 0.2 |
| IIWA_Coffee_D1 | 0.0 ± 0.0 | 0.0 ± 0.0 | 0.0 ± 0.0 | 0.0 ± 0.0 |
| Sawyer_Coffee_D1 | 0.07 ± 0.12 | 0.0 ± 0.0 | 0.07 ± 0.12 | 0.0 ± 0.0 |
| UR5e_Coffee_D1 | 0.13 ± 0.12 | 0.0 ± 0.0 | 0.2 ± 0.2 | 0.33 ± 0.31 |
| IIWA_Coffee_D2 | 0.0 ± 0.0 | 0.0 ± 0.0 | 0.0 ± 0.0 | 0.0 ± 0.0 |
| UR5e_Coffee_D2 | 0.0 ± 0.0 | 0.1 ± 0.14 | 0.2 ± 0.0 | 0.07 ± 0.12 |
| IIWA_HammerCleanup_D0 | 0.0 ± 0.0 | 0.0 ± 0.0 | 0.0 ± 0.0 | 0.0 ± 0.0 |
| Sawyer_HammerCleanup_D0 | 0.73 ± 0.12 | 0.9 ± 0.14 | 0.93 ± 0.12 | 0.87 ± 0.23 |
| UR5e_HammerCleanup_D0 | 1.0 ± 0.0 | 0.9 ± 0.14 | 1.0 ± 0.0 | 0.93 ± 0.12 |
| IIWA_HammerCleanup_D1 | 0.0 ± 0.0 | 0.0 ± 0.0 | 0.0 ± 0.0 | 0.0 ± 0.0 |
| Sawyer_HammerCleanup_D1 | 0.2 ± 0.2 | 0.2 ± 0.0 | 0.27 ± 0.23 | 0.4 ± 0.35 |
| UR5e_HammerCleanup_D1 | 0.47 ± 0.12 | 0.4 ± 0.28 | 0.8 ± 0.2 | 0.6 ± 0.0 |
| IIWA_Kitchen_D0 | 0.0 ± 0.0 | 0.0 ± 0.0 | 0.0 ± 0.0 | 0.0 ± 0.0 |
| UR5e_Kitchen_D0 | 0.93 ± 0.12 | 0.8 ± 0.0 | 1.0 ± 0.0 | 1.0 ± 0.0 |
| UR5e_Kitchen_D1 | 0.0 ± 0.0 | 0.0 ± 0.0 | 0.0 ± 0.0 | 0.07 ± 0.12 |
| IIWA_MugCleanup_D0 | 0.0 ± 0.0 | 0.0 ± 0.0 | 0.0 ± 0.0 | 0.0 ± 0.0 |
| IIWA_MugCleanup_D1 | 0.0 ± 0.0 | 0.0 ± 0.0 | 0.0 ± 0.0 | 0.0 ± 0.0 |
| UR5e_MugCleanup_D1 | 0.07 ± 0.12 | 0.0 ± 0.0 | 0.13 ± 0.12 | 0.13 ± 0.12 |
| IIWA_NutAssembly_D0 | 0.0 ± 0.0 | 0.0 ± 0.0 | 0.0 ± 0.0 | 0.0 ± 0.0 |
| Sawyer_NutAssembly_D0 | 0.0 ± 0.0 | 0.0 ± 0.0 | 0.07 ± 0.12 | 0.0 ± 0.0 |
| UR5e_NutAssembly_D0 | 0.0 ± 0.0 | 0.0 ± 0.0 | 0.0 ± 0.0 | 0.07 ± 0.12 |
| IIWA_PickPlace_D0 | 0.0 ± 0.0 | 0.0 ± 0.0 | 0.0 ± 0.0 | 0.0 ± 0.0 |
| Sawyer_PickPlace_D0 | 0.0 ± 0.0 | 0.0 ± 0.0 | 0.0 ± 0.0 | 0.0 ± 0.0 |
| UR5e_PickPlace_D0 | 0.0 ± 0.0 | 0.0 ± 0.0 | 0.0 ± 0.0 | 0.0 ± 0.0 |
| IIWA_Square_D0 | 0.0 ± 0.0 | 0.0 ± 0.0 | 0.0 ± 0.0 | 0.0 ± 0.0 |
| Sawyer_Square_D0 | 0.2 ± 0.2 | 0.4 ± 0.28 | 0.33 ± 0.12 | 0.53 ± 0.23 |
| UR5e_Square_D0 | 0.13 ± 0.23 | 0.3 ± 0.42 | 0.27 ± 0.12 | 0.53 ± 0.23 |
| IIWA_Square_D1 | 0.0 ± 0.0 | 0.0 ± 0.0 | 0.0 ± 0.0 | 0.0 ± 0.0 |
| Sawyer_Square_D1 | 0.0 ± 0.0 | 0.0 ± 0.0 | 0.0 ± 0.0 | 0.0 ± 0.0 |
| UR5e_Square_D1 | 0.0 ± 0.0 | 0.0 ± 0.0 | 0.0 ± 0.0 | 0.0 ± 0.0 |
| IIWA_StackThree_D0 | 0.0 ± 0.0 | 0.0 ± 0.0 | 0.0 ± 0.0 | 0.0 ± 0.0 |
| Sawyer_StackThree_D0 | 0.0 ± 0.0 | 0.0 ± 0.0 | 0.0 ± 0.0 | 0.0 ± 0.0 |
| UR5e_StackThree_D0 | 0.0 ± 0.0 | 0.0 ± 0.0 | 0.0 ± 0.0 | 0.0 ± 0.0 |
| IIWA_StackThree_D1 | 0.0 ± 0.0 | 0.0 ± 0.0 | 0.0 ± 0.0 | 0.0 ± 0.0 |
| Sawyer_StackThree_D1 | 0.0 ± 0.0 | 0.0 ± 0.0 | 0.0 ± 0.0 | 0.07 ± 0.12 |
| UR5e_StackThree_D1 | 0.0 ± 0.0 | 0.0 ± 0.0 | 0.0 ± 0.0 | 0.0 ± 0.0 |
| IIWA_Stack_D0 | 0.0 ± 0.0 | 0.0 ± 0.0 | 0.0 ± 0.0 | 0.0 ± 0.0 |
| Sawyer_Stack_D0 | 0.47 ± 0.31 | 0.2 ± 0.0 | 0.6 ± 0.2 | 0.4 ± 0.2 |
| UR5e_Stack_D0 | 0.4 ± 0.2 | 0.3 ± 0.14 | 0.87 ± 0.12 | 0.67 ± 0.12 |
| IIWA_Stack_D1 | 0.0 ± 0.0 | 0.0 ± 0.0 | 0.0 ± 0.0 | 0.0 ± 0.0 |
| Sawyer_Stack_D1 | 0.2 ± 0.2 | 0.0 ± 0.0 | 0.4 ± 0.2 | 0.27 ± 0.12 |
| UR5e_Stack_D1 | 0.6 ± 0.0 | 0.1 ± 0.14 | 0.73 ± 0.12 | 0.4 ± 0.2 |
| IIWA_Threading_D0 | 0.0 ± 0.0 | 0.0 ± 0.0 | 0.0 ± 0.0 | 0.0 ± 0.0 |
| Sawyer_Threading_D0 | 0.13 ± 0.12 | 0.0 ± 0.0 | 0.07 ± 0.12 | 0.13 ± 0.12 |
| UR5e_Threading_D0 | 0.27 ± 0.31 | 0.1 ± 0.14 | 0.4 ± 0.2 | 0.4 ± 0.2 |
| IIWA_Threading_D1 | 0.0 ± 0.0 | 0.0 ± 0.0 | 0.0 ± 0.0 | 0.0 ± 0.0 |
| Sawyer_Threading_D1 | 0.0 ± 0.0 | 0.0 ± 0.0 | 0.13 ± 0.12 | 0.0 ± 0.0 |
| UR5e_Threading_D1 | 0.07 ± 0.12 | 0.0 ± 0.0 | 0.0 ± 0.0 | 0.0 ± 0.0 |
| IIWA_ThreePieceAssembly_D0 | 0.0 ± 0.0 | 0.0 ± 0.0 | 0.0 ± 0.0 | 0.0 ± 0.0 |
| Sawyer_ThreePieceAssembly_D0 | 0.0 ± 0.0 | 0.0 ± 0.0 | 0.0 ± 0.0 | 0.0 ± 0.0 |
| UR5e_ThreePieceAssembly_D0 | 0.0 ± 0.0 | 0.0 ± 0.0 | 0.13 ± 0.12 | 0.0 ± 0.0 |
| IIWA_ThreePieceAssembly_D1 | 0.0 ± 0.0 | 0.0 ± 0.0 | 0.0 ± 0.0 | 0.0 ± 0.0 |
| Sawyer_ThreePieceAssembly_D1 | 0.0 ± 0.0 | 0.0 ± 0.0 | 0.0 ± 0.0 | 0.0 ± 0.0 |
| UR5e_ThreePieceAssembly_D1 | 0.0 ± 0.0 | 0.0 ± 0.0 | 0.0 ± 0.0 | 0.0 ± 0.0 |
| IIWA_ThreePieceAssembly_D2 | 0.0 ± 0.0 | 0.0 ± 0.0 | 0.0 ± 0.0 | 0.0 ± 0.0 |
| Sawyer_ThreePieceAssembly_D2 | 0.0 ± 0.0 | 0.0 ± 0.0 | 0.0 ± 0.0 | 0.0 ± 0.0 |
| UR5e_ThreePieceAssembly_D2 | 0.0 ± 0.0 | 0.0 ± 0.0 | 0.0 ± 0.0 | 0.0 ± 0.0 |

**Table 13:** Raw Scores for Composuite, Part1.

| Task | DT | Mamba | xLSTM [1:0] | xLSTM [7:1] |
|------|-----|--------|-------------|-------------|
| IIWA_Box_None_PickPlace | 402.74 ± 14.4 | 414.73 ± 10.49 | 424.35 ± 12.95 | 421.33 ± 11.39 |
| IIWA_Box_None_Push | 388.61 ± 35.63 | 427.0 ± 2.03 | 424.4 ± 4.63 | 427.0 ± 0.68 |
| IIWA_Box_None_Shelf | 370.3 ± 80.53 | 417.61 ± 1.44 | 417.78 ± 0.96 | 416.41 ± 1.87 |
| IIWA_Box_None_Trashcan | 329.27 ± 113.43 | 424.39 ± 1.04 | 429.54 ± 1.57 | 426.07 ± 3.98 |
| IIWA_Box_GoalWall_PickPlace | 367.68 ± 81.93 | 428.6 ± 4.11 | 428.0 ± 2.32 | 429.29 ± 1.97 |
| IIWA_Box_GoalWall_Push | 299.69 ± 77.03 | 337.81 ± 88.42 | 344.59 ± 28.19 | 318.19 ± 50.76 |
| IIWA_Box_GoalWall_Shelf | 360.92 ± 48.29 | 405.81 ± 9.82 | 408.1 ± 5.92 | 402.31 ± 3.08 |
| IIWA_Box_GoalWall_Trashcan | 376.45 ± 83.64 | 422.34 ± 3.61 | 429.15 ± 2.72 | 425.64 ± 3.88 |
| IIWA_Box_ObjectDoor_PickPlace | 389.21 ± 47.22 | 417.89 ± 0.92 | 413.82 ± 4.06 | 414.08 ± 3.83 |
| IIWA_Box_ObjectDoor_Push | 406.51 ± 0.32 | 403.59 ± 5.82 | 373.61 ± 40.95 | 397.45 ± 1.89 |
| IIWA_Box_ObjectDoor_Shelf | 329.42 ± 67.73 | 353.67 ± 56.2 | 367.47 ± 43.7 | 396.33 ± 2.67 |
| IIWA_Box_ObjectDoor_Trashcan | 325.45 ± 72.77 | 372.51 ± 41.55 | 358.72 ± 76.22 | 391.58 ± 16.76 |
| IIWA_Box_ObjectWall_PickPlace | 393.52 ± 51.47 | 425.76 ± 2.29 | 420.61 ± 2.99 | 421.61 ± 1.06 |
| IIWA_Box_ObjectWall_Push | 420.21 ± 3.5 | 412.76 ± 1.67 | 410.19 ± 1.62 | 411.5 ± 3.13 |
| IIWA_Box_ObjectWall_Shelf | 400.86 ± 3.66 | 408.22 ± 1.63 | 401.42 ± 3.93 | 396.64 ± 10.55 |
| IIWA_Box_ObjectWall_Trashcan | 414.43 ± 2.93 | 413.71 ± 3.47 | 417.11 ± 1.69 | 414.46 ± 0.8 |
| IIWA_Dumbbell_None_PickPlace | 386.95 ± 51.87 | 422.35 ± 2.94 | 421.32 ± 2.03 | 421.94 ± 1.48 |
| IIWA_Dumbbell_None_Push | 360.62 ± 90.94 | 413.39 ± 6.13 | 414.23 ± 6.04 | 393.34 ± 36.66 |
| IIWA_Dumbbell_None_Shelf | 310.45 ± 73.45 | 344.81 ± 53.72 | 380.51 ± 5.34 | 350.8 ± 52.16 |
| IIWA_Dumbbell_None_Trashcan | 386.09 ± 40.69 | 396.08 ± 0.7 | 414.03 ± 3.78 | 412.34 ± 3.36 |
| IIWA_Dumbbell_GoalWall_PickPlace | 413.6 ± 1.16 | 415.64 ± 3.28 | 410.7 ± 7.64 | 413.51 ± 1.23 |
| IIWA_Dumbbell_GoalWall_Push | 316.49 ± 38.69 | 367.45 ± 4.81 | 336.67 ± 82.13 | 371.92 ± 5.91 |
| IIWA_Dumbbell_GoalWall_Shelf | 395.63 ± 3.19 | 372.77 ± 30.32 | 376.75 ± 8.62 | 372.77 ± 4.25 |
| IIWA_Dumbbell_GoalWall_Trashcan | 379.45 ± 58.51 | 374.31 ± 55.11 | 412.22 ± 4.09 | 406.03 ± 5.03 |
| IIWA_Dumbbell_ObjectDoor_PickPlace | 358.13 ± 26.76 | 364.62 ± 40.18 | 393.83 ± 2.05 | 347.28 ± 39.81 |
| IIWA_Dumbbell_ObjectDoor_Push | 400.9 ± 8.95 | 383.81 ± 8.46 | 382.93 ± 0.7 | 364.06 ± 35.78 |
| IIWA_Dumbbell_ObjectDoor_Shelf | 369.75 ± 14.29 | 325.7 ± 30.94 | 350.7 ± 21.76 | 335.84 ± 40.36 |
| IIWA_Dumbbell_ObjectDoor_Trashcan | 393.05 ± 3.92 | 358.77 ± 36.88 | 397.23 ± 1.73 | 389.54 ± 9.14 |
| IIWA_Dumbbell_ObjectWall_PickPlace | 403.51 ± 12.08 | 407.37 ± 0.09 | 404.28 ± 1.23 | 401.15 ± 10.64 |
| IIWA_Dumbbell_ObjectWall_Push | 330.77 ± 30.29 | 296.98 ± 68.18 | 334.41 ± 22.28 | 307.4 ± 33.85 |
| IIWA_Dumbbell_ObjectWall_Shelf | 353.9 ± 29.5 | 374.39 ± 6.58 | 358.29 ± 33.75 | 358.76 ± 18.87 |
| IIWA_Dumbbell_ObjectWall_Trashcan | 394.48 ± 4.39 | 361.99 ± 39.17 | 398.06 ± 0.59 | 383.43 ± 32.4 |
| IIWA_Plate_None_PickPlace | 427.3 ± 0.59 | 424.44 ± 1.82 | 424.59 ± 2.01 | 425.99 ± 1.2 |
| IIWA_Plate_None_Push | 424.25 ± 1.13 | 419.86 ± 3.96 | 418.13 ± 3.55 | 418.42 ± 1.3 |
| IIWA_Plate_None_Shelf | 408.07 ± 0.95 | 397.02 ± 6.49 | 396.55 ± 10.03 | 394.93 ± 10.81 |
| IIWA_Plate_None_Trashcan | 419.62 ± 1.81 | 420.24 ± 0.33 | 420.37 ± 0.91 | 419.42 ± 2.61 |
| IIWA_Plate_GoalWall_PickPlace | 424.69 ± 2.67 | 423.93 ± 1.77 | 421.83 ± 1.01 | 420.13 ± 8.21 |
| IIWA_Plate_GoalWall_Push | 409.69 ± 3.55 | 397.97 ± 13.41 | 390.46 ± 14.79 | 388.89 ± 3.01 |
| IIWA_Plate_GoalWall_Shelf | 404.92 ± 0.82 | 396.09 ± 4.6 | 393.01 ± 5.77 | 401.81 ± 8.93 |
| IIWA_Plate_GoalWall_Trashcan | 420.47 ± 1.88 | 420.68 ± 2.82 | 420.29 ± 1.48 | 421.31 ± 1.93 |
| IIWA_Plate_ObjectDoor_PickPlace | 408.48 ± 1.12 | 403.23 ± 7.83 | 397.51 ± 1.65 | 401.53 ± 1.76 |
| IIWA_Plate_ObjectDoor_Push | 404.34 ± 4.45 | 395.97 ± 16.84 | 389.33 ± 7.78 | 385.77 ± 1.21 |
| IIWA_Plate_ObjectDoor_Shelf | 377.91 ± 21.42 | 373.43 ± 5.34 | 369.41 ± 4.97 | 374.16 ± 13.75 |
| IIWA_Plate_ObjectDoor_Trashcan | 400.27 ± 3.16 | 400.74 ± 0.53 | 399.28 ± 1.63 | 400.23 ± 0.63 |
| IIWA_Plate_ObjectWall_PickPlace | 417.35 ± 3.15 | 416.76 ± 6.18 | 409.31 ± 1.26 | 411.62 ± 0.97 |
| IIWA_Plate_ObjectWall_Push | 413.47 ± 3.92 | 408.16 ± 6.53 | 405.51 ± 3.71 | 405.27 ± 1.34 |
| IIWA_Plate_ObjectWall_Shelf | 393.23 ± 1.39 | 376.64 ± 12.49 | 386.41 ± 8.65 | 382.81 ± 6.78 |
| IIWA_Plate_ObjectWall_Trashcan | 410.85 ± 1.07 | 408.87 ± 3.95 | 408.98 ± 0.82 | 409.35 ± 2.6 |
| IIWA_Hollowbox_None_PickPlace | 378.13 ± 94.18 | 427.5 ± 6.93 | 428.62 ± 3.62 | 426.38 ± 3.26 |
| IIWA_Hollowbox_None_Push | 386.22 ± 36.15 | 422.49 ± 8.01 | 427.73 ± 1.97 | 426.12 ± 2.3 |
| IIWA_Hollowbox_None_Shelf | 416.65 ± 6.66 | 419.89 ± 11.03 | 418.34 ± 6.49 | 415.11 ± 0.89 |
| IIWA_Hollowbox_None_Trashcan | 424.38 ± 2.77 | 421.62 ± 1.4 | 426.9 ± 2.35 | 425.99 ± 1.81 |
| IIWA_Hollowbox_GoalWall_PickPlace | 430.17 ± 3.37 | 427.76 ± 0.48 | 427.91 ± 0.76 | 426.47 ± 1.62 |
| IIWA_Hollowbox_GoalWall_Push | 401.33 ± 3.96 | 373.0 ± 41.02 | 390.09 ± 9.46 | 394.35 ± 14.43 |
| IIWA_Hollowbox_GoalWall_Shelf | 424.55 ± 2.3 | 379.05 ± 64.32 | 423.51 ± 1.31 | 419.69 ± 3.38 |
| IIWA_Hollowbox_GoalWall_Trashcan | 425.95 ± 0.73 | 425.27 ± 0.66 | 424.8 ± 1.0 | 420.68 ± 3.33 |
| IIWA_Hollowbox_ObjectDoor_PickPlace | 276.87 ± 109.64 | 369.45 ± 57.47 | 374.76 ± 45.83 | 301.41 ± 112.33 |
| IIWA_Hollowbox_ObjectDoor_Push | 326.56 ± 109.6 | 352.22 ± 53.97 | 390.78 ± 6.35 | 324.09 ± 55.59 |
| IIWA_Hollowbox_ObjectDoor_Shelf | 339.03 ± 43.75 | 370.75 ± 8.36 | 362.72 ± 30.31 | 353.98 ± 38.19 |
| IIWA_Hollowbox_ObjectDoor_Trashcan | 395.1 ± 8.7 | 370.39 ± 35.98 | 387.21 ± 14.61 | 387.99 ± 21.95 |
| IIWA_Hollowbox_ObjectWall_PickPlace | 364.95 ± 27.07 | 355.61 ± 76.66 | 356.01 ± 8.3 | 369.47 ± 24.62 |
| IIWA_Hollowbox_ObjectWall_Push | 422.04 ± 2.08 | 414.47 ± 8.08 | 414.39 ± 5.5 | 408.53 ± 8.05 |
| IIWA_Hollowbox_ObjectWall_Shelf | 400.82 ± 2.4 | 400.31 ± 1.28 | 403.69 ± 2.06 | 401.27 ± 1.97 |
| IIWA_Hollowbox_ObjectWall_Trashcan | 415.82 ± 0.9 | 416.68 ± 0.14 | 392.79 ± 44.13 | 417.34 ± 0.77 |

**Table 14:** Raw Scores for Composuite, Part 2.

| Task | DT | Mamba | xLSTM [1:0] | xLSTM [7:1] |
|---|---|---|---|---|
| Jaco_Box_None_PickPlace | 401.38 ± 3.88 | 400.41 ± 0.63 | 399.74 ± 5.35 | 396.54 ± 4.99 |
| Jaco_Box_None_Push | 399.84 ± 3.29 | 397.79 ± 1.71 | 392.77 ± 1.12 | 397.31 ± 1.39 |
| Jaco_Box_None_Shelf | 383.53 ± 0.31 | 384.65 ± 5.31 | 385.85 ± 1.1 | 386.34 ± 3.47 |
| Jaco_Box_None_Trashcan | 374.88 ± 43.66 | 398.46 ± 2.69 | 397.66 ± 4.99 | 398.21 ± 0.91 |
| Jaco_Box_GoalWall_PickPlace | 394.75 ± 2.52 | 395.12 ± 0.38 | 392.3 ± 5.3 | 389.93 ± 3.83 |
| Jaco_Box_GoalWall_Push | 317.78 ± 67.67 | 343.43 ± 7.49 | 351.67 ± 20.65 | 336.02 ± 8.59 |
| Jaco_Box_GoalWall_Shelf | 374.62 ± 20.35 | 387.0 ± 1.42 | 387.73 ± 2.11 | 384.74 ± 1.19 |
| Jaco_Box_GoalWall_Trashcan | 374.07 ± 30.72 | 393.81 ± 0.68 | 395.49 ± 1.23 | 392.53 ± 3.46 |
| Jaco_Box_ObjectDoor_PickPlace | 396.05 ± 1.12 | 391.81 ± 4.67 | 388.37 ± 1.26 | 383.39 ± 9.07 |
| Jaco_Box_ObjectDoor_Push | 364.64 ± 38.39 | 383.07 ± 5.73 | 366.91 ± 33.04 | 387.51 ± 2.93 |
| Jaco_Box_ObjectDoor_Shelf | 373.8 ± 2.81 | 379.75 ± 1.45 | 375.38 ± 6.27 | 376.86 ± 1.37 |
| Jaco_Box_ObjectDoor_Trashcan | 388.4 ± 1.28 | 353.97 ± 52.06 | 389.38 ± 2.0 | 389.81 ± 2.89 |
| Jaco_Box_ObjectWall_PickPlace | 394.31 ± 2.66 | 385.33 ± 5.43 | 388.54 ± 7.62 | 387.82 ± 2.26 |
| Jaco_Box_ObjectWall_Push | 387.4 ± 9.34 | 384.75 ± 4.29 | 383.61 ± 7.58 | 383.32 ± 7.73 |
| Jaco_Box_ObjectWall_Shelf | 364.38 ± 2.57 | 361.28 ± 8.2 | 367.38 ± 2.04 | 369.22 ± 2.79 |
| Jaco_Box_ObjectWall_Trashcan | 385.73 ± 6.85 | 385.9 ± 1.13 | 385.34 ± 0.74 | 380.01 ± 5.08 |
| Jaco_Dumbbell_None_PickPlace | 319.87 ± 1.83 | 334.2 ± 1.93 | 376.46 ± 9.19 | 334.95 ± 68.5 |
| Jaco_Dumbbell_None_Push | 388.29 ± 1.98 | 372.13 ± 5.46 | 373.3 ± 6.88 | 369.49 ± 4.36 |
| Jaco_Dumbbell_None_Shelf | 300.81 ± 61.26 | 344.47 ± 15.49 | 361.77 ± 6.21 | 362.88 ± 8.22 |
| Jaco_Dumbbell_None_Trashcan | 369.52 ± 11.5 | 369.83 ± 13.39 | 387.28 ± 1.88 | 377.27 ± 9.7 |
| Jaco_Dumbbell_GoalWall_PickPlace | 306.12 ± 40.29 | 306.26 ± 32.85 | 349.04 ± 18.3 | 348.42 ± 37.3 |
| Jaco_Dumbbell_GoalWall_Push | 107.91 ± 29.9 | 136.11 ± 9.04 | 245.71 ± 30.15 | 188.19 ± 58.09 |
| Jaco_Dumbbell_GoalWall_Shelf | 300.97 ± 114.65 | 368.99 ± 0.5 | 363.58 ± 9.74 | 346.57 ± 27.41 |
| Jaco_Dumbbell_GoalWall_Trashcan | 321.81 ± 87.58 | 317.94 ± 23.15 | 376.09 ± 2.22 | 378.49 ± 4.52 |
| Jaco_Dumbbell_ObjectDoor_PickPlace | 382.35 ± 1.62 | 380.2 ± 5.17 | 349.1 ± 32.92 | 372.44 ± 7.6 |
| Jaco_Dumbbell_ObjectDoor_Push | 382.32 ± 1.08 | 353.42 ± 7.17 | 353.85 ± 6.83 | 338.66 ± 19.03 |
| Jaco_Dumbbell_ObjectDoor_Shelf | 312.14 ± 64.22 | 330.22 ± 47.38 | 343.51 ± 30.97 | 331.5 ± 37.18 |
| Jaco_Dumbbell_ObjectDoor_Trashcan | 371.06 ± 8.48 | 375.34 ± 4.07 | 373.78 ± 6.05 | 370.06 ± 8.94 |
| Jaco_Dumbbell_ObjectWall_PickPlace | 279.55 ± 111.58 | 314.05 ± 21.02 | 360.29 ± 15.75 | 360.38 ± 12.02 |
| Jaco_Dumbbell_ObjectWall_Push | 381.11 ± 3.7 | 351.38 ± 1.82 | 349.16 ± 2.93 | 352.64 ± 11.94 |
| Jaco_Dumbbell_ObjectWall_Shelf | 354.95 ± 1.59 | 316.33 ± 42.6 | 342.43 ± 7.94 | 332.97 ± 15.33 |
| Jaco_Dumbbell_ObjectWall_Trashcan | 367.01 ± 8.38 | 354.32 ± 22.23 | 365.47 ± 7.45 | 363.25 ± 3.18 |
| Jaco_Plate_None_PickPlace | 397.25 ± 0.77 | 389.99 ± 6.44 | 384.38 ± 5.92 | 380.69 ± 2.55 |
| Jaco_Plate_None_Push | 395.18 ± 1.01 | 390.69 ± 9.12 | 381.68 ± 6.86 | 380.2 ± 3.48 |
| Jaco_Plate_None_Shelf | 380.49 ± 0.75 | 381.62 ± 0.09 | 356.49 ± 41.25 | 380.99 ± 2.43 |
| Jaco_Plate_None_Trashcan | 391.97 ± 0.76 | 390.62 ± 0.57 | 391.2 ± 1.38 | 390.3 ± 1.83 |
| Jaco_Plate_GoalWall_PickPlace | 379.45 ± 24.14 | 378.13 ± 6.34 | 377.33 ± 11.32 | 376.12 ± 4.31 |
| Jaco_Plate_GoalWall_Push | 293.6 ± 38.38 | 319.4 ± 24.13 | 320.49 ± 24.25 | 320.5 ± 31.85 |
| Jaco_Plate_GoalWall_Shelf | 358.04 ± 22.32 | 369.8 ± 15.11 | 367.73 ± 12.97 | 362.35 ± 3.32 |
| Jaco_Plate_GoalWall_Trashcan | 383.53 ± 7.45 | 387.55 ± 1.56 | 389.51 ± 2.03 | 388.57 ± 1.98 |
| Jaco_Plate_ObjectDoor_PickPlace | 390.4 ± 1.3 | 381.92 ± 15.09 | 376.2 ± 7.51 | 380.34 ± 9.73 |
| Jaco_Plate_ObjectDoor_Push | 372.01 ± 4.07 | 366.41 ± 16.51 | 359.43 ± 10.46 | 355.71 ± 3.99 |
| Jaco_Plate_ObjectDoor_Shelf | 366.15 ± 6.61 | 357.96 ± 8.35 | 368.82 ± 4.35 | 362.39 ± 7.11 |
| Jaco_Plate_ObjectDoor_Trashcan | 382.66 ± 0.58 | 384.3 ± 0.38 | 384.0 ± 1.92 | 383.57 ± 1.1 |
| Jaco_Plate_ObjectWall_PickPlace | 390.73 ± 1.55 | 378.98 ± 6.95 | 376.76 ± 8.54 | 373.98 ± 5.41 |
| Jaco_Plate_ObjectWall_Push | 378.3 ± 4.49 | 372.47 ± 10.13 | 364.42 ± 8.12 | 360.69 ± 3.82 |
| Jaco_Plate_ObjectWall_Shelf | 364.2 ± 3.52 | 364.64 ± 3.01 | 368.33 ± 1.95 | 360.73 ± 6.42 |
| Jaco_Plate_ObjectWall_Trashcan | 374.17 ± 3.76 | 375.68 ± 1.54 | 382.5 ± 2.76 | 373.86 ± 4.91 |
| Jaco_Hollowbox_None_PickPlace | 402.23 ± 2.04 | 386.75 ± 25.35 | 396.5 ± 1.04 | 398.48 ± 3.76 |
| Jaco_Hollowbox_None_Push | 392.65 ± 9.62 | 396.56 ± 4.13 | 397.09 ± 7.5 | 396.63 ± 0.38 |
| Jaco_Hollowbox_None_Shelf | 377.5 ± 2.78 | 382.06 ± 6.3 | 384.26 ± 5.2 | 381.68 ± 4.82 |
| Jaco_Hollowbox_None_Trashcan | 394.85 ± 1.28 | 394.82 ± 3.27 | 393.68 ± 3.67 | 392.87 ± 1.71 |
| Jaco_Hollowbox_GoalWall_PickPlace | 395.2 ± 1.44 | 385.82 ± 13.41 | 378.92 ± 9.41 | 379.34 ± 7.17 |
| Jaco_Hollowbox_GoalWall_Push | 349.5 ± 34.56 | 337.43 ± 15.64 | 348.44 ± 11.76 | 340.9 ± 2.77 |
| Jaco_Hollowbox_GoalWall_Shelf | 357.89 ± 19.58 | 349.29 ± 10.1 | 344.53 ± 6.27 | 333.97 ± 12.22 |
| Jaco_Hollowbox_GoalWall_Trashcan | 385.01 ± 1.04 | 385.4 ± 1.7 | 386.58 ± 0.37 | 384.52 ± 0.05 |
| Jaco_Hollowbox_ObjectDoor_PickPlace | 335.16 ± 76.71 | 387.66 ± 8.98 | 375.68 ± 4.01 | 344.62 ± 44.5 |
| Jaco_Hollowbox_ObjectDoor_Push | 356.64 ± 41.54 | 386.82 ± 11.07 | 383.4 ± 9.21 | 385.73 ± 7.74 |
| Jaco_Hollowbox_ObjectDoor_Shelf | 371.32 ± 0.65 | 362.29 ± 13.12 | 366.72 ± 4.12 | 360.22 ± 15.51 |
| Jaco_Hollowbox_ObjectDoor_Trashcan | 358.07 ± 46.79 | 385.01 ± 1.12 | 383.6 ± 2.35 | 385.17 ± 0.42 |
| Jaco_Hollowbox_ObjectWall_PickPlace | 393.5 ± 2.63 | 377.85 ± 3.53 | 378.61 ± 8.16 | 375.96 ± 5.55 |
| Jaco_Hollowbox_ObjectWall_Push | 391.74 ± 4.74 | 382.69 ± 12.26 | 387.67 ± 9.52 | 379.01 ± 6.44 |
| Jaco_Hollowbox_ObjectWall_Shelf | 371.33 ± 3.41 | 367.26 ± 11.73 | 365.73 ± 7.59 | 356.39 ± 16.14 |
| Jaco_Hollowbox_ObjectWall_Trashcan | 382.6 ± 1.63 | 385.72 ± 2.03 | 382.62 ± 1.19 | 382.01 ± 4.22 |

**Table 15:** Raw Scores for Composuite, Part 3.

| Task | DT | Mamba | xLSTM [1:0] | xLSTM [7:1] |
|---|---|---|---|---|
| Kinova3_Box_None_PickPlace | 432.49 ± 3.69 | 432.11 ± 7.68 | 432.28 ± 3.45 | 431.06 ± 2.67 |
| Kinova3_Box_None_Push | 398.81 ± 44.71 | 416.96 ± 17.33 | 428.52 ± 1.83 | 416.41 ± 18.69 |
| Kinova3_Box_None_Shelf | 411.22 ± 3.9 | 413.65 ± 0.42 | 415.58 ± 4.21 | 411.67 ± 3.98 |
| Kinova3_Box_None_Trashcan | 378.21 ± 81.97 | 426.67 ± 2.1 | 431.01 ± 0.89 | 427.82 ± 1.12 |
| Kinova3_Box_GoalWall_PickPlace | 347.29 ± 145.33 | 430.92 ± 1.73 | 431.3 ± 2.19 | 408.26 ± 40.64 |
| Kinova3_Box_GoalWall_Push | 325.78 ± 131.68 | 390.05 ± 6.59 | 382.78 ± 2.17 | 388.29 ± 6.07 |
| Kinova3_Box_GoalWall_Shelf | 357.79 ± 96.22 | 395.77 ± 28.11 | 418.95 ± 2.7 | 417.37 ± 1.02 |
| Kinova3_Box_GoalWall_Trashcan | 373.8 ± 80.27 | 424.09 ± 0.02 | 428.12 ± 3.66 | 427.05 ± 0.87 |
| Kinova3_Box_ObjectDoor_PickPlace | 425.72 ± 1.7 | 427.38 ± 0.43 | 424.25 ± 2.86 | 424.5 ± 3.45 |
| Kinova3_Box_ObjectDoor_Push | 395.44 ± 30.77 | 414.0 ± 5.47 | 406.02 ± 0.61 | 410.58 ± 8.15 |
| Kinova3_Box_ObjectDoor_Shelf | 381.62 ± 37.98 | 326.93 ± 2.6 | 408.55 ± 2.3 | 381.75 ± 45.62 |
| Kinova3_Box_ObjectDoor_Trashcan | 392.17 ± 40.87 | 415.87 ± 2.48 | 419.24 ± 0.61 | 416.46 ± 1.78 |
| Kinova3_Box_ObjectWall_PickPlace | 405.45 ± 21.25 | 387.27 ± 50.08 | 425.83 ± 2.68 | 423.06 ± 3.66 |
| Kinova3_Box_ObjectWall_Push | 419.98 ± 2.8 | 414.6 ± 1.04 | 412.82 ± 1.07 | 415.16 ± 7.28 |
| Kinova3_Box_ObjectWall_Shelf | 399.47 ± 4.56 | 399.51 ± 1.29 | 402.37 ± 2.66 | 402.42 ± 1.48 |
| Kinova3_Box_ObjectWall_Trashcan | 416.15 ± 4.57 | 412.41 ± 0.4 | 399.87 ± 31.99 | 394.97 ± 36.15 |
| Kinova3_Dumbbell_None_PickPlace | 380.36 ± 55.46 | 418.88 ± 5.8 | 419.3 ± 7.37 | 416.89 ± 2.86 |
| Kinova3_Dumbbell_None_Push | 394.84 ± 25.64 | 396.29 ± 13.63 | 367.03 ± 53.29 | 390.74 ± 22.17 |
| Kinova3_Dumbbell_None_Shelf | 290.98 ± 123.89 | 394.73 ± 4.82 | 386.09 ± 19.99 | 397.38 ± 2.93 |
| Kinova3_Dumbbell_None_Trashcan | 358.26 ± 43.32 | 377.36 ± 53.06 | 413.01 ± 6.02 | 414.39 ± 1.97 |
| Kinova3_Dumbbell_GoalWall_PickPlace | 408.52 ± 19.13 | 392.63 ± 23.38 | 404.51 ± 4.31 | 412.68 ± 11.05 |
| Kinova3_Dumbbell_GoalWall_Push | 294.63 ± 35.99 | 358.66 ± 10.09 | 321.72 ± 41.37 | 310.79 ± 67.84 |
| Kinova3_Dumbbell_GoalWall_Shelf | 384.01 ± 20.53 | 383.06 ± 15.17 | 395.02 ± 0.83 | 377.15 ± 28.52 |
| Kinova3_Dumbbell_GoalWall_Trashcan | 377.28 ± 51.33 | 370.59 ± 31.83 | 413.63 ± 2.06 | 378.76 ± 27.34 |
| Kinova3_Dumbbell_ObjectDoor_PickPlace | 415.58 ± 5.38 | 404.89 ± 11.83 | 405.77 ± 7.4 | 410.95 ± 8.75 |
| Kinova3_Dumbbell_ObjectDoor_Push | 359.17 ± 15.53 | 265.44 ± 62.94 | 367.39 ± 23.91 | 311.57 ± 45.56 |
| Kinova3_Dumbbell_ObjectDoor_Shelf | 360.34 ± 28.19 | 379.36 ± 6.7 | 385.26 ± 2.74 | 363.99 ± 37.65 |
| Kinova3_Dumbbell_ObjectDoor_Trashcan | 409.92 ± 1.78 | 407.09 ± 1.26 | 407.79 ± 0.71 | 407.57 ± 2.85 |
| Kinova3_Dumbbell_ObjectWall_PickPlace | 404.63 ± 16.95 | 409.29 ± 4.6 | 406.14 ± 2.11 | 411.69 ± 6.71 |
| Kinova3_Dumbbell_ObjectWall_Push | 311.79 ± 94.94 | 285.81 ± 62.32 | 342.04 ± 22.98 | 244.56 ± 16.32 |
| Kinova3_Dumbbell_ObjectWall_Shelf | 378.68 ± 3.03 | 378.63 ± 0.91 | 376.92 ± 0.76 | 361.79 ± 25.06 |
| Kinova3_Dumbbell_ObjectWall_Trashcan | 400.98 ± 4.19 | 398.65 ± 3.89 | 401.96 ± 1.45 | 395.81 ± 3.51 |
| Kinova3_Plate_None_PickPlace | 424.09 ± 4.78 | 427.36 ± 4.29 | 424.82 ± 1.31 | 425.02 ± 2.92 |
| Kinova3_Plate_None_Push | 412.25 ± 19.8 | 422.75 ± 2.79 | 417.63 ± 6.13 | 416.41 ± 4.33 |
| Kinova3_Plate_None_Shelf | 409.96 ± 0.2 | 409.11 ± 0.52 | 410.28 ± 0.65 | 409.52 ± 1.61 |
| Kinova3_Plate_None_Trashcan | 422.54 ± 2.13 | 422.07 ± 1.15 | 421.73 ± 1.36 | 422.97 ± 0.74 |
| Kinova3_Plate_GoalWall_PickPlace | 427.74 ± 0.81 | 421.23 ± 6.67 | 416.44 ± 1.6 | 416.35 ± 15.86 |
| Kinova3_Plate_GoalWall_Push | 401.46 ± 2.17 | 385.01 ± 15.39 | 377.6 ± 3.14 | 386.87 ± 12.31 |
| Kinova3_Plate_GoalWall_Shelf | 410.49 ± 0.77 | 409.46 ± 0.15 | 409.63 ± 0.65 | 407.67 ± 3.33 |
| Kinova3_Plate_GoalWall_Trashcan | 421.05 ± 0.88 | 421.19 ± 0.48 | 422.63 ± 0.81 | 423.21 ± 1.16 |
| Kinova3_Plate_ObjectDoor_PickPlace | 423.26 ± 0.3 | 407.55 ± 0.81 | 406.43 ± 2.07 | 414.11 ± 7.32 |
| Kinova3_Plate_ObjectDoor_Push | 258.58 ± 18.57 | 278.08 ± 34.02 | 300.72 ± 90.5 | 257.79 ± 48.13 |
| Kinova3_Plate_ObjectDoor_Shelf | 404.4 ± 0.95 | 403.82 ± 0.86 | 405.9 ± 0.31 | 401.09 ± 2.61 |
| Kinova3_Plate_ObjectDoor_Trashcan | 415.34 ± 1.08 | 415.81 ± 0.35 | 416.09 ± 0.31 | 414.34 ± 1.85 |
| Kinova3_Plate_ObjectWall_PickPlace | 420.16 ± 2.07 | 413.68 ± 5.5 | 408.0 ± 2.29 | 411.83 ± 4.11 |
| Kinova3_Plate_ObjectWall_Push | 400.11 ± 16.39 | 403.95 ± 3.67 | 406.48 ± 5.73 | 403.65 ± 6.23 |
| Kinova3_Plate_ObjectWall_Shelf | 391.09 ± 3.65 | 391.99 ± 6.62 | 386.25 ± 16.53 | 391.7 ± 5.14 |
| Kinova3_Plate_ObjectWall_Trashcan | 413.36 ± 1.11 | 413.44 ± 3.93 | 413.82 ± 2.45 | 415.14 ± 1.46 |
| Kinova3_Hollowbox_None_PickPlace | 424.86 ± 6.23 | 433.78 ± 0.13 | 430.43 ± 1.11 | 430.84 ± 1.55 |
| Kinova3_Hollowbox_None_Push | 361.99 ± 40.33 | 369.17 ± 8.0 | 396.28 ± 28.04 | 380.94 ± 28.74 |
| Kinova3_Hollowbox_None_Shelf | 417.73 ± 13.43 | 417.46 ± 0.36 | 423.26 ± 3.53 | 424.02 ± 2.62 |
| Kinova3_Hollowbox_None_Trashcan | 424.65 ± 1.15 | 409.34 ± 12.4 | 425.0 ± 2.72 | 416.0 ± 15.33 |
| Kinova3_Hollowbox_GoalWall_PickPlace | 386.68 ± 49.29 | 425.24 ± 0.83 | 421.85 ± 8.69 | 420.32 ± 9.71 |
| Kinova3_Hollowbox_GoalWall_Push | 403.57 ± 0.96 | 383.09 ± 8.37 | 384.13 ± 10.01 | 381.43 ± 8.58 |
| Kinova3_Hollowbox_GoalWall_Shelf | 385.7 ± 36.06 | 395.01 ± 4.51 | 423.93 ± 5.1 | 417.05 ± 13.43 |
| Kinova3_Hollowbox_GoalWall_Trashcan | 406.37 ± 27.44 | 404.11 ± 3.64 | 405.09 ± 22.54 | 389.36 ± 32.05 |
| Kinova3_Hollowbox_ObjectDoor_PickPlace | 344.01 ± 63.38 | 364.3 ± 13.82 | 387.53 ± 20.66 | 324.36 ± 55.48 |
| Kinova3_Hollowbox_ObjectDoor_Push | 390.98 ± 46.38 | 416.05 ± 8.96 | 405.41 ± 5.34 | 406.76 ± 16.92 |
| Kinova3_Hollowbox_ObjectDoor_Shelf | 359.0 ± 25.63 | 381.87 ± 12.39 | 390.42 ± 6.21 | 357.94 ± 48.51 |
| Kinova3_Hollowbox_ObjectDoor_Trashcan | 405.87 ± 4.17 | 411.24 ± 1.26 | 414.92 ± 3.6 | 408.73 ± 5.66 |
| Kinova3_Hollowbox_ObjectWall_PickPlace | 424.57 ± 0.92 | 408.98 ± 6.4 | 417.83 ± 5.67 | 419.63 ± 9.2 |
| Kinova3_Hollowbox_ObjectWall_Push | 249.37 ± 176.18 | 319.13 ± 111.09 | 324.39 ± 76.09 | 335.61 ± 74.98 |
| Kinova3_Hollowbox_ObjectWall_Shelf | 394.7 ± 9.3 | 328.52 ± 61.08 | 357.89 ± 37.75 | 362.16 ± 40.05 |
| Kinova3_Hollowbox_ObjectWall_Trashcan | 354.65 ± 48.89 | 353.43 ± 78.59 | 407.99 ± 1.96 | 408.29 ± 4.94 |

**Table 16:** Raw Scores for Composuite, Part 4.

| Task | DT | Mamba | xLSTM [1:0] | xLSTM [7:1] |
|---|---|---|---|---|
| Panda_Box_None_PickPlace | 409.21 ± 5.27 | 408.66 ± 7.81 | 409.83 ± 1.87 | 405.46 ± 3.84 |
| Panda_Box_None_Push | 402.52 ± 2.55 | 373.74 ± 49.95 | 400.35 ± 2.32 | 399.37 ± 9.95 |
| Panda_Box_None_Shelf | 383.69 ± 4.34 | 381.42 ± 3.66 | 383.55 ± 5.74 | 386.01 ± 1.29 |
| Panda_Box_None_Trashcan | 400.37 ± 5.64 | 395.77 ± 2.77 | 407.95 ± 1.92 | 406.17 ± 3.36 |
| Panda_Box_GoalWall_PickPlace | 401.53 ± 6.39 | 389.57 ± 18.4 | 397.12 ± 4.39 | 401.64 ± 9.81 |
| Panda_Box_GoalWall_Push | 272.61 ± 79.58 | 257.61 ± 57.4 | 263.72 ± 45.71 | 281.71 ± 31.21 |
| Panda_Box_GoalWall_Shelf | 384.43 ± 1.66 | 389.06 ± 3.69 | 388.59 ± 3.9 | 383.94 ± 2.0 |
| Panda_Box_GoalWall_Trashcan | 400.68 ± 4.51 | 400.18 ± 6.03 | 403.24 ± 5.65 | 392.28 ± 16.82 |
| Panda_Box_ObjectDoor_PickPlace | 359.01 ± 12.2 | 365.3 ± 5.97 | 359.63 ± 0.79 | 359.27 ± 10.88 |
| Panda_Box_ObjectDoor_Push | 363.07 ± 3.13 | 352.85 ± 13.71 | 340.37 ± 6.06 | 340.5 ± 4.97 |
| Panda_Box_ObjectDoor_Shelf | 346.29 ± 2.53 | 345.8 ± 4.91 | 349.82 ± 6.46 | 341.44 ± 11.05 |
| Panda_Box_ObjectDoor_Trashcan | 361.19 ± 1.65 | 356.77 ± 3.24 | 356.66 ± 5.73 | 337.69 ± 32.63 |
| Panda_Dumbbell_None_PickPlace | 342.62 ± 39.18 | 310.15 ± 24.64 | 318.76 ± 2.7 | 342.02 ± 31.28 |
| Panda_Dumbbell_None_Push | 299.34 ± 78.28 | 341.64 ± 42.57 | 359.06 ± 42.88 | 263.35 ± 154.81 |
| Panda_Dumbbell_None_Shelf | 264.01 ± 101.29 | 362.15 ± 0.87 | 319.71 ± 33.9 | 297.54 ± 67.67 |
| Panda_Dumbbell_None_Trashcan | 174.45 ± 64.43 | 329.06 ± 43.08 | 373.77 ± 16.73 | 327.93 ± 68.84 |
| Panda_Dumbbell_GoalWall_PickPlace | 310.61 ± 42.65 | 268.34 ± 147.91 | 329.02 ± 62.28 | 360.39 ± 5.25 |
| Panda_Dumbbell_GoalWall_Push | 249.21 ± 43.29 | 282.01 ± 4.89 | 270.81 ± 11.98 | 285.28 ± 5.25 |
| Panda_Dumbbell_GoalWall_Shelf | 319.5 ± 68.89 | 347.34 ± 20.01 | 364.15 ± 2.6 | 318.6 ± 33.85 |
| Panda_Dumbbell_GoalWall_Trashcan | 377.5 ± 5.27 | 360.98 ± 9.73 | 379.05 ± 7.52 | 337.19 ± 40.73 |
| Panda_Dumbbell_ObjectDoor_PickPlace | 344.54 ± 5.77 | 346.57 ± 0.33 | 340.15 ± 8.5 | 338.46 ± 10.42 |
| Panda_Dumbbell_ObjectDoor_Push | 289.31 ± 11.14 | 308.25 ± 9.24 | 309.4 ± 5.02 | 304.1 ± 8.06 |
| Panda_Dumbbell_ObjectDoor_Shelf | 323.26 ± 3.52 | 279.85 ± 18.84 | 313.19 ± 17.79 | 323.49 ± 0.27 |
| Panda_Dumbbell_ObjectDoor_Trashcan | 334.05 ± 5.55 | 337.49 ± 0.68 | 341.0 ± 3.14 | 333.06 ± 7.77 |
| Panda_Plate_None_PickPlace | 384.37 ± 30.37 | 404.77 ± 5.27 | 397.34 ± 1.3 | 398.41 ± 2.51 |
| Panda_Plate_None_Push | 397.95 ± 1.05 | 398.1 ± 4.91 | 397.42 ± 3.32 | 397.64 ± 2.7 |
| Panda_Plate_None_Shelf | 352.29 ± 37.8 | 372.12 ± 13.92 | 370.46 ± 3.11 | 367.5 ± 6.03 |
| Panda_Plate_None_Trashcan | 392.99 ± 1.41 | 393.63 ± 2.91 | 394.05 ± 3.74 | 393.71 ± 1.27 |
| Panda_Plate_GoalWall_PickPlace | 398.36 ± 3.95 | 398.24 ± 4.51 | 393.0 ± 1.9 | 399.02 ± 4.53 |
| Panda_Plate_GoalWall_Push | 387.68 ± 0.49 | 377.79 ± 11.92 | 355.01 ± 34.01 | 350.1 ± 22.72 |
| Panda_Plate_GoalWall_Shelf | 380.05 ± 0.52 | 367.67 ± 22.6 | 339.46 ± 40.63 | 359.76 ± 5.67 |
| Panda_Plate_GoalWall_Trashcan | 391.41 ± 3.83 | 389.44 ± 3.8 | 395.4 ± 2.49 | 393.96 ± 2.68 |
| Panda_Plate_ObjectDoor_PickPlace | 350.33 ± 18.2 | 348.67 ± 8.14 | 329.35 ± 4.62 | 336.64 ± 16.61 |
| Panda_Plate_ObjectDoor_Push | 346.4 ± 9.33 | 337.36 ± 17.06 | 326.32 ± 7.92 | 323.51 ± 2.24 |
| Panda_Plate_ObjectDoor_Shelf | 290.68 ± 11.21 | 321.54 ± 17.89 | 326.04 ± 18.76 | 305.25 ± 20.96 |
| Panda_Plate_ObjectDoor_Trashcan | 348.09 ± 3.63 | 349.43 ± 4.05 | 351.8 ± 0.25 | 349.29 ± 1.91 |
| Panda_Hollowbox_None_PickPlace | 410.32 ± 6.76 | 412.25 ± 3.0 | 408.01 ± 1.93 | 405.29 ± 5.3 |
| Panda_Hollowbox_None_Push | 404.95 ± 1.07 | 406.74 ± 4.03 | 401.61 ± 6.16 | 402.46 ± 4.04 |
| Panda_Hollowbox_None_Shelf | 387.59 ± 5.19 | 380.86 ± 10.45 | 369.22 ± 14.85 | 369.57 ± 4.84 |
| Panda_Hollowbox_None_Trashcan | 399.09 ± 2.01 | 400.52 ± 5.27 | 401.03 ± 5.27 | 392.82 ± 7.37 |
| Panda_Hollowbox_GoalWall_PickPlace | 406.02 ± 10.18 | 403.47 ± 0.97 | 405.96 ± 0.39 | 407.16 ± 3.77 |
| Panda_Hollowbox_GoalWall_Push | 259.87 ± 75.12 | 293.02 ± 117.06 | 341.55 ± 23.29 | 281.79 ± 42.98 |
| Panda_Hollowbox_GoalWall_Shelf | 387.38 ± 3.45 | 369.01 ± 6.14 | 365.26 ± 6.74 | 316.46 ± 81.46 |
| Panda_Hollowbox_GoalWall_Trashcan | 377.54 ± 44.77 | 395.3 ± 4.85 | 396.82 ± 4.17 | 401.54 ± 5.21 |
| Panda_Hollowbox_ObjectDoor_PickPlace | 334.94 ± 35.48 | 341.18 ± 32.31 | 342.71 ± 7.54 | 353.64 ± 2.45 |
| Panda_Hollowbox_ObjectDoor_Push | 192.69 ± 6.49 | 294.01 ± 57.68 | 257.48 ± 13.16 | 230.54 ± 8.56 |
| Panda_Hollowbox_ObjectDoor_Shelf | 343.92 ± 10.22 | 202.17 ± 4.87 | 328.01 ± 42.52 | 285.35 ± 64.92 |
| Panda_Hollowbox_ObjectDoor_Trashcan | 338.02 ± 36.48 | 363.04 ± 2.59 | 360.88 ± 2.45 | 363.04 ± 1.29 |

