# OpenReview forum: "A Large Recurrent Action Model: xLSTM enables Fast Inference for Robotics Tasks"
_ICLR.cc/2025/Conference — Submitted to ICLR 2025_

### Official Review · Reviewer_uzTK · 2024-10-29

**Soundness:** 2
**Presentation:** 3
**Contribution:** 3
**Rating:** 6
**Confidence:** 4

**Summary:**

This paper investigates recurrent architectures for large action models that enable efficient inference. It proposes the Large Recurrent Action Model (LRAM), featuring an xLSTM core, which achieves linear-time inference complexity and supports natural extrapolation to longer sequence lengths. Experiments conducted on 432 tasks across six domains demonstrate that LRAM performs competitively with Transformers, offering advantages in both speed and performance.

**Strengths:**

1. The paper is well-written and easy-to-read.
2. The study compiles 894 million transitions and conducts experiments across 432 tasks in six domains, demonstrating a substantial research effort.

**Weaknesses:**

1. There is a lack of analysis regarding certain experimental observations. Specifically, an explanation for the lower performance of the Decision Transformer (DT) compared to recurrent-based agents would be valuable, especially since DT has more parameters, which, according to scaling laws, would be expected to yield higher performance.
2. Additionally, I am interested in understanding why xLSTM outperforms Mamba, as both architectures are designed for linear-time inference complexity. In Line 300, what is meant by "This is an important advantage of xLSTM, as LRAM agents can strongly benefit from more data and consequently larger models"?
3. It would enhance clarity to make key findings more prominent, as in DeMa [1], which also investigates the benefits of integrating Mamba within sequence modeling frameworks.

[1] Yang Dai, et al. Is Mamba Compatible with Trajectory Optimization in Offline Reinforcement Learning? 2024

**Questions:**

1. Why is only xLSTM mentioned as the core of LRAM in the abstract and introduction? The authors also indicate that both xLSTM and Mamba provide parallelization benefits during training similar to the Transformer architecture while enabling faster inference.
2. How is the shared action head used across all environments, given that each environment may have a different output dimension and that continuous actions require discretization?
3. In Figure 2(b), should the performance of different environments be weighted according to the number of tasks in each environment to prevent any single environment from disproportionately influencing the results? For example, only in ProcGen, xLSTM [1:0] appears to show a distinct advantage.
4. Why did the authors choose to place the results of fine-tuning in Section 4.3 in the appendix, considering that there is still space to present these results in the main paper?
5. I am interested in the training details for multi-task training. With 432 tasks, could the learning curve be affected by conflicting gradients inherent to multi-task learning?
6. In Section 4.4, the authors investigate inference time within a simulation environment. However, during inference, it is common to set $B=1$, where the latency of DT and xLSTM is comparable. A notable difference in inference time only appears when $B=16$. Given this, is the use of xLSTM truly necessary?

---

> ### Author Response · Authors · 2024-11-21
>
> Thank you for your feedback, which helped us to considerably improve our paper. We are happy that you find that our work demonstrates substantial research effort.
>
> **Lower performance of DT?** We want to clarify that DT and xLSTM have the same number of parameters in all our experiments to ensure a fair comparison. Only Mamba has a slightly higher number of parameters (see x-axis in Figure 2). With respect to the scaling laws, xLSTM results in slightly better performance than DT, but also comes with linear inference time. A similar performance advantage has been observed in the original xLSTM publication [1].
>
> **Why does xLSTM outperform Mamba?** It is true that both xLSTM and Mamba are designed for linear-time inference complexity. Line 300 tries to say that we observe performance differences at larger model scales such as 206M (see Figure 2b, last point), which indicates that xLSTM (and also other backbones) may continue to benefit from more parameters and more data. To provide preliminary empirical evidence for this, we present additional model performances at the 408M model scale in Figure 14 (only 1 seed for computational reasons), and find that performance continues to improve slightly.
> One other benefit of xLSTM over Mamba may be the state tracking abilities of sLSTM blocks [2], which can be useful for partially observable observation spaces such as Darkoom. Therefore, we conduct an additional ablation to study the effect of the ratio of mLSTM-to-sLSTM blocks in Appendix D.3. However, we believe that more research is necessary to fully investigate the difference between modern recurrent architectures, and hope our work is a step in this direction.
>
> **Discussion on DeMA [3]:** Thank you for bringing up DeMa [3]. We added the paper to our related work section.

---

> > ### Author Response · Authors · 2024-11-21
> >
> > Regarding your **questions**:
> > 1. It is true that both xLSTM and Mamba provide parallelization benefits during training similar to the Transformer architecture while enabling faster inference. We found that xLSTM gives slightly better final performance (see Figure 2), which is why we focus on this backbone in the introduction.
> > 2. We make use of discretization of continuous actions. Every action dimension is discretized into 256 uniformly spaced bins. The shared action head is used to predict the action bins of all continuous dimensions jointly. While environments may have different action dimensions, the model predicts all action dimensions jointly. At inference time, the number of action dimensions of the current environment is known, and we extract the respective dimensions from the joint predictions. We describe this procedure in more detail in Appendix B.3.
> > 3. For our experiments on the 432 tasks, we decided to report the average across the domain performances, instead of reporting the average across all tasks. Otherwise, the final performance would be heavily influenced by the domain that contains the highest number of tasks (Composuite, 256 tasks). While both ways of reporting the average performance are valid, we focused on the former, because we aim to study how the backbones perform across different domains. For reference, we also added the raw performances scores at the 206M model scale to Appendix F.
> > 4. It is true that ICLR permits submissions with up to 10 pages, but the recommended page limit is 9 full pages. Therefore, we aimed for the 9-page limit and opted to describe the fine-tuning results in the main text (see Section 4.3) and to put the plots in the appendix.
> > 5. For experiment details, we refer the reviewer to Appendix B of our updated manuscript. We agree with the reviewer that studying the interference between the 432 tasks could be interesting. We believe that it is likely that some interference exists, as has been found in prior works on multi-task models [4,5]. While we believe that a thorough investigation on interference is out-of-scope for this work, studying this phenomenon in large action models could be very interesting for future work. We hope that our data processing pipeline and datasets may facilitate such future studies.
> > 6. We agree with the reviewer that B=1 is common. For B=1, the recurrent backbone is almost twice as fast as the Transformer, even when comparing against the highly optimized FlashAttention (Figure 6). Importantly, the inference speed and memory consumption of xLSTM does not change with the sequence lengths. This is particularly apparent in the memory footprint (Figure 7). Note that we conduct this comparison on a high-end data-center GPU with 40GB of RAM. In contrast, applications on the edge, like robotics, may have to deal with less powerful accelerators. Therefore, the recurrent backbone enables processing longer sequences without increasing requirements in terms of speed and memory. This is important because this property may enable handling high sample rates as prevalent in real-world applications, or multi-episodic context as used for in-context RL (see Lines 72-78). To highlight the benefits of longer context, we conduct additional ablation studies in Appendix D.1 of our updated manuscript.
> >
> > We revised our manuscript, and believe it has become considerably better because of your suggestions. In case any questions remain, we would be happy to answer them!
> >
> > [1] “xLSTM: Extended Long Short-Term Memory”, NeurIPS 2024 \
> > [2]“The Illusion of State in State-Space Models”, ICML 2024 \
> > [3] “Yang Dai, et al. Is Mamba Compatible with Trajectory Optimization in Offline Reinforcement Learning?” NeurIPS 2024 \
> > [4] “Gradient Surgery for Multi-Task Learning”, NeurIPS 2020 \
> > [5] “Ray Interference: a Source of Plateaus in Deep Reinforcement Learning”, ArXiv 2019

---

> ### Comment · Reviewer_uzTK · 2024-11-22
>
> Thank you for the rebuttal and the inclusion of detailed experimental results. I appreciate the large-scale experimental settings and the analysis of the xLSTM architectures within the RL domain. I hope the authors will consider releasing the collected datasets and corresponding codes soon. I have raised my score, good luck!

---

> > ### Author Response · Authors · 2024-11-29
> >
> > Thank you for your additional time and feedback! We are very glad that you like our experiments/analyses and appreciate that you raised your score.

---

### Official Review · Reviewer_C3KT · 2024-11-01

**Soundness:** 3
**Presentation:** 3
**Contribution:** 3
**Rating:** 6
**Confidence:** 3

**Summary:**

This paper explores the application of a recurrent model (xLSTM) in robotic policy learning, introducing a novel approach called the Large Recurrent Action Model (LRAM), which enhances inference efficiency. In comparison to conventional sequential models used in Offline RL, such as the Decision Transformer, LRAM demonstrates superior performance on tasks within the Procgen and DMControl environments, while achieving competitive results on other benchmarks like Meta-World, Atari, MimicGen, and Composuite. Notably, LRAM benefits from significantly higher throughput, presenting a distinct advantage. Comprehensive experiments and ablation studies highlight the effectiveness of LRAM, incorporating xLSTM or Mamba as its core, showing favorable evaluation performance across various model scales when compared to Transformer-based approaches.

**Strengths:**

1. The paper is well-written and organized, making it easy to follow.

2. The reviewer appreciates the solid and robust experiments across diverse benchmarks, including thorough ablation studies on context length and action token.

3. Analysis provided in Section 4.3 makes the paper an insightful and valuable study, which may benefit the community.

4. The inclusion of detailed explanations for dataset preparation and training processes is appreciated.

**Weaknesses:**

The reviewer believes the following aspects can be further improved:

1. The authors are encouraged to discuss the key differences between xLSTM, Mamba, and a typical transformer (such as GPT-2) used in the Decision Transformer.

2. It would be better if the authors could present the normalized performance of the offline datasets (ground truth) presented in Figure 2b.

**Minor**

1. The bottom border of Table 5 is missing.

2. In Lines 1282-1283, the scientific notation should be written as $1e-4$ instead of $1e^{-4}$
.

**Questions:**

1. How does the ratio of mLSTM to sLSTM blocks impact performance? Could you provide further observations on specific configurations, such as xLSTM[3:1]?

2. As a follow-up, what would happen if Transformer layers in DT were gradually replaced with sLSTM or mLSTM layers?

3. Could you elaborate on the visual encoder referenced in Lines 186-187? Specifically, is it jointly trained with the downstream policy, and how is it initialized?

---

> ### Author Response · Authors · 2024-11-21
>
> Thank you for your constructive feedback. We appreciate your positive assessment and are glad that you find our work insightful and a valuable study.
>
> **Differences between xLSTM/Mamba/Transformer:** We agree with the reviewer that a discussion on the differences between xLSTM, Mamba and Transformers can be useful and added a discussion to the paper. Important differences include:
> - The complexity difference between Transformers and recurrent backbones such as xLSTM and Mamba. While the self-attention in Transformers is quadratic, the recurrent backbones are linear. This can have advantages for inference speeds and memory requirements (see Figures 6, 7), in particular for long sequences. Note that we conduct our comparison on a high-end data-center GPU with 40GB of RAM. In contrast, applications on the edge, like robotics, may have to deal with less powerful accelerators, making linear-time inference more attractive.
> - Transformers have other advantages. For example, for tasks where exact recall is required, self-attention is typically superior (see Figure 5 in [1]), which can be important for decision-making tasks [2]. xLSTM, in particular, can be better for tasks that require state-tracking [3], which Mamba and Transformers cannot model.
>
> **Normalized vs. Raw scores:** Throughout the paper, we presented the normalized performances (e.g., in Figure 2b). This is to make the scores from different domains more comparable. Did the reviewer mean that we should also add tables for the raw-scores to the paper? If so, we added the raw performance scores for all 432 tasks for the 206M model scale to Appendix F of our manuscript.
>
> Thank you for spotting our mistakes on the bottom border of Table 5 and the scientific notation of the learning rate. We corrected them!
>
> Regarding your questions:
> 1. **Ratio of mLSTM to sLSTM blocks:** We agree with the reviewer that other ratios would be interesting to study. The rationale for the ratios we chose is that we adopted the ratios of the original xLSTM publication [1] (see Appendix B.3). To get a better understanding of the effects of this design choice, we conduct additional ablation studies on the ratio in Appendix D.3. We compare different ratios (1:0, 0:1, 1:1, 3:1, 7:1) and different placements of the sLSTM blocks, both on the 432 tasks used in the main paper, and on the Darkroom environment. While we do not find benefits on the 432 tasks at the 16M model scale (see Figure 29), we do find that different ratios (such as 3:1) can be beneficial on Darkoom (see Figure 30). Therefore, the effectiveness of sLSTM layers depends on the task at hand. For more complex tasks with long horizons or partial observability (as is common in real-world applications), sLSTM may be a useful tool for practitioners.
> 2. **Replacing Transformer blocks with xLSTM blocks:** We agree with the reviewer that combinations of Transformer and xLSTM blocks may be interesting and effective. While we cannot provide empirical evidence for this combination, it is possible that they may lead to performance improvements, by combining the benefits of both architectures. Similarly, combinations of SSM and Transformer layers have been found to be effective in Griffin [4] and Jamba [5]. However, the combination comes at the cost of losing the linear inference time benefit.
> 3. **Visual encoder:** The visual encoder is trained jointly with the downstream policy. We use a CNN architecture similar to [6], and do not make use of ViT-style patchification of images (as used in [7,8]). Consequently, every image is a single embedded “token”. In preliminary experiments, we found that patchification led to similar results as using a separate image encoder, but comes at the cost of increased sequence length. The image encoder is randomly initialized, but initializing from a pre-trained model and fine-tuning may also be effective [9]. We describe the image encoder in Appendix B.3.
>
> Once again, thank you for your helpful comments and positive assessment of our work.
>
> [1] “xLSTM: Extended Long Short-Term Memory”, NeurIPS 2024 \
> [2] “When Do Transformers Shine in RL? Decoupling Memory from Credit Assignment”, NeurIPS 2023 \
> [3] “The Illusion of State in State-Space Models”, ICML 2024 \
> [4] “Griffin: Mixing Gated Linear Recurrences with Local Attention for Efficient Language Models”, ArXiv 2024 \
> [5] “Jamba: A Hybrid Transformer-Mamba Language Model”, ArXiv 2024 \
> [6] “IMPALA: Scalable Distributed Deep-RL with Importance Weighted Actor-Learner Architectures”, ICML 2018 \
> [7] “Multi-Game Decision Transformers”, NeurIPS 2022 \
> [8] “A Generalist Agent”, TMLR 2022 \
> [9] “RT-1: Robotics Transformer for Real-World Control at Scale”, ArXiv 2023

---

> > ### Comment · Reviewer_C3KT · 2024-11-24
> >
> > I appreciate the authors' response and the additional clarifications provided. It would be beneficial if these clarifications could be incorporated into the revised version of the paper, if they are not already included.
> >
> > To clarify my earlier comments on the "normalized performance of the offline datasets (ground truth)," my main interest lies in evaluating the performance of the expert trajectories that the model is trained on in this behavior cloning setting. Specifically, I would like to understand how well the learned expert policy performs—essentially, how close the learned policy is to the original expert policy. Providing reference numbers for these comparisons would offer valuable insight.
> >
> > Regarding the term "normalized," I understand that normalization is used to summarize performance across environments. My suggestion is to apply the same normalization method to the performance of the expert trajectories so that the results are directly comparable. I apologize for any confusion caused by my earlier wording.

---

> > > ### Author Response · Authors · 2024-11-24
> > >
> > > We thank the reviewer for their additional time and comments. We agree that these clarifications are beneficial to the paper. We have already included:
> > > - the discussion on the differences between xLSTM/Mamba/Transformer
> > > - the raw scores for all 432 tasks for the 206M model scale
> > > - the ablation on the ration between mLSTM and sLSTM blocks
> > > - the discussion on the visual encoder.
> > >
> > > We will include the discussion on replacing the Transformer blocks with xLSTM blocks in our next revision.
> > >
> > > Thank you for the clarification on the “normalized performance of the offline datasets (ground truth)". We now understand your point, and we agree that these numbers are important. To clarify, we report data-normalized scores for Meta-World, DMControl, Procgen, Composuite and Mimicgen (as proposed by [1]), and human-normalized scores for Atari (see Section 4.2, “Performance per domain”). The normalized scores are obtained by normalizing using the performances achieved by a random agent and an expert agent on the task at hand. For the expert score, the mean return in the dataset is used. Therefore, the reported scores directly indicate how close the learned policy is to the expert. The values we use for normalization (random and expert) are available in the accompanying source code. For our next revision, we will include them in our manuscript.
> > >
> > > We hope this clarifies your comments.
> > >
> > > [1] “D4RL: Datasets for Deep Data-Driven Reinforcement Learning”, ArXiv 2020

---

> > > > ### Author Response · Authors · 2024-11-29
> > > >
> > > > Does the reviewer have any remaining open questions? If yes, we would be very happy to engage in further discussion. If not, we would highly appreciate it if the reviewer would consider raising their score in light of our responses and the new version of our manuscript.

---

### Official Review · Reviewer_Vb2E · 2024-11-04

**Soundness:** 3
**Presentation:** 3
**Contribution:** 2
**Rating:** 6
**Confidence:** 4

**Summary:**

This paper conducts a comprehensive empirical study on architectural choices for sequence modeling in offline reinforcement learning. The results show that xLSTM models scale effectively with training tasks and data, outperforming transformers and mamba in both policy performance and inference speed.

**Strengths:**

* The empirical evaluation is thorough and extensive. THe paper also provides rich experimental details in appendix, providing valuable comparisons across different architectures.
* The release of the data pipeline and dataset can be a useful resource for the community.

**Weaknesses:**

* The paper is built upon the decision transformer approach to offline RL. However, this approach is not widely used in the recent recent developments of large-scale embodied or robotic foundation models. To show that xLSTM is a competative architecture, it would be important to validate its advantages for other learning approaches, e.g., generative behavior cloning.
* The ablation study examines the influence of context length, both including and excluding action; however, it looks that even without actions, all choices of context length converge to similar performance. This result goes against the necessity of using xLSTM or other forms of recurrent models. It would be good to more clearly justify when xLSTM could benefit from extended history.
* In terms of specific architecture designs, the paper considers xLSTM [7:1] and [1:0]. However, the rationale behind these choices is unclear. Moreover, the results look puzzling: the former works better in atari, while the latter works better in procgen. Providing deeper insight into these choices would strengthen the paper.

**Questions:**

In line 380, the authors note `While removing actions improves performance on the robotics domains, it does not affect performance on discrete control. We assume this is because ... by observing previous actions, the agent learns shortcut`.

To verify this explanation, could you provide correlation metrics between sequeantial actions in robotic vs discrete tasks, similar to Fig 3 in [1]?

[1] Fighting Copycat Agents in Behavioral Cloning from Observation Histories, NeurIPS 2020

---

> ### Author Response · Authors · 2024-11-21
>
> Thank you for your helpful feedback. We are glad that you appreciate the experimental details we provide and that you find our data pipeline useful for the community.
>
> **Decision Transformer vs. Behavior Cloning:** We agree with the reviewer that it is important to validate whether our findings translate to the behavior cloning setting. It is true that many recent works focus on behavior cloning without return-conditioning, as used in Decision Transformers. Therefore, we conduct an additional ablation study on the behavior cloning setting at the 206M parameter scale. Indeed, we find that the same trends hold when training models without return-conditioning (see Appendix D.2 of our updated manuscript). Thank you for the suggestion!
>
> **Influence of context length:** It is true that most sequence lengths in Figure 23b tend to result in similar performance. We believe that this is an artifact of the Meta-World environment. Episodes in Meta-World last for 200 steps and, therefore, the effect of longer context may not be visible. To further investigate the effect of the context length, we conduct an additional ablation study for models trained on all 432 tasks and with varying context lengths in Appendix D.1. Indeed, we find that performance improves both for DT and xLSTM as the sequence length increases. For DT, the average normalized score increases from 0.3 with C=1 to 0.7 with C=50 (see Figure 24). Similarly, for xLSTM the normalized score increases from 0.4 to 0.8 (see Figure 26). Furthermore, we find that domains with longer episode lengths, like DMControl or Atari, benefit more from increasing context length than Meta-World (see Figures 25, 27). This result highlights the fact that large action models can strongly benefit from increased context length, even on the simulated environments we consider in this work. We believe that this effect may be even bigger for complex real-world applications that require longer-term interactions.
>
> **mLSTM to sLSTM ratio:** The reviewer is correct that we consider the ratios [7:1] and [1:0] for xLSTM. For example, [7:1] indicates that there is 1 sLSTM block for every 7 mLSTM blocks. We adopted the ratios from the original xLSTM publication [1] (see Appendix B.3 for a discussion on this choice). While mLSTM is fully parallelizable, sLSTM enables state-tracking [2], which can be useful for logic tasks or partially observable environments. To get a better understanding of the effects of this design choice, we conduct additional ablation studies on the ratio in Appendix D.3. We compare different ratios (1:0, 0:1, 1:1, 3:1, 7:1) and different placements of the sLSTM blocks, both on the 432 tasks used in the main paper, and on the partially observable Darkroom environment. While we do not find benefits on the 432 tasks at the 16M model scale (see Figure 29), we do find that different ratios (such as 3:1) can be beneficial on Darkoom, while others hurt performance (see Figure 30). Therefore, the effectiveness of sLSTM layers depends on the task at hand. For more complex tasks with long horizons or partial observability (as is common in real-world applications), sLSTM may be a useful tool for practitioners.
>
> **Removing the action condition:** Thank you for bringing up the “copycat problem” paper [3], which we now refer to in our work. We believe this paper studies a similar problem as we encountered, even though we came up with a different solution. In preliminary experiments, we found that robotics domains were more strongly affected by this problem than discrete control domains (Procgen, Atari). To provide further empirical support for this, we conducted an ablation study on this in Appendix D.1.2. We find that Meta-World and DMControl are strongly affected, but Atari and Procgen are not (see Figure 25). Furthermore, we also found that the robotics domain Composuite is also not affected. Therefore, we compute the average MSEs between subsequent actions in the datasets (similar to [3], see Table 7) and find that the MSE for Composuite is considerably higher. Therefore, we conclude that removing actions from the agent’s context is particularly effective for domains where actions change smoothly (such as Meta-World, DMC). However, note that the final performances on domains “unaffected” by the copycat problem are still considerably worse when training with actions than when training without actions in the context (0.35 vs. 0.7).
>
> We hope to have answered all your questions and believe our manuscript has become substantially better due to your suggestions.
>
> [1] “xLSTM: Extended Long Short-Term Memory”, NeurIPS 2024 \
> [2] “The Illusion of State in State-Space Models”, ICML 2024 \
> [3] Fighting Copycat Agents in Behavioral Cloning from Observation Histories, NeurIPS 2020

---

> ### Comment · Reviewer_Vb2E · 2024-11-27
>
> I appreciate the detailed response from the authors, and agree on the value of this work as large-scale empirical research. However, as noted in my initial review, demonstrating xLSTM's relevance for robotics requires evaluation on learning frameworks that are more widely used in the field.
>
> The new Appendix D.2 is a step in this direction, but it does not provide direct comparisons with *any of the models cited by the author* in the section (L1951): RT-1, RT-2, and Octo. The added experiments are still restricted to offline RL, removing `return` tokens but keeping `state` and `reward` tokens, which fundamentally differs from behavior cloning. Most of the SoTA models in robotics have *no access to reward* and *take only recent observations* (e.g., 1–4 image frames) as input.
>
> Overall, while I value the empirical contributions of this work for xLSTM, its claims on robotics are rather debatable. I therefore consider this work borderline and cannot champion it in its current form.

---

> > ### Author Response · Authors · 2024-11-29
> >
> > We thank the reviewer for the response and are happy that the reviewer sees the value of the work. In addition to Appendix D.2 (Decision Transformer vs. Behavior Cloning), please also consider the new analyses we provided on the influence of context length (Appendix D.1), the ratio of mLSTM to sLSTM blocks (Appendix D.3), and removing the action condition (Appendix D.1.2).
> >
> > We agree that Appendix D.2. is a good step in the direction of validating that our findings hold in the BC setting. We want to address your remaining concerns on this topic:
> >
> >
> >
> > 1. The sequence we considered in this setting contains `state` and `reward` tokens, but we remove the `return-to-go` tokens (see Appendix D.2). Note that in DTs only the RTG tokens are used to condition the model on high-quality actions at inference time. There is no conditioning on rewards. In preliminary experiments, we found that keeping the reward tokens in the sequence does not considerably alter the behavior of the system, compared to when only using state tokens. Therefore, we kept the reward tokens in the sequence for the experiments reported in Appendix D.2. However, to verify this point, we repeat the same experiment with removed `reward` tokens. Indeed, we find that the same performance trends hold for BC. We cannot update our PDF at this point, but are happy to provide the summary table for BC with state tokens only (206M model, performance at 200K):
> >
> > | Method  | Validation PPL  |  Evaluation Performance |
> > |---|:---:|:---:|
> > | Transformer  | 43.26  | 0.623  |
> > |  Mamba       | 42.97  | 0.678  |
> > | xLSTM [1:0]  | 42.93  | 0.696  |
> > | xLSTM [7:1]  | 42.84  |0.707   |
> >
> >
> > 2. The purpose of this work is to better understand whether modern recurrent backbones can be alternatives to the Transformer for large action models. Therefore, we focus our analysis on comparing the different backbones, rather than comparing against methods like RT-1, RT-2 or Octo. The primary difference to these methods is that they train/evaluate on real-robot data, while we conduct our experiments in simulation. While we keep experiments on real robots for future work (see Section 5) we do believe that our results are transferable because the architectures of these methods do not differ substantially from our setup (except for the language instructions). For example, similar to our work, RT-1 uses a separate vision encoder, uses action tokenization and directly predicts actions. RT-2, on the other hand, relies on a large (55B) VLM, making a direct comparison to modern recurrent backbones difficult.
> >
> > We hope to have answered your remaining points. If so, we would be happy if you considered increasing your score in light of our responses and the updated version of our paper.

---

> > > ### Comment · Reviewer_Vb2E · 2024-12-03
> > >
> > > Thank you for your efforts in addressing my questions. I've increased my rating of the submission, given its value as a rigorous and comprehensive large-scale empirical study.
> > >
> > > That said, I still think the lack of a direct comparison with other recent models in robotics, e.g., diffusion- or flow-based policy, trained through imitation learning, remains a significant limitation and thus would not recommend a higher score in its current form.

---

### Official Review · Reviewer_oYy1 · 2024-11-05

**Soundness:** 3
**Presentation:** 3
**Contribution:** 2
**Rating:** 5
**Confidence:** 3

**Summary:**

The paper introduces a Large Recurrent Action Model (LRAM) that utilizes xLSTM to address the limitations of Transformer-based models, specifically their slow inference times, which make them impractical for real-time applications like robotics. The authors claim that LRAM offers linear-time inference complexity and natural sequence length extrapolation capabilities. They support these claims with experiments across various domains and 432 tasks, demonstrating favorable performance and speed compared to Transformers.

**Strengths:**

1. The paper provides a thorough experimental evaluation, covering a wide range of domains and tasks, including common RL benchmarks (Atari, DMControl, etc), robotics simulation environments, and generalization benchmarks.
2. The empirical studies on these benchmarks, especially the comparison between Transformer-based models and LSTM-based models, have brought some useful practices to the community.

**Weaknesses:**

1. From current status,  the contribution of this paper may be limited. I appreciate the comprehensive experiments in this paper, while the use of xLSTM for faster inference is not surprising, as LSTM-based models are typically faster than Transformer-based models.
2. I think it would be better if the paper discuss when and why to use LSTM-based models versus Transformer-based models. If the primary goal is to improve the inference speed of Transformers, there are already several engineering solutions, such as FlashAttention, that can significantly accelerate Transformer inference.
3. From Figure 6, it appears that the inference speed of Decision Transformers (DT) is acceptable when the batch size (B) is 1, which suggests that the need for an alternative model might be less urgent in some scenarios. As far as I known, in real-world applications, most robotics models perform inference at batch size 1. I suggest the authors do some investigation into scenarios where LSTM-based models are most beneficial.
4.I did some comparative experiments between Transformer and LSTMs architectures. My own visualization experiments suggest that Transformers and LSTMs have different focus points. Transformers tend to have longer memory, while LSTMs often focus on recent states. For more challenging tasks, such as those with sparse rewards, Transformer-based models may perform better due to their ability to handle long-term dependencies, which is consistent to the original DT paper. Maybe the authors could do some similar visualization experiments to investigate the difference among vanilla LSTM, xLSTM and Transformers on sparse and dense reward tasks, respectively.

**Questions:**

I suggest the authors provide a more detailed comparison with other recurrent architectures and explain why xLSTM is a superior choice for this specific application. Additionally, the authors may offer more insights into the trade-offs between LSTM-based and Transformer-based models, and under what conditions each is more suitable.
This may not be a question, but I think it will be valuable to the community if the paper can further investigate the above point.

---

> ### Author Response · Authors · 2024-11-21
>
> Thank you for your helpful feedback on our work. We are glad that you find that our empirical studies bring some useful practices to the community. We address your open points in the following.
>
> **When and why to use recurrent models?** We agree with the reviewer that an additional discussion on LSTM-based versus Transformer-based models can be useful, and we added the discussion to our updated manuscript. Regarding the inference speed, we want to highlight that we do compare against FlashAttention in our experiments. While FlashAttention speeds up the computation of the self-attention, the computational complexity remains quadratic with the sequence length in terms of speed and memory. In contrast, for xLSTM and Mamba the complexity is linear. We observe benefits in inference speed (Figure 6) and importantly also in memory consumption (Figure 7). Consequently, we believe that xLSTM can be a more efficient alternative to Transformer, for example for applications on edge devices. Furthermore, xLSTM enables state tracking [1] via the sLSTM block, which Transformers and Mamba cannot. This property can be particularly useful for partially observable environments. To provide more insights on this, we conduct additional ablations on the ratio of mLSTM to sLSTM blocks in Appendix D3 of our updated manuscript. On the other hand, Transformers may be particularly effective for applications that require exact recall of memories [2]. Consequently, the “best” choice among architectures is a question the community is still exploring, and may depend on the task at hand.
>
> **Inference time comparisons:** We agree with the reviewer that batch size 1 is commonly used for inference of real-world robotics models. For B=1, xLSTM is almost twice as fast as the Transformer, which makes use of highly optimized FlashAttention (Figure 6). Importantly, the inference speed and memory consumption of xLSTM does not change with the sequence length (Figure 7). Note that we conduct this comparison on a high-end data-center GPU with 40GB of RAM, and the Transformer goes OOM for larger context sizes. In contrast, applications on edge devices, like robotics, may have to deal with less powerful accelerators. Therefore, the recurrent backbone enables processing significantly longer sequence lengths without increased requirements in terms of speed and memory. This is important because this property enables handling high sample rates as prevalent in real-world applications, or multi-episodic context as used for in-context RL (see Lines 72-78). To highlight the benefits of longer context, we conduct additional ablation studies in Appendix D.1 of our updated manuscript.
>
> **Differences of recurrent vs. Transformer.** We agree with the reviewer that sparse reward tasks are interesting to study the difference between recurrent backbones and Transformers. We do conduct preliminary experiments on Dark-Room, which exhibits sparse rewards and a partially-observable observation space. We find that xLSTM compares favorably to Transformers on Dark-Room (see Figure 4 and 17 of our updated manuscript). We also agree that additional experiments in this direction could be of value. While we consider them out-of-scope for this work, we aim to investigate this in future work. However, for a similar comparison of vanilla LSTM to the Transformer, we refer to [2]. Furthermore, we refer to the xLSTM publication [3] Figure 5, for a comparison of xLSTM and Transformer on associative recall tasks. We added a discussion on both references to our updated manuscript.
>
> Our manuscript improved considerably by addressing and incorporating your comments: thank you.
>
> [1] “The Illusion of State in State-Space Models”, ICML 2024 \
> [2] “When Do Transformers Shine in RL? Decoupling Memory from Credit Assignment”, NeurIPS 2023

---

> > ### Author Response · Authors · 2024-12-02
> >
> > We thank the reviewer again for taking the time and effort to help improve our paper. As the end of the author-reviewer discussion period is near, we are reaching out to ask you if we have addressed your remaining concerns. If you have any remaining open questions, we would be happy to engage in further discussion. If not, we would highly appreciate it if you considered raising your score in light of our responses and the new version of our paper.

---

### Author Response · Authors · 2024-11-21

Dear Reviewers,

We thank you for your helpful comments and constructive feedback! We addressed your questions and comments in our individual responses. Your comments convinced us to perform additional experiments, which we report in the revised manuscript.
We believe that these changes considerably improved our paper - thank you! We appeal to the reviewers to re-check their assessment in light of our responses and the new version of our paper.

Kind Regards

---

### Meta-Review · Area_Chair_ksDp · 2024-12-08

**Metareview:**

This work conducts an empirical study of using xLSTM as a large-scale recurrent neural networks for sequence modeling in offline RL. The experimental result shows that the xLSTM models scale better when training with more tasks and tasks, outperforming transformers and mamba in performance and inference speed.

After rebuttal and discussion, this submission receives review scores of 5,6,6,6. So it is a borderline paper.

Since it is an experiment paper, reviewers mainly raised concerns about more comparisons and more experiments, which the authors did a good job addressing.

The reviewer who gave a 5 score argued that the contribution of this paper may be limited, which is echoed by the AC.

AC has checked the submission, the reviews, and the discussions. AC appreciated the comprehensive experiments done by the authors in this paper. However, AC couldn't find much depth and intellectual merits in this submission. This work is indeed a wonderful experimental report, but AC expected an ICLR paper also needs to go beyond an experimental report and exhibit some intellectual novelty or methodology contributions.   For example, after investigating the architecture comparison, the authors can offer more insights into the trade-offs, strengths, and weaknesses of LSTM and Transformer models under certain conditions and develop a new hybrid method to take the best of the two worlds. Furthermore, no real-world robotics tasks were evaluated in the submission. It remains unknown how much this evaluated architecture can generalize to the real world.

Given the above reasons, AC recommends a rejection.

AC has communicated this decision with the Senior AC, and the Senior AC has gone through the submission and discussion and confirmed the AC's rejection decision. AC and senior AC believe the paper would be a better fit for a robotics conference, where the robotics community will be excited about applying existing but new architecture to robotic tasks.

**Additional Comments On Reviewer Discussion:**

There were discussions between the authors and reviewers. However, adding more discussions, experiments, and clarification won't change the nature of this work as a large-scale experimental report comparing different existing neural architectures for offline RL tasks. The intellectual merits and insights are still lacking.

---

### Decision · Program_Chairs · 2025-01-22

Reject